# Bilinear Exponential Family of MDPs: Frequentist Regret Bound with Tractable Exploration & Planning

## Abstract

We study the problem of episodic reinforcement learning in continuous state-action spaces with unknown rewards and transitions. Specifically, we consider the setting where the rewards and transitions are modeled using parametric bilinear exponential families. We propose an algorithm, `BEF-RLSVI`, that a) uses penalized maximum likelihood estimators to learn the unknown parameters, b) injects a calibrated Gaussian noise in the parameter of rewards to ensure exploration, and c) leverages linearity of the exponential family with respect to an underlying RKHS to perform tractable planning. We further provide a frequentist regret analysis of `BEF-RLSVI` that yields an upper bound of $\tilde{\mathcal{O}}(\sqrt{d^3 H^3 K})$, where $d$ is the dimension of the parameters, $H$ is the episode length, and $K$ is the number of episodes. Our analysis improves the existing bounds for the bilinear exponential family of MDPs by $\sqrt{H}$ and removes the handcrafted clipping deployed in existing `RLSVI`-type algorithms. Our regret bound is order-optimal with respect to $H$ and $K$.

## 1 Introduction

Reinforcement Learning (RL) is a well-studied and popular framework for sequential decision making, where an agent aims to compute a *policy* that allows her to maximize the accumulated reward over a horizon by interacting with an *unknown* environment [SB18].

**Episodic RL.** In this paper, we consider the episodic finite-horizon MDP formulation of RL, in short *Episodic RL* [ORVR13, AOM17, DLB17]. Episodic RL is a tuple $\mathcal{M} = \langle \mathcal{S}, \mathcal{A}, \mathbb{P}, r, K, H \rangle$, where the state (resp. action) space $\mathcal{S}$ (resp. $\mathcal{A}$) might be continuous. In episodic RL, the agent interacts with the environment in episodes consisting of $H$ steps. Episode $k$ starts by observing state $s_1^k$. Then, for $t = 1, \dots H$, the agent draws action $a_t^k$ from a (possibly time-dependent) policy $\pi_t(s_t^k)$, observes the reward $r(s_t^k, a_t^k) \in [0, 1]$, and transits to a state $s_{t+1}^k \sim \mathbb{P}(. \mid s_t^k, a_t^k)$ according to the transition function $\mathbb{P}$. The performance of a policy $\pi$ is measured by the total expected reward $V_1^\pi$ starting from a state $s \in \mathcal{S}$, the value function and the state-action value functions at step $h \in [H]$ are defined as

$$V_h^\pi(s) \stackrel{\text{def}}{=} \mathbb{E}\left[\sum_{t=h}^{H} r(s_t, a_t) \mid s_h = s\right], \quad \text{and} \quad Q_h^\pi(s, a) \stackrel{\text{def}}{=} \mathbb{E}\left[\sum_{t=h}^{H} r(s_t, a_t) \mid s_h = s, a_h = a\right].$$

Here, computing the policy leading to maximization of cumulative reward requires the agent to strategically control the actions in order to learn the transition functions and reward functions as precisely as required. This tension between learning the unknown environment and reward maximization is quantified as *regret*: the typical performance measure of an episodic RL algorithm. *Regret* is defined as the difference between the *expected cumulative reward* or *value* collected by the optimal agent that knows the environment and the expected cumulative reward or value obtained by

an agent that has to learn about the unknown environment. Formally, the regret over $K$ episodes is

$$\mathcal{R}(K) \triangleq \sum_{k=1}^{K} \left( V_1^{\pi^*}(s_1^k) - V_1^{\pi_t}(s_1^k) \right).$$

**Key Challenges.** *The first key challenge in episodic RL is to tackle the exploration–exploitation trade-off.* This is traditionally addressed with the *optimism principle* that either carefully crafts optimistic upper bounds on the value (or state-action value) functions [AOM17], or maintains a posterior on the parameters to perform posterior sampling [ORVR13], or perturbs the value (or state-action value) function estimates with calibrated noise [OVRW16]. Though the first two approaches induce theoretically optimal exploration, they might not yield tractable algorithms for large/continuous state-action spaces as they either involve optimization in the optimistic set or maintaining a high-dimensional posterior. Thus, *we focus on extending the third approach of* Randomized Least-Square Value Iteration (RLSVI) *framework, and inject noise only in rewards to perform tractable exploration.*

*The second challenge*, which emerges *for continuous state-action spaces, is to learn a parametric functional approximation of either the value function or the rewards and transitions* in order to perform planning and exploration. Different functional representations (or models), such as linear [JYWJ20], bilinear [DKL+21], and bilinear exponential families [CGM21], are studied in literature to develop optimal algorithms for episodic RL with continuous state-action spaces. Since the linear assumption is restrictive in real-life -where non-linear structures are abundant-, generalized representations have obtained more attention recently [CGM21, LLS+21, DKL+21, FKQR21]. The bilinear exponential family model is of special interest as it is expressive enough to represent tabular MDPs (discrete state-action), factored MDPs [KK99], linear MDPs [JYWJ20], linearly controlled dynamical systems (such as Linear Quadratic Regulators [AYS11]) as special cases [CGM21]. Thus, in this paper, *we study the bilinear exponential family of MDPs, i.e. the episodic RL setting where the rewards and transition functions can be modelled with bilinear exponential families.*

*The third challenge is to perform tractable planning[1] given the perturbation for exploration and the model class.* Existing work [OVR14, CGM21] assumes an oracle to perform planning and yield policies that aren't explicit. The main difficulty in such planning approaches is that dynamic programming requires calculating $\int \mathbb{P}(s' \mid s, a) V_h(s)$ for all $(s, a)$ pairs. This is not trivial unless the transition is assumed to be linear and decouples $s'$ from $(s, a)$, which is not known to hold except for tabular MDPs. Much ink has been spilled about this challenge recently, *e.g.* [DKWY19] asks when misspecified linear representations are enough for a polynomial sample complexity in several settings. [SS20, LSW20, VRD19] provide positive answers for specific linear settings. In this paper, *we aim to address this issue by designing a tractable planner for the bilinear exponential family representation.*

In this paper, we aim to address the following question that encompasses the three challenges:

Can we design an algorithm that performs **tractable exploration** and **planning** for *bilinear exponential family of MDPs* yielding a **near-optimal frequentist regret bound**?

**Our Contributions.** Our contributions to this question are three-fold.

1. *Formalism:* We assume that neither rewards nor transitions are known, whereas existing efforts on the bilinear exponential family of MDPs assume knowledge of rewards. This makes the addressed problem harder, practical, and more general. We also observe that though the transition model can represent non-linear dynamics, it implies a linear behavior (see Section 2) in a Reproducible Kernel Hilbert Space (RKHS). This observation contributes to the tractability of planning.

2. *Algorithm:* We propose an algorithm BEF-RLSVI that extends the RLSVI framework to bilinear exponential families (see Section 3). BEF-RLSVI a) injects calibrated Gaussian noise in the rewards to perform exploration, b) leverages the linearity of the transition model with respect to an underlying RKHS to perform tractable planning and c) uses penalized maximum likelihood estimators to learn the parameters corresponding to rewards and transitions (see Section 4). To the best of our knowledge, *BEF-RLSVI is the first algorithm for the bilinear exponential family of MDPs with tractable exploration and planning under unknown rewards and transitions.*

---

[1]By tractable planning, we mean having a planner with (pseudo-)polynomial complexity in the problem parameters, i.e. dimension of parameters, dimension of features, horizon, and number of episodes.

Table 1: A comparison of RL Algorithms for continuous state-actions with functional representations.

| Algo | Regret | Tractable exploration | Tractable planning | Free of clipping | Model, assumptions |
|---|---|---|---|---|---|
| Thompson sampling [RZSD21] | $\sqrt{d^2 H^3 K}$ (Bayesian) | ✗ | ✓ | N.A | Gaussian $\mathbb{P}$ Known rewards |
| LSVI-PHE [ICN$^+$21] | $\sqrt{d^3 H^4 K}$ (Freq.) | ✓ | ✓ | ✗ | Generalized $V$ approx Tabular, anti-concentration |
| OPT-RLSVI [ZBB$^+$20] | $\sqrt{d^4 H^5 K}$ (Freq.) | ✓ | ✓ | ✗ | Linear $V$ |
| EXP-UCRL [CGM21] | $\sqrt{d^2 H^4 K}$ (Freq.) | ✗ | ✗ | N.A | Bilinear Exp family known rewards |
| BEF-RLSVI This work | $\sqrt{d^3 H^3 K}$ (Freq.) | ✓ | ✓ | ✓ | Bilinear Exp family |

3. *Analysis:* We carefully develop an analysis of BEF-RLSVI that yields $\tilde{\mathcal{O}}(\sqrt{d^3 H^3 K})$ regret which improves the existing regret bound for bilinear exponential family of MDPs with known reward by a factor of $\sqrt{H}$ (Section 3.2). Our analysis (Section 5) builds on existing analyses of RLSVI-type algorithms [OVRW16], but contrary to them, we remove the need to handcraft a clipping of the value functions [ZBB$^+$20]. We also do not need to *assume* anti-concentration bounds as we can explicitly control it by the injected noise. This was not done previously except for the linear MDPs. We illustrate this comparison in Table 1. We highlight three technical tools that we used to improve the previous analyses: 1) Using transportation inequalities instead of the simulation lemma reduces a $\sqrt{H}$ factor compared to [RZSD21], 2) Leveraging the observation that true value functions are bounded enables using an improved elliptical lemma (compared to [CGM21]), and 3) Noticing that the norm of features can only be large for a finite amount of time allows us to forgo clipping and reduce a $\sqrt{d}$ factor from the regret compared to [ZBB$^+$20].

## 2 Bilinear exponential family of MDPs

In this section, we introduce the bilinear exponential family model coined in [CGM21] and extend it to parametric rewards. Then, we state a novel observation about linearity of this representation.

**Bilinear exponential family model.** We consider both transition and reward kernels to be unknown and modeled with bilinear exponential families. Specifically,

$$\mathbb{P}\left(\tilde{s} \mid s, a\right) = \exp\left(\psi(\tilde{s})^\top M_{\theta^{\mathrm{p}}} \varphi(s,a) - Z_{s,a}^{\mathrm{p}}(\theta^{\mathrm{p}})\right), \tag{1}$$

$$\mathbb{P}\left(r \mid s, a\right) = \exp\left(r\, B^\top M_{\theta^{\mathrm{r}}} \varphi(s,a) - Z_{s,a}^{\mathrm{r}}(\theta^{\mathrm{r}})\right), \tag{2}$$

where $\varphi \in (\mathbb{R}_+^q)^{\mathcal{S} \times \mathcal{A}}$ and $\psi \in (\mathbb{R}_+^p)^{\mathcal{S}}$ are known feature functions, and $B \in \mathbb{R}^p$ is a known scaling factor. The unknown reward and transition parameters are $\theta^{\mathrm{p}}, \theta^{\mathrm{r}} \in \mathbb{R}^d$. $M_{\theta^\cdot} \stackrel{\text{def}}{=} \sum_{i=1}^d \theta_i A_i$, where $A_i$ is a known $p \times q$ matrix for each $i$. Finally, $Z$ denotes the log partition function:

$$Z_{s,a}^{\mathrm{p}}(\theta^{\mathrm{p}}) \stackrel{\text{def}}{=} \log \int_{\mathcal{S}} \exp\left(\psi(\tilde{s})^\top M_{\theta^{\mathrm{p}}} \varphi(s,a)\right) d\tilde{s},$$

$Z^{\mathrm{r}}$ is defined similarly. We denote $V_{\theta^{\mathrm{p}}, \theta^{\mathrm{r}}, h}^\pi$, respectively $Q_{\theta^{\mathrm{p}}, \theta^{\mathrm{r}}, h}^\pi$, the value, respectively state-action value function for policy $\pi$ in the MDP parameterized by $(\theta^{\mathrm{p}}, \theta^{\mathrm{r}})$ at time $h$. A policy $\pi^\star$ is *optimal* if for all $s \in \mathcal{S}$, $V_{\theta,h}^{\pi^\star}(s) = \max_{\pi \in \Pi} V_{\theta,h}^\pi(s)$. A learning algorithm minimizes the (pseudo-)regret defined as:

$$\mathcal{R}(K) \triangleq \sum_{k=1}^K \left( V_{\theta,1}^{\pi^\star}(s_1^k) - V_{\theta,1}^{\pi^t}(s_1^k) \right). \tag{3}$$

**Linearity of transitions.** Now, we state an observation about the bilinear exponential family and discuss how it helps with the challenge of planning in episodic RL. Specifically, the popular assumption of linearity of the transition kernel is a direct consequence of our model. Indeed,

$$2\psi\left(s'\right)^\top M_{\theta^{\mathrm{p}}} \varphi(s,a) = -\|(\psi(s') - M_{\theta^{\mathrm{p}}} \varphi(s,a)\|^2 + \|\psi(s')\|^2 + \|M_{\theta^{\mathrm{p}}} \varphi(s,a)\|^2.$$

Notice that the quadratic term resembles the Radial Basis Function (RBF) kernel. More precisely, for an RBF kernel with covariance $\Sigma = I_p$ and $k(x,y) \overset{\mathrm{def}}{=} \exp\left(-\|x-y\|^2/2\right)$, we find

$$\mathbb{P}\left(s' \mid s, a\right) = \langle \phi^{\mathrm{p}}(s,a), \mu^{\mathrm{p}}(s') \rangle_{\mathcal{H}}, \tag{4}$$

where $\mathcal{H}$ is the RKHS associated with the kernel, $\mu^{\mathrm{p}}(s') = (2\pi)^{-p/2} k\left(\psi(s'),.\right) \exp\left(\|\psi\left(s'\right)\|^2/2\right)$, and $\phi^{\mathrm{p}}(s,a) = k\left(M_{\theta^{\mathrm{p}}}^{\top}\varphi(s,a),.\right) \exp\left(\|M_{\theta^{\mathrm{p}}}\varphi(s,a)\|^2/2 - Z_{s,a}(\theta^{\mathrm{p}})\right)$. Equation (4) shows that $s'$ is decoupled from $(s,a)$, we see hereafter why this is crucial to reducing the complexity of planning.

**Remark.** *Up to our knowledge, [RZSD21] is the only work providing an example of linear transition kernel for RL with continuous state-action spaces. They consider Gaussian transitions with an unknown mean ($f^\star(s,a)$) and known variance ($\sigma^2$). Actually, linear $f^\star$ is a special case of the bilinear exponential family model, where $\psi(s') = (s', \|s'\|^2)$ and $M_\theta\varphi(s,a) = (f_\theta(s,a)/\sigma^2, -1/\sigma^2)$.*

**Importance of linearity.** To understand the planning challenge in RL, recall the Bellman equation:

$$Q_h^\pi(s,a) = r(s,a) + \int_{\tilde{s} \in \mathcal{S}} P(s' \mid s, a) V_{h+1}^\pi(\tilde{s}) d\tilde{s},$$

We must approximate the integral at the R.H.S.for $(s,a) \in \mathcal{S} \times \mathcal{A}$. For a tabular MDP with $|S|$ states and $|A|$ actions, we need to evaluate $(Q_h^\pi)_{h \in [H]}$, i.e. to approximate $|S| \times |A| \times H$ integrals per episode, which can be very expensive. However, if the transition model is linear (Equation (4)), then

$$Q_{\theta,h}^\pi(s,a) = r(s,a) + \left\langle \phi^{\mathrm{p}}(s,a), \int_{\mathcal{S}} \mu^{\mathrm{p}}(\tilde{s}) V_{\theta,h+1}^\pi(\tilde{s}) d\tilde{s} \right\rangle. \tag{5}$$

When $\phi^{\mathrm{p}}, \mu^{\mathrm{p}} \in \mathbb{R}^\tau$, we can obtain $Q_{\theta^{\mathrm{p}},\theta^{\mathrm{r}},h}$ by computing $\tau$ integrals per timestep, reducing the state-action space complexity to $\tau$ only. For our model, although $\phi^{\mathrm{p}}$ and $\mu^{\mathrm{p}}$ are infinite dimensional, we show in Section 4 (§ planning) that the planning complexity is still significantly reduced.

# 3 `BEF-RLSVI`: algorithm design and frequentist regret bound

In this section, we formally introduce the Bilinear Exponential Family Randomized Least-Squares Value Iteration (`BEF-RLSVI`) algorithm along with a high probability upper-bound on its regret.

## 3.1 `BEF-RLSVI`: algorithm design

`BEF-RLSVI` is based on `RLSVI` [OVRW16] framework with the distinction that we only perturb the reward parameters and not all the parameters of the value function. `RLSVI` algorithms are reminiscent of Thompson Sampling, yet more tractable with better control over the probability to be optimistic.

---

**Algorithm 1** `BEF-RLSVI`

1: **Input:** failure rate $\delta$, constants $\alpha^{\mathrm{p}}, \eta$ and $(x_k)_{k \in [K]} \in \mathbb{R}^+$
2: **for** episode $k = 1, 2, \ldots$ **do**
3:    Observe initial state $s_1^k$
4:    Sample noise $\xi_k \sim \mathcal{N}\left(0, x_k(\bar{G}_k^{\mathrm{p}})^{-1}\right)$ such that
$$\bar{G}_k^{\mathrm{p}} = \frac{\eta}{\alpha^{\mathrm{p}}}\mathbb{A} + \sum_{\tau=1}^{k-1}\sum_{h=1}^{H}(\varphi(s_h^\tau, a_h^\tau)^\top A_i^\top A_j \varphi(s_h^\tau, a_h^\tau))_{i,j \in [d]}$$
5:    Perturb reward parameter: $\tilde{\theta}^{\mathrm{r}}(k) = \hat{\theta}^{\mathrm{r}}(k) + \xi_k$
6:    Compute $(Q_{\hat{\theta}^{\mathrm{p}},\tilde{\theta}^{\mathrm{r}},h}^k)_{h \in [H]}$ via Bellman-backtracking, see Algorithm 2
7:    **for** $h = 1, \ldots, H$ **do**
8:       Pull action $a_h^k = \arg\max_a Q_{\hat{\theta}^{\mathrm{p}},\tilde{\theta}^{\mathrm{r}},h}(s_h^k, a)$
9:       Observe reward $r(s_h^k, a_h^k)$ and state $s_{h+1}^k$.
10:    **end for**
11:    Update the penalized ML estimators $\hat{\theta}^{\mathrm{p}}(k), \hat{\theta}^{\mathrm{r}}(k)$, see Equation (6) and Equation (8)
12: **end for**

---

We can see that Algorithm 1 performs exploration by a Gaussian perturbation of the reward parameter (Line 4). Contrary to optimistic approaches, this method is explicit and also more efficient since it does not a involve high-dimensional optimization.

---
**Algorithm 2** Bellman Backtracking
---
1: **Input** Parameters $\hat{\theta}^{\mathrm{p}}, \tilde{\theta}^{\mathrm{r}}$, initialize $\tilde{\theta} = (\tilde{\theta}^{\mathrm{r}}, \hat{\theta}^{\mathrm{p}})$ and $\forall s, V_{H+1}(s) = 0$
2: **for** steps $h = H - 1, H - 2, \cdots, 0$ **do**
3:     Calculate $Q_{\tilde{\theta}, h}(s, a) = \mathbb{E}_{s,a}^{\tilde{\theta}^{\mathrm{r}}}[r] + \langle \phi^{\mathrm{p}}(s, a), \int V_{\tilde{\theta}, h+1}(s') \mu^{\mathrm{p}}(s') ds' \rangle_{\mathcal{H}}$.
4: **end for**
---

We can approximate Line 3 of Algorithm 2 with $\mathcal{O}(pH^3 K \log(HK))$ complexity and without harming the learning process (*cf.* § planning, Section 4). Therefore, here, planning is tractable.

## 3.2 `BEF-RLSVI`: regret upper-bound

We state the standard smoothness assumptions on the model [CGM21, JBNW17, LMT21].

**Assumption 1.** *There exist constants $\alpha^{\mathrm{p}}, \alpha^{\mathrm{r}}, \beta^{\mathrm{p}}, \beta^{\mathrm{r}} > 0$, such that the representation model satisfies:*

$$\forall (s, a) \in \mathcal{S} \times \mathcal{A}, \forall \theta, x \in \mathbb{R}^d \quad \alpha^{\mathrm{p}} \leq x^\top C_{s,a}^\theta[\psi] x \leq \beta^{\mathrm{p}}$$

$$\forall (s, a) \in \mathcal{S} \times \mathcal{A}, \forall \theta, x \in \mathbb{R}^d \quad \alpha^{\mathrm{r}} \leq \mathbb{V}\mathrm{ar}_{s,a}^\theta(r) \, x^\top B^\top B x \leq \beta^{\mathrm{r}}$$

*where* $\mathbb{C}_{s,a}^\theta[\psi(s')] \triangleq \mathbb{E}_{s' \sim \mathbb{P}_\theta|s,a}\left[\psi(s')\psi(s')^\top\right] - \mathbb{E}_{s' \sim \mathbb{P}_\theta|s,a}[\psi(s')] \mathbb{E}_{s' \sim \mathbb{P}_\theta|s,a}\left[\psi(s')^\top\right]$ *and*

$\mathbb{V}\mathrm{ar}_{s,a}^\theta(r) \triangleq \left(\mathbb{E}_{s,a}^\theta[r^2] - \mathbb{E}_{s,a}^\theta[r]^2\right)$ *is the variance of the reward under $\theta$.*

A closer look at the derivatives of the model (see Appendix D.3) tells us that previous inequalities directly imply a control over the eigenvalues of the Hessian matrices of the log-normalizers.

We now state our main result, the regret upper-bound of `BEF-RLSVI`.

**Theorem 2** (Regret bound). *Let $\mathbb{A} \triangleq (\mathrm{tr}(A_i A_j^\top))_{i,j \in [d]}$ and $G_{s,a} \triangleq (\varphi(s, a)^\top A_i^\top A_j \varphi(s, a))_{i,j \in [d]}$.*
*Under Assumption 1 and further considering that*

*1. $\max\{\|\theta^{\mathrm{r}}\|_{\mathbb{A}}, \|\theta^{\mathrm{p}}\|_{\mathbb{A}}\} \leq B_{\mathbb{A}}$, $\|\mathbb{A}^{-1} G_{s,a}\| \leq B_{\varphi, \mathbb{A}}$ and $\mathbb{E}_{\theta^{\mathrm{r}}}[r(s, a)] \in [0, 1]$ for all $(s, a)$.*

*2. noise $\xi_k \sim \mathcal{N}(0, x_k(\bar{G}_k^{\mathrm{p}})^{-1})$ satisfies $x_k \geq \left(H\sqrt{\frac{\beta^{\mathrm{p}} \beta^{\mathrm{p}}(K, \delta)}{\alpha^{\mathrm{p}} \alpha^{\mathrm{r}}}} + \frac{\sqrt{\beta^{\mathrm{r}} \beta^{\mathrm{r}}(K, \delta) \min\{1, \frac{\alpha^{\mathrm{p}}}{\alpha^{\mathrm{r}}}\}}}{2\alpha^{\mathrm{r}}}\right)^2 \propto dH^2$,*

*then for all $\delta \in (0, 1]$, with probability at least $1 - 7\delta$,*

$$\mathcal{R}(K) \leq \sqrt{KH}\left[\underbrace{2H\left(\sqrt{\frac{2\beta^{\mathrm{p}}}{\alpha^{\mathrm{p}}}}\beta^{\mathrm{p}}(K, \delta)\gamma_K^{\mathrm{p}} + (1 + \sqrt{\gamma_K^{\mathrm{r}}})\sqrt{\log(1/\delta^2)}\right)}_{\text{Transition concentration} \approx dH} + \underbrace{\beta^{\mathrm{r}}\sqrt{\frac{\beta^{\mathrm{r}}(n, \delta)\gamma_K^{\mathrm{r}}}{2\alpha^{\mathrm{r}}}}}_{\text{Reward concentration} \approx d}\right.$$

$$+ \underbrace{c\beta^{\mathrm{r}}\sqrt{x_K d\gamma_K^{\mathrm{r}} \log(dK/\delta)} + \frac{\beta^{\mathrm{r}}\sqrt{x_K d\gamma_K^{\mathrm{r}} \log(e/\delta^2)}}{\Phi(-1)}(1 + \sqrt{\log(d/\delta)})}_{\text{Noise concentration} \approx d^{3/2}H}\right]$$

$$+ \sqrt{H\gamma_K^{\mathrm{r}}}\left[\underbrace{\beta^{\mathrm{r}} C_d\left(\sqrt{\frac{\beta^{\mathrm{r}}(K, \delta)}{2\alpha^{\mathrm{r}}}} + c\sqrt{x_K d\log(dK/\delta)}\right)}_{\text{Estimation error for no clipping} \approx dH}\right.$$

$$\left.+ \underbrace{\frac{\beta^{\mathrm{r}} d\sqrt{x_K}}{\Phi(-1)}(1 + \sqrt{\log(d/\delta)})\sqrt{C_d\left(1 + \frac{\alpha^{\mathrm{r}} B_{\varphi, A} H}{\eta}\right)}}_{\text{Learning error for no clipping} \approx (dH)^{3/2}}\right],$$

*where for $i \in [\mathbf{p}, \mathbf{r}]$, $\beta^i(K, \delta) \triangleq \frac{\eta}{2}B_{\mathbb{A}}^2 + \gamma_K^i + \log(1/\delta)$, and $\gamma_K^i \triangleq d\log(1 + \frac{\beta^i}{\eta}B_{\varphi, \mathbb{A}} HK)$. Also,*
*$C_d \triangleq \frac{3d}{\log(2)}\log\left(1 + \frac{\alpha^{\mathrm{r}}\|\mathbb{A}\|_2^2 B_{\varphi, \mathbb{A}}^2}{\eta\log(2)}\right)$, $\Phi$ is the Gaussian CDF, and $c$ is a universal constant.*

Theorem 2 entails a regret $\mathcal{R}(K) = \mathcal{O}(\sqrt{d^3 H^3 K})$ for `BEF-RLSVI`, where $d$ is the number of parameters of the bilinear exponential family model, $K$ is the number of episodes, and $H$ is the horizon of an episode. We now clarify how this contrasts with related literature.

145 *Comparison with Other Bounds.* The closest work to ours is [CGM21] as it considers the same
146 model for transitions but with known rewards. They propose a UCRL-type and PSRL-type algorithm,
147 which achieve a regret of order $\widetilde{O}(\sqrt{d^2 H^4 K})$. There are two notable algorithmic differences with
148 our work. First, they do exploration using intractable-optimistic upper bounds or high-dimensional
149 posteriors, while we do it with explicit perturbation. The second difference is in planning. While
150 they assume access to a planning oracle, we do it explicitly with pseudo-polynomial complexity
151 (Section 4). Moreover, we improve the regret bound by a $\sqrt{H}$ factor thanks to an improved analysis,
152 (*cf.* Lemma 18). But similar to all RLSVI-type algorithms, we pick up an extra $\sqrt{d}$ (*cf.* [AL17]).

153 [ZBB+20] proposes a variant of RLSVI for continuous state-action spaces, where there are low-rank
154 models of transitions and rewards. They show a regret bound $R(K) = \widetilde{O}(\sqrt{d^4 H^5 K})$, which is larger
155 than that of BEF-RLSVI by $O(\sqrt{d H^2})$. In algorithm design, we improve on their work by removing
156 the need to carefully clip the value function. Analytically, our model allows us to use transportation
157 inequalities (*cf.* Lemma 13) instead of the simulation lemma, which saves us a $\sqrt{H}$ factor.

158 [RZSD21] considers Gaussian transitions, i.e. $s' = f^*(s, a) + \epsilon$ such that $\epsilon \sim \mathcal{N}\left(0, \sigma^2\right)$. This is a
159 particular case of our model. They propose to use Thompson Sampling, and have the merit of being
160 the first to have observed linearity of the value function from this transition structure. But they do not
161 connect it to the finite dimensional approximation of [RR07] unlike us (Section 4). Finally, they show
162 a Bayesian regret bound of $O(\sqrt{d^2 H^3 K})$. This notion of regret is weaker than frequentist regret,
163 hence this result is not directly comparable with Theorem 2.

164 *Tightness of Regret Bound.* A lower bound for episodic RL with continuous state-action spaces is
165 still missing. However, for tabular RL, [DMKV21] proves a lower bound of order $\Omega(\sqrt{H^3 SAK})$.
166 If we represent a tabular MDP in our model, we would need $d = S^2 \times A$ parameters (Section 4.3,
167 [CGM21]). In this case, our bound becomes $R(K) = O(\sqrt{(S^2 A)^3 H^3 K})$, which is clearly not tight
168 is $S$ and $A$. This is understandable due to the relative generality of our setting. We are however
169 positively surprised that **our bound is tight in terms of its dependence on $H$ and $K$.**

# 4   Algorithm design: building blocks of BEF-RLSVI

171 We present necessary details about BEF-RLSVI and discuss the key algorithm design techniques.

172 **Estimation of parameters.** We estimate transitions and rewards from observations similar to
173 EXP-UCRL [CGM21], *i.e.* by using a penalized maximum likelihood estimator

$$\hat{\theta}^{\mathrm{p}}(k) \in \argmin_{\theta \in \mathbb{R}^d} \sum_{t=1}^{k} \sum_{h=1}^{H} - \log \mathbb{P}_\theta \left( s^t_{h+1} \mid s^t_h, a^t_h \right) + \eta \operatorname{pen}(\theta).$$

174 Here, $\operatorname{pen}(\theta)$ is a trace-norm penalty: $\operatorname{pen}(\theta) = \frac{1}{2}\|\theta\|_{\mathbb{A}}$ and $\mathbb{A} = (\operatorname{tr}(A_i A_j^\top))_{i,j}$. By properties of
175 the exponential family, the penalized maximum likelihood estimator verifies, for all $i \leq d$:

$$\sum_{t=1}^{k} \sum_{h=1}^{H} \left( \psi\left(s^t_{h+1}\right) - \mathbb{E}^{\hat{\theta}^{\mathrm{p}}_k}_{s^t_h, a^t_h} \left[\psi\left(s'\right)\right] \right)^\top A_i \varphi\left(s^t_h, a^t_h\right) = \eta \nabla_i \operatorname{pen}\left(\hat{\theta}^{\mathrm{p}}_k\right). \tag{6}$$

176 Equation (6) can be solved in closed form for simple distributions, like Gaussian, but it can involve
177 integral approximations for other distribution. We estimate the parameter for reward, *i.e.* $\theta_r$, similarly

$$\hat{\theta}^{\mathrm{r}}(k) \in \argmin_{\theta \in \mathbb{R}^d} \sum_{t=1}^{k} \sum_{h=1}^{H} - \log \mathbb{P}_\theta \left( r_t \mid s^t_h, a^t_h \right) + \eta \operatorname{pen}(\theta), \tag{7}$$

$$\implies \quad \sum_{t=1}^{k} \sum_{h=1}^{H} \left( r_t - \mathbb{E}^{\hat{\theta}^{\mathrm{r}}_k}_{s^t_h, a^t_h} \left[r\right] \right) B^\top A_i \varphi\left(s^t_h, a^t_h\right) = \eta \nabla_i \operatorname{pen}\left(\hat{\theta}^{\mathrm{r}}_k\right) \quad \forall i \in [d]. \tag{8}$$

178 **Exploration.** A significant challenge in RL is handling exploration in continuous spaces. The majority
179 of the literature is split between intractable, upper confidence bound-style optimism or Thompson
180 sampling algorithms with high-dimensional posterior and guarantees only in terms of Bayesian
181 regret. In BEF-RLSVI, we adopt the approach of reward perturbation motivated by the RLSVI-
182 framework [ZBB+20, OVRW16]. We show that perturbing the reward estimation can guarantee

optimism with a constant probability, *i.e.* there exists $\nu \in (0, 1]$ such that for all $k \in [K]$ and $s_1^k \in \mathcal{S}$,

$$\mathbb{P}\left(\tilde{V}_1(s_1^k) - V_1^\star(s_1^k) \geq 0\right) \geq \nu.$$

[ZBB+20] proves that this suffices to bound the learning error. However, their method clashes with not clipping the value function, as it modifies the probability of optimism. Thus, [ZBB+20] proposes an involved clipping procedure to handle the issue of unstable values. Instead, by careful geometric analysis (*cf.* Lemma 19), we bound the occurrences of the unstable values, and in turn, upper bound the regret without clipping. Note that unlike [ICN+21], `BEF-RLSVI` does not guarantee that the estimated value function is optimistic but still is able to control the learning error (*cf.* Section 5).

**Planning.** Recall that with our model assumptions, we can write the state-action value function linearly (Equation (5)). Using `BEF-RLSVI`, we have at step $h$:

$$Q_{\hat{\theta}^{\mathrm{p}}, \tilde{\theta}^{\mathrm{r}}, h}^\pi(s, a) = \mathbb{E}_{\tilde{\theta}^{\mathrm{r}}}[r(s, a)] + \left\langle \phi^{\mathrm{p}}(s, a), \int_{\mathcal{S}} \mu^{\mathrm{p}}(\tilde{s}) V_{\hat{\theta}^{\mathrm{p}}, \tilde{\theta}^{\mathrm{r}}, h+1}^\pi(\tilde{s}) d\tilde{s} \right\rangle.$$

Then, we select the best action greedily using dynamic programming to compute $Q_h(s, a)$. Although our model yields infinite dimensional $\phi^{\mathrm{p}}$ and $\psi^{\mathrm{p}}$, approximating them (*cf.* next paragraph) with linear features of dimension $\mathcal{O}(pH^2K \log(HK))$ is possible without increasing the regret. Thus, the planning is done in $\mathcal{O}(pH^3K \log(HK))$, which is pseudo-polynomial in $p$, $H$ and $K$, *i.e.* tractable.

For details about the finite-dimensional approximation of our transition kernel, refer to Appendix D.5. Now, we highlight the schematic of a finite-dimensional approximation of $\phi^{\mathrm{p}}$ and $\psi^{\mathrm{p}}$. We proceed in three steps. **1)** We have with high probability $\mathbb{S}(V_{\hat{\theta}^{\mathrm{p}}, \tilde{\theta}^{\mathrm{r}}, h}) \leq dH^{3/2}$ (Section 5). **2)** If we have a uniform $\epsilon$-approximation of $\mathbb{P}_{\theta^{\mathrm{p}}}$, we show that using it incurs at most an extra $\mathcal{O}(\epsilon dH^{5/2}K)$ regret. **3)** Finally, following [RR07], we approximate uniformly the shift invariant kernels, here the RBF in Equation (4), within $\epsilon$ error and with features of dimensions $\mathcal{O}(p\epsilon^{-2} \log \frac{1}{\epsilon^2})$, where $p$ is dimension of $\psi$. Associating these three elements and choosing $\epsilon = 1/\sqrt{(H^2K)}$, we establish our claim.

# 5 Theoretical analysis: proof outline

To convey the novelties in our analysis, we provide a proof sketch for Theorem 2. We start by decomposing the regret into an estimation loss and a learning error, as given below

$$R(K) = \sum_{k=1}^K (V_{\theta^{\mathrm{p}}, \theta^{\mathrm{r}}, 1}^\star - V_{\theta^{\mathrm{p}}, \theta^{\mathrm{r}}, 1}^{\pi_k})(s_{1k}) = \sum_{k=1}^K (\underbrace{V_{\theta^{\mathrm{p}}, \theta^{\mathrm{r}}, 1}^\star - V_{\hat{\theta}^{\mathrm{p}}, \tilde{\theta}^{\mathrm{r}}, 1}^{\pi_k}}_{learning} + \underbrace{V_{\hat{\theta}^{\mathrm{p}}, \tilde{\theta}^{\mathrm{r}}, 1}^{\pi_k} - V_{\theta^{\mathrm{p}}, \theta^{\mathrm{r}}, 1}^{\pi_k}}_{Estimation})(s_{1k}). \quad (9)$$

For the **estimation error**, we use smoothness arguments with concentrations of parameters up to some novelties. Regarding the **learning error**, we show that the injected noise ensures a constant probability of anti-concentration. Applying Assumption 1 and Lemma 18 leads to the upper-bound.

## 5.1 Bounding the estimation error

We further decompose the estimation error into the errors in estimating transitions and rewards.

$$V_{\hat{\theta}^{\mathrm{p}}, \tilde{\theta}^{\mathrm{r}}}^\pi(s_{1k}) - V_{\theta^{\mathrm{p}}, \theta^{\mathrm{r}}}^\pi(s_{1k}) = \underbrace{V_{\hat{\theta}^{\mathrm{p}}, \theta^{\mathrm{r}}}^\pi(s_{1k}) - V_{\theta^{\mathrm{p}}, \theta^{\mathrm{r}}}^\pi(s_{1k})}_{\text{transition estimation}} + \underbrace{V_{\hat{\theta}^{\mathrm{p}}, \tilde{\theta}^{\mathrm{r}}}^\pi(s_{1k}) - V_{\hat{\theta}^{\mathrm{p}}, \theta^{\mathrm{r}}}^\pi(s_{1k})}_{\text{reward estimation}} \quad (10)$$

**Transition estimation** Since the reward parameter is exact, the value function's span is $\leq H$. Then, using the transportation of Lemma 13 we obtain the bound $H \sum_{h=1}^H \sqrt{2 \mathrm{KL}_{s_{hk}, a_{hk}}(\theta^{\mathrm{p}}, \hat{\theta}^{\mathrm{p}})}$. We notice that since the reward parameter is exact, the bound is actually $H \min\{1, \sum_{h=1}^H \sqrt{2 \mathrm{KL}_{s_{hk}, a_{hk}}(\theta^{\mathrm{p}}, \hat{\theta}^{\mathrm{p}})}\}$. Using Lemma 18 under Assumption 1, we win a $\sqrt{H}$ factor compared to the analysis of [CG19].

**Reward estimation** Previous work uses clipping to help control this error, but in this case it can hinder the optimism probability by biasing the noise. [ZBB+20] proposes an involved clipping depending on the norms $\|(A_i \varphi(s_h^k, a_h^k))_{i \in [d]}\|_{(\bar{G}_k^{\mathrm{p}})^{-1}}$, which is somewhat delicate to analyze and

deploy. We remedy the situation acting solely in the proof. First let's define what we call the set of "bad rounds": $\left\{k \in [K], \exists h : \|(A_i\varphi(s_h^k, a_h^k))_{i\in[d]}\|_{(\bar{G}_k^{\mathrm{p}})^{-1}} \geq 1\right\}$, these rounds are why clipping is necessary. Thanks to Lemma 19, we know that the number of such rounds is at most $\mathcal{O}(d)$. Surprisingly, it depends neither on $H$ nor on $K$. We show that the "bad rounds" incur at most $O(d^{3/2}H^2)$ regret, independent of $K$. Therefore, our algorithm can forgo clipping for free.

**Remark.** *If it wasn't for the episodic nature of our setting, we could have used the forward algorithm to eliminate the span control issue. We refer to [Vov01, AW01] for a description of this algorithm, [OMP21] for a stochastic analysis, and Section 4 therein for an application to linear bandits.*

## 5.2 Bounding the learning error

To upper-bound this term of the regret, we first show that the estimated value function is optimistic with a constant probability. Then, we show that this is enough to control the learning error.

**Stochastic optimism.** The perturbation ensures a constant probability of optimism. Specifically,

$$
\begin{aligned}
(V_{\hat{\theta}^{\mathrm{p}},\tilde{\theta}^{\mathrm{r}},1} - V_{\theta^{\mathrm{p}},\theta^{\mathrm{r}},1}^{\star})(s_1) &\geq (Q_{\hat{\theta}^{\mathrm{p}},\tilde{\theta}^{\mathrm{r}},1}^{\star} - Q_1^{\star})(s_1, \pi^{\star}(s_1)) \\
&\geq \underbrace{V_{\hat{\theta}^{\mathrm{p}},\theta^{\mathrm{r}}}^{\pi^{\star}}(s_1) - V_{\theta^{\mathrm{p}},\theta^{\mathrm{r}}}^{\pi^{\star}}(s_1)}_{\text{first term}} + \underbrace{V_{\hat{\theta}^{\mathrm{p}},\hat{\theta}^{\mathrm{r}}}^{\pi^{\star}}(s_1) - V_{\hat{\theta}^{\mathrm{p}},\theta^{\mathrm{r}}}^{\pi^{\star}}(s_1)}_{\text{second term}} + \underbrace{V_{\hat{\theta}^{\mathrm{p}},\tilde{\theta}^{\mathrm{r}}}^{\pi^{\star}}(s_1) - V_{\hat{\theta}^{\mathrm{p}},\hat{\theta}^{\mathrm{r}}}^{\pi^{\star}}(s_1)}_{\text{third term}}
\end{aligned}
$$

The first and second terms are perturbation free, we handle them similarly to the estimation error, *i.e.* using concentration arguments for $\hat{\theta}^{\mathrm{p}}$ and $\hat{\theta}^{\mathrm{r}}$. For the third term, we use transportation of rewards (Lemma 17) and anti-concentration of $\xi_k$ (Lemma 12). We find that with probability at least $1 - 2\delta$

$$
\begin{aligned}
(V_{\hat{\theta}^{\mathrm{p}},\tilde{\theta}^{\mathrm{r}},1} - V_{\theta^{\mathrm{p}},\theta^{\mathrm{r}},1}^{\star})(s_1) \geq & \xi_k^\top \, \mathbb{E}_{(\tilde{s}_t)_{t\in[H]}\sim\hat{\theta}^{\mathrm{p}}|s_1^k}\left[\sum_{t=1}^H \frac{\mathbb{V}\mathrm{ar}^{\theta_j^{\mathrm{r}}}(r)}{2}(A_i\varphi(\tilde{s}_t, \pi^{\star}(\tilde{s}_t)))_{i\in[d]}\right] B \\
& - Hc(n,\delta)\left\|\sum_{h=1}^H \mathbb{E}_{(\tilde{s}_t)_{t\in[H]}\sim\hat{\theta}^{\mathrm{p}}|s_1^k}\left[(A_i\varphi(\tilde{s}_h, \pi^{\star}(\tilde{s}_h)))_{i\in[d]}\right]\right\|_{(\bar{G}_k^{\mathrm{p}})^{-1}},
\end{aligned}
$$

where $c(n,\delta) = \left(\sqrt{\beta^{\mathrm{p}}\beta^{\mathrm{p}}(n,\delta)/\alpha^{\mathrm{p}}} + \sqrt{\beta^{\mathrm{r}}\beta^{\mathrm{r}}(n,\delta)}\min\{1, \alpha^{\mathrm{p}}/\alpha^{\mathrm{r}}\}/(2\alpha^{\mathrm{r}})\right)$. Since $\xi_k \sim \mathcal{N}(0, x_k(\bar{G}_k^{\mathrm{p}})^{-1})$ and $x_k \geq H^2 c(n,\delta)^2$, we get $\mathbb{P}\left(V_{\hat{\theta}^{\mathrm{p}},\tilde{\theta}^{\mathrm{r}},1}^{\pi}(s_1) - V_{\theta^{\mathrm{p}},\theta^{\mathrm{r}},1}^{\star}(s_1) \geq 0\right) \geq \Phi(-1)$, where $\Phi$ is the normal CDF. This is ensured by the anti-concentration property of Gaussian random variables, see Lemma 12.

**From stochastic optimism to error control:** Existing algorithms require the value function to be optimistic (*i.e.* negative learning error) with large probability. Contrary to them, `BEF-RLSVI` only requires the estimated value to be optimistic with a constant probability. When it is, the learning happens. Otherwise, the policy is still close to a good one thanks to the decreasing estimation error, and the learning still happens. This part of the proof is similar in spirit to that of [ZBB$^{+}$20].

*Upper bound on $V_1^\star$:* Draw $(\bar{\xi}_k)_{k\in[K]}$ i.i.d copies of $(\xi_k)_{k\in[K]}$ and define the event where optimism holds as $\bar{O}_k \triangleq \{V_{\hat{\theta}^{\mathrm{p}},\tilde{\theta}_k^{\mathrm{r}},1}(s_1^k) - V_1^\star(s_1^k) \geq 0\}$. This implies that $\quad V_1^\star(s_1^k) \leq \mathbb{E}_{\bar{\xi}_k|\bar{O}_k}[V_{\hat{\theta}^{\mathrm{p}},\hat{\theta}^{\mathrm{r}}+\bar{\xi}_k,1}(s_1^k)]$.

*Lower bound on $V_{\hat{\theta}^{\mathrm{p}},\tilde{\theta}^{\mathrm{r}}}$ :* Consider $\underline{V}_1(s_1^k)$ to be a solution of the optimization problem

$$
\min_{\xi_k} V_{\hat{\theta}^{\mathrm{p}},\hat{\theta}^{\mathrm{r}}+\xi_k,1}(s_1^k) \quad \text{subject to: } \|\xi_k\|_{\bar{G}_k} \leq \sqrt{x_k d \log(d/\delta)},
$$

As the injected noise concentrates, we obtain $\underline{V}_1(s_1^k) \leq V_{\hat{\theta}^{\mathrm{p}},\tilde{\theta}^{\mathrm{r}}}(s_1^k)$.

*Combination:* Using these upper and lower bounds, we show that with probability at least $1 - \delta$,

$$
\begin{aligned}
V_1^\star(s_1^k) - V_{\hat{\theta}^{\mathrm{p}},\hat{\theta}^{\mathrm{r}}+\bar{\xi}_k,1}(s_1^k) &\leq \mathbb{E}_{\bar{\xi}_k|\bar{O}_k}[V_{\hat{\theta}^{\mathrm{p}},\hat{\theta}^{\mathrm{r}}+\bar{\xi}_k,1}(s_1^k) - \underline{V}_1(s_1^k)] \\
&\leq \left(\mathbb{E}_{\bar{\xi}_k}[V_{\hat{\theta}^{\mathrm{p}},\hat{\theta}^{\mathrm{r}}+\bar{\xi}_k,1}(s_1^k) - \underline{V}_1(s_1^k)] - \mathbb{E}_{\bar{\xi}_k|\bar{O}_k^{\mathrm{c}}}[V_{\hat{\theta}^{\mathrm{p}},\hat{\theta}^{\mathrm{r}}+\bar{\xi}_k,1}(s_1^k) - \underline{V}_1(s_1^k)]\mathbb{P}(\bar{O}_k^{\mathrm{c}})\right)/\mathbb{P}(\bar{O}_k),
\end{aligned}
$$

The last step follows from the tower rule. Note that the term inside the expectations is positive with high probability but not necessarily in expectation. We follow the lines of the estimation error analysis to complete the proof of Theorem 2. We refer to Appendix B.2 for the detailed proof.

## 6 Related works: functional representations with regret and tractability

Our work extends the endeavor of using functional representations to perform optimal regret minimization in continuous state-action MDPs. We now provide a few complementary details.

*General functional representation.* [DSL+18] provides the first convergence guarantee for general nonlinear function representations in the Maximum Entropy RL setting, where entropy of a policy is used as a regularizer to induce exploration. Thus, the analysis cannot address episodic RL, where we have to explicitly ensure exploration with optimism. [WSY20] proposes a framework that leverages the optimism with confidence bound approach for general functional representations with bounded Eluder dimensions, which is a complexity measure in RL. However, knowing the Eluder dimension is crucial for the optimistic confidence bound in their algorithm. Eluder dimension is not known for MDPs except linear and tabular MDPs. *To concretize our design, we focus on the general but explicit bilinear exponential family of MDPs than any abstract representation.*

*Bilinear exponential family of MDPs.* Exponential families are studied widely in RL theory, from bandits to MDPs [LMT21, KKM13, FCGS10, KH06], as an expressive parametric family to design theoretically-grounded model-based algorithms. [CGM21] first studies episodic RL with Bilinear Exponential Family (BEF) of transitions, which is linear in both state-action pairs and the next-state. It proposes a regularized log-likelihood method to estimate the model parameters, and two optimistic algorithms with upper confidence bounds and posterior sampling. Due to its generality to unifiedly model tabular MDPs, factored MDPs, linear MDPs, and linearly controlled dynamical systems, the BEF-family of MDPs has received increasing attention [LLS+21]. [LLS+21] estimates the model parameters based on score matching that enables them to replace regularity assumption on the log-partition function with Fisher-information and assumption on the parameters. Both [CGM21, LLS+21] achieve a worst-case regret of order $\tilde{O}(\sqrt{d^2 H^4 K})$ for known reward. On a different note, [DKL+21, FKQR21] also introduces a new structural framework for generalization in RL, called bilinear classes as it requires the Bellman error to be upper bounded by a bilinear form. Instead of using bilinear forms to capture non-linear structures, this class is not identical to BEF class of MDPs, and studying the connection is out of the scope of this paper. Specifically, *we address the shortcomings of the existing works on BEF-family of MDPs that assume known rewards, absence of RLSVI-type algorithms, and access to oracle planners.*

*Tractable planning and linearity.* Planning is a major byproduct of the chosen functional representation. In general, planning can incur high computational complexity if done naïvely. Specially, [DKWY19] shows that for some settings, even with a linear $\epsilon$-approximation of the $Q$-function, a planning procedure able to produce an $\epsilon$-optimal policy has a complexity at least $2^H$. Thus, different works [SS20, LSW20, VRD19] propose to leverage different low-dimensional representations of value functions or transitions to perform efficient planning. Here, we take note from [RZSD21] that Gaussian transitions induce an explicit linear value function in an RKHS. And generalize this observation with the bilinear exponential. Moreover, using uniformly good features [RR07] to approximate transition dynamics from our model enables us to design a tractable planner. We provide a detailed discussion of this approximation in Section 4. More practically, [RZSD21, NY21] use representations given by random Fourier features [RR07] to approximate the transition dynamics and provide experiments validating the benefits of this approach for high-dimensional Atari-games.

## 7 Conclusion and future work

We propose the `BEF-RLSVI` algorithm for the bilinear exponential family of MDPs in the setting of episodic-RL. `BEF-RLSVI` explores using a Gaussian perturbation of rewards, and plans tractably (complexity of $\mathcal{O}(pH^3 K \log(HK))$) thanks to properties of the RBF kernel. Our proof shows that clipping can be forwent for similar `RLSVI`-type algorithms. Moreover, we prove a $\sqrt{d^3 H^3 K}$ frequentist regret bound, which improves over existing work, accommodates unknown rewards, and matches the lower bound in terms of $H$ and $K$. Regarding future work, we believe that our proof approach can be extended to rewards with bounded variance. We also believe that the extra $\sqrt{d}$ in our bound is an artefact of the proof, and specifically, the anti-concentration. We will investigate it further. Finally, we plan to study the practical efficiency of `BEF-RLSVI` through experiments on tasks with continuous state-action spaces in an extended version of this work.

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
