}^\mathbf{p}, \tilde{\theta}^\mathbf{r}$, initialize $\tilde{\theta} = (\tilde{\theta}^\mathbf{r}, \hat{\theta}^\mathbf{p})$ and $\forall s, V_{H+1}(s) = 0$
2: **for** steps $h = H - 1, H - 2, \cdots, 0$ **do**
3:     Calculate $Q_{\tilde{\theta},h}(s, a) = \mathbb{E}_{s,a}^{\tilde{\theta}^\mathbf{r}}[r] + \langle \phi^\mathbf{p}(s, a), \int V_{\tilde{\theta},h+1}(s')\mu^\mathbf{p}(s')ds'\rangle_{\mathcal{H}}.$
4: **end for**

---

We can approximate Line 3 of Algorithm 2 with $\mathcal{O}(pH^3K\log(HK))$ complexity and without harming the learning process (*cf.* § planning, Section 4). Therefore, here, planning is tractable.

### 3.2 BEF-RLSVI: regret upper-bound

We state the standard smoothness assumptions on the model [CGM21, JBNW17, LMT21].

**Assumption 1.** *There exist constants $\alpha^\mathbf{p}, \alpha^\mathbf{r}, \beta^\mathbf{p}, \beta^\mathbf{r} > 0$, such that the representation model satisfies:*

$$\forall (s, a) \in \mathcal{S} \times \mathcal{A}, \forall \theta, x \in \mathbb{R}^d \quad \alpha^\mathbf{p} \leq x^\top C_{s,a}^\theta[\psi] x \leq \beta^\mathbf{p}$$

$$\forall (s, a) \in \mathcal{S} \times \mathcal{A}, \forall \theta, x \in \mathbb{R}^d \quad \alpha^\mathbf{r} \leq \mathbb{V}\text{ar}_{s,a}^\theta(r) \, x^\top B^\top B x \leq \beta^\mathbf{r}$$

*where* $\mathbb{C}_{s,a}^\theta[\psi(s')] \triangleq \mathbb{E}_{s'\sim\mathbb{P}_\theta|s,a}\left[\psi(s')\psi(s')^\top\right] - \mathbb{E}_{s'\sim\mathbb{P}_\theta|s,a}\left[\psi(s')\right]\mathbb{E}_{s'\sim\mathbb{P}_\theta|s,a}\left[\psi(s')^\top\right]$ *and*

$\mathbb{V}\text{ar}_{s,a}^\theta(r) \triangleq \left(\mathbb{E}_{s,a}^\theta\left[r^2\right] - \mathbb{E}_{s,a}^\theta\left[r\right]^2\right)$ *is the variance of the reward under $\theta$.*

A closer look at the derivatives of the model (see Appendix D.3) tells us that previous inequalities directly imply a control over the eigenvalues of the Hessian matrices of the log-normalizers.

We now state our main result, the regret upper-bound of BEF-RLSVI.

**Theorem 2** (Regret bound). *Let $\mathbb{A} \triangleq (\text{tr}(A_i A_j^\top))_{i,j\in[d]}$ and $G_{s,a} \triangleq (\varphi(s,a)^\top A_i^\top A_j \varphi(s,a))_{i,j\in[d]}$. Under Assumption 1 and further considering that*

*1. $\max\{\|\theta^\mathbf{r}\|_\mathbb{A}, \|\theta^\mathbf{p}\|_\mathbb{A}\} \leq B_\mathbb{A}$, $\|\mathbb{A}^{-1}G_{s,a}\| \leq B_{\varphi,\mathbb{A}}$ and $\mathbb{E}_{\theta^\mathbf{r}}[r(s,a)] \in [0,1]$ for all $(s,a)$.*

*2. noise $\xi_k \sim \mathcal{N}(0, x_k(\bar{G}_k^\mathbf{p})^{-1})$ satisfies $x_k \geq \left(H\sqrt{\frac{\beta^\mathbf{p}\beta^\mathbf{p}(K,\delta)}{\alpha^\mathbf{p}\alpha^\mathbf{r}}} + \frac{\sqrt{\beta^\mathbf{r}\beta^\mathbf{r}(K,\delta)\min\{1,\frac{\alpha^\mathbf{p}}{\alpha^\mathbf{r}}\}}}{2\alpha^\mathbf{r}}\right)^2 \propto dH^2,$*

*then for all $\delta \in (0,1]$, with probability at least $1 - 7\delta$,*

$$\mathcal{R}(K) \leq \sqrt{KH}\left[\underbrace{2H\left(\sqrt{\frac{2\beta^\mathbf{p}}{\alpha^\mathbf{p}}\beta^\mathbf{p}(K,\delta)\gamma_K^\mathbf{p}} + (1+\sqrt{\gamma_K^\mathbf{r}})\sqrt{\log(1/\delta^2)}\right)}_{\text{Transition concentration} \approx dH} + \underbrace{\beta^\mathbf{r}\sqrt{\frac{\beta^\mathbf{r}(n,\delta)\gamma_K^\mathbf{r}}{2\alpha^\mathbf{r}}}}_{\text{Reward concentration} \approx d}\right.$$

$$+ \underbrace{c\beta^\mathbf{r}\sqrt{x_K d\gamma_K^\mathbf{r}\log(dK/\delta)} + \frac{\beta^\mathbf{r}\sqrt{x_K d\gamma_K^\mathbf{r}\log(e/\delta^2)}}{\Phi(-1)}(1+\sqrt{\log(d/\delta)})}_{\text{Noise concentration} \approx d^{3/2}H}\Bigg]$$

$$+ \sqrt{H\gamma_K^\mathbf{r}}\left[\underbrace{\beta^\mathbf{r}C_d\left(\sqrt{\frac{\beta^\mathbf{r}(K,\delta)}{2\alpha^\mathbf{r}}} + c\sqrt{x_K d\log(dK/\delta)}\right)}_{\text{Estimation error for no clipping} \approx dH}\right.$$

$$+ \underbrace{\frac{\beta^\mathbf{r}d\sqrt{x_K}}{\Phi(-1)}(1+\sqrt{\log(d/\delta)})\sqrt{C_d\left(1+\frac{\alpha^\mathbf{r}B_{\varphi,A}H}{\eta}\right)}}_{\text{Learning error for no clipping} \approx (dH)^{3/2}}\Bigg],$$

*where for $i \in [\mathbf{p}, \mathbf{r}]$, $\beta^i(K,\delta) \triangleq \frac{\eta}{2}B_\mathbb{A}^2 + \gamma_K^i + \log(1/\delta)$, and $\gamma_K^i \triangleq d\log(1 + \frac{\beta^i}{\eta}B_{\varphi,\mathbb{A}}HK)$. Also,*

$C_d \triangleq \frac{3d}{\log(2)}\log\left(1 + \frac{\alpha^\mathbf{r}\|\mathbb{A}\|_2^2 B_{\varphi,\mathbb{A}}^2}{\eta\log(2)}\right)$, *$\Phi$ is the Gaussian CDF, and $c$ is a universal constant.*

Theorem 2 entails a regret $\mathcal{R}(K) = \mathcal{O}(\sqrt{d^3 H^3 K})$ for BEF-RLSVI, where $d$ is the number of parameters of the bilinear exponential family model, $K$ is the number of episodes, and $H$ is the horizon of an episode. We now clarify how this contrasts with related literature.

145 *Comparison with other bounds.* The closest work to ours is [CGM21] as it considers the same
146 model for transitions but with known rewards. They propose a UCRL-type and PSRL-type algorithm,
147 which achieve a regret of order $\widetilde{O}(\sqrt{d^2 H^4 K})$. There are two notable algorithmic differences with
148 our work. First, they do exploration using intractable-optimistic upper bounds or high-dimensional
149 posteriors, while we do it with explicit perturbation. The second difference is in planning. While
150 they assume access to a planning oracle, we do it explicitly with pseudo-polynomial complexity
151 (Section 4). Moreover, we improve the regret bound by a $\sqrt{H}$ factor thanks to an improved analysis,
152 (*cf.* Lemma 18). But similar to all RLSVI-type algorithms, we pick up an extra $\sqrt{d}$ (*cf.* [AL17]).

153 [ZBB+20] proposes a variant of RLSVI for continuous state-action spaces, where there are low-rank
154 models of transitions and rewards. They show a regret bound $R(K) = \widetilde{O}(\sqrt{d^4 H^5 K})$, which is larger
155 than that of BEF-RLSVI by $O(\sqrt{dH^2})$. In algorithm design, we improve on their work by removing
156 the need to carefully clip the value function. Analytically, our model allows us to use transportation
157 inequalities (*cf.* Lemma 13) instead of the simulation lemma, which saves us a $\sqrt{H}$ factor.

158 [RZSD21] considers Gaussian transitions, i.e. $s' = f^*(s, a) + \epsilon$ such that $\epsilon \sim \mathcal{N}\left(0, \sigma^2\right)$. This is a
159 particular case of our model. They propose to use Thompson Sampling, and have the merit of being
160 the first to have observed linearity of the value function from this transition structure. But they do not
161 connect it to the finite dimensional approximation of [RR07] unlike us (Section 4). Finally, they show
162 a Bayesian regret bound of $O(\sqrt{d^2 H^3 K})$. This notion of regret is weaker than frequentist regret,
163 hence this result is not directly comparable with Theorem 2.

164 *Tightness of regret bound.* A lower bound for episodic RL with continuous state-action spaces is
165 still missing. However, for tabular RL, [DMKV21] proves a lower bound of order $\Omega(\sqrt{H^3 SAK})$.
166 If we represent a tabular MDP in our model, we would need $d = S^2 \times A$ parameters (Section 4.3,
167 [CGM21]). In this case, our bound becomes $R(K) = O(\sqrt{(S^2 A)^3 H^3 K})$, which is clearly not tight
168 is $S$ and $A$. This is understandable due to the relative generality of our setting. We are however
169 positively surprised that **our bound is tight in terms of its dependence on $H$ and $K$.**

## 4  Algorithm design: building blocks of BEF-RLSVI

171 We present necessary details about BEF-RLSVI and discuss the key algorithm design techniques.

172 **Estimation of parameters.** We estimate transitions and rewards from observations similar to
173 EXP-UCRL [CGM21], *i.e.* by using a penalized maximum likelihood estimator

$$\hat{\theta}^{\mathtt{p}}(k) \in \underset{\theta \in \mathbb{R}^d}{\arg\min} \sum_{t=1}^{k} \sum_{h=1}^{H} - \log \mathbb{P}_\theta \left(s_{h+1}^t \mid s_h^t, a_h^t\right) + \eta \operatorname{pen}(\theta).$$

174 Here, $\operatorname{pen}(\theta)$ is a trace-norm penalty: $\operatorname{pen}(\theta) = \frac{1}{2} \|\theta\|_{\mathbb{A}}$ and $\mathbb{A} = (\operatorname{tr}(A_i A_j^\top))_{i,j}$. By properties of
175 the exponential family, the penalized maximum likelihood estimator verifies, for all $i \leq d$:

$$\sum_{t=1}^{k} \sum_{h=1}^{H} \left( \psi\left(s_{h+1}^t\right) - \mathbb{E}_{s_h^t, a_h^t}^{\hat{\theta}_k^{\mathtt{p}}} \

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

# Appendix

## Table of Contents

## A  Notations

We dedicate this section to index all the notations used in this paper. Note that every notation is defined when it is introduced as well.

Table 2: Notations

| | | |
|---|---|---|
| $H$ | $\overset{\text{def}}{=}$ | number of steps in a given episode |
| $K$ | $\overset{\text{def}}{=}$ | number of episodes |
| $T$ | $\overset{\text{def}}{=}$ | $KH$, total number of steps |
| $s_h^k$ | $\overset{\text{def}}{=}$ | state at time $h$ of episode $k$, denoted $s_h$ when $k$ is clear from context |
| $a_h^k$ | $\overset{\text{def}}{=}$ | action at time $h$ of episode $k$, denoted $a_h$ when $k$ is clear from context |
| $r(s,a)$ | $\overset{\text{def}}{=}$ | realization of the reward in state $s$ under action $a$ |
| $\theta^{\text{p}}$ | $\overset{\text{def}}{=}$ | parameter of the transition distribution, $\in \mathbb{R}^d$ |
| $\theta^{\text{r}}$ | $\overset{\text{def}}{=}$ | parameter of the reward distribution, $\in \mathbb{R}^d$ |
| $\theta$ | $\overset{\text{def}}{=}$ | $\in \mathbb{R}^d$ denotes either $\theta^{\text{r}}$ or $\theta^{\text{p}}$, unless stated otherwise |
| $\hat{\theta}$ | $\overset{\text{def}}{=}$ | $\theta$ estimator with Maximum Likelihood unless stated otherwise |
| $\tilde{\theta}$ | $\overset{\text{def}}{=}$ | $\hat{\theta} + \xi$ where $\xi$ is a chosen noise. Perturbed estimation of $\theta$. |
| $[\theta_1, \theta_2]$ | $\overset{\text{def}}{=}$ | the d-dimensional $\ell_\infty$ hypercube joining $\theta_1$ and $\theta_2$ |
| $\mathbb{P}_{\theta^{\text{p}}}$ | $\overset{\text{def}}{=}$ | transition under the exponential family model with parameter $\theta^{\text{p}}$ |
| $\psi$ | $\overset{\text{def}}{=}$ | feature function, $\in (\mathbb{R}_+^p)^{\mathcal{S}}$ |
| $\varphi$ | $\overset{\text{def}}{=}$ | feature function, $\in (\mathbb{R}_+^q)^{\mathcal{S} \times \mathcal{A}}$ |
| $B$ | $\overset{\text{def}}{=}$ | $p$-dimensional vector |
| $M_\theta$ | $\overset{\text{def}}{=}$ | $\sum_{i=1}^d \theta_i A_i$, where $A_i$ are $p \times q$ matrices. |
| $Z^{\text{r}}$ | $\overset{\text{def}}{=}$ | the rewards' log partition function |
| $Z^{\text{p}}$ | $\overset{\text{def}}{=}$ | the transitions' log partition function |
| $\mathcal{H}$ | $\overset{\text{def}}{=}$ | Hilbert space where we decompose transitions |
| $\mu^{\text{p}}$ | $\overset{\text{def}}{=}$ | feature function after decomposition, $\in (\mathbb{R}_+)^{\mathcal{S} \times \mathcal{H}}$ |
| $\phi^{\text{p}}$ | $\overset{\text{def}}{=}$ | feature function after decomposition, $\in (\mathbb{R}_+)^{\mathcal{S} \times \mathcal{A} \times \mathcal{H}}$ |
| $G_{s,a}$ | $\overset{\text{def}}{=}$ | $\left( \varphi(s,a)^\top A_i^\top A_j \varphi(s,a) \right)_{i,j \in [d]}$ |
| $\bar{G}_k^{\text{r}}$ | $\overset{\text{def}}{=}$ | $\bar{G}_{(k-1)h}^{\text{r}} = \frac{\eta}{\alpha^{\text{r}}} \mathbb{A} + \sum_{\tau=1}^{k-1} \sum_{h=1}^{H} G_{s_h^\tau, a_h^\tau}$ |
| $\bar{G}_k^{\text{p}}$ | $\overset{\text{def}}{=}$ | $\bar{G}_{(k-1)h}^{\text{p}} = \frac{\eta}{\alpha^{\text{p}}} \mathbb{A} + \sum_{\tau=1}^{k-1} \sum_{h=1}^{H} G_{s_h^\tau, a_h^\tau}$ |
| $\mathbb{C}_{s,a}^\theta [\psi(s')]$ | $\overset{\text{def}}{=}$ | $\mathbb{E}_{s,a}^\theta \left[ \psi(s') \psi(s')^\top \right] - \mathbb{E}_{s,a}^\theta [\psi(s')] \mathbb{E}_{s,a}^\theta \left[ \psi(s')^\top \right]$ |
| $\beta^{\text{p}}$ | $\overset{\text{def}}{=}$ | $\sup_{\theta,s,a} \lambda_{\max} \left( \mathbb{C}_{s,a}^\theta [\psi(s')] \right)$ linked to the maximum eigenvalue of $\nabla^2 Z^{\text{p}}$ |
| $\alpha^{\text{p}}$ | $\overset{\text{def}}{=}$ | $\inf_{\theta,s,a} \lambda_{\max} \left( \mathbb{C}_{s,a}^\theta [\psi(s')] \right)$ linked to the minimum eigenvalue of $\nabla^2 Z^{\text{p}}$ |
| $\beta^{\text{r}}$ | $\overset{\text{def}}{=}$ | $\lambda_{\max} \left( BB^\top \right) \sup_{\theta,s,a} \mathbb{V}\text{ar}_{s,a}^\theta(r)$, linked to the maximum eigenvalue of $\nabla^2 Z^{\text{r}}$ |
| $\alpha^{\text{r}}$ | $\overset{\text{def}}{=}$ | $\lambda_{\min} \left( BB^\top \right) \inf_{\theta,s,a} \mathbb{V}\text{ar}_{s,a}^\theta(r)$, linked to the minimum eigenvalue of $\nabla^2 Z^{\text{r}}$ |

# B  Regret analysis

We provide a high probability analysis of the regret of `BEF-RLSVI` under standard regularity assumptions of the representation. First we recall the regret definition then we separate the perturbation error from the statistical estimation:

$$\mathcal{R}(K) = \sum_{k=1}^{K}(V_{\theta^p,\theta^r,1}^{\star} - V_{\theta^p,\theta^r,1}^{\pi_k})(s_1^k) = \sum_{k=1}^{K}\left(\underbrace{V_{\theta^p,\theta^r,1}^{\star} - V_{\hat{\theta}^p,\tilde{\theta}^r,1}^{\pi_k}}_{learning} + \underbrace{V_{\hat{\theta}^p,\tilde{\theta}^r,1}^{\pi_k} - V_{\theta^p,\theta^r,1}^{\pi_k}}_{Estimation}\right)(s_1^k)$$

## B.1  Estimation error

To show that the estimation error $\left(\sum_{k=1}^{K}V_{\hat{\theta}^p,\tilde{\theta}^r,1}^{\pi_k} - V_{\theta^p,\theta^r,1}^{\pi_k}\right)$ can be controlled, we decompose it to an error that comes from the estimation of the transition parameter and one that comes from the estimation of the reward parameter:

$$V_{\hat{\theta}^p,\tilde{\theta}^r}^{\pi}(s_1^k) - V_{\theta^p,\theta^r}^{\pi}(s_1^k) = \underbrace{V_{\hat{\theta}^p,\theta^r}^{\pi}(s_1^k) - V_{\theta^p,\theta^r}^{\pi}(s_1^k)}_{\text{transition estimation}} + \underbrace{V_{\hat{\theta}^p,\tilde{\theta}^r}^{\pi}(s_1^k) - V_{\hat{\theta}^p,\theta^r}^{\pi}(s_1^k)}_{\text{reward estimation}},$$

we control each term separately in Section B.1.1 and Section B.1.2. Therefore, we obtain the following lemma controlling the estimation error.

**Lemma 3.** *The estimation error satisfies, with probability at least $1 - 5\delta$*

$$\sum_{k=1}^{K}V_{\hat{\theta}^p,\tilde{\theta}^r,1}^{\pi}(s_1^k) - V_{\theta^p,\theta^r,1}^{\pi}(s_1^k) \le 2H\sqrt{\frac{2\beta^p}{\alpha^p}\beta^p(N,\delta)N\gamma_K^p} + 2H\sqrt{2N\log(1/\delta)}$$

$$+ \left[\sqrt{KHd\log\left(1 + \alpha^r\eta^{-1}B_{\varphi,\mathbb{A}}n\right)} + C_d\sqrt{Hd\log(1 + \alpha\eta^{-1}B_{\varphi,\mathbb{A}}H)}\right] \times \left(\sqrt{\frac{\beta^r(n,\delta)}{2\alpha^r}}\right.$$

$$\left. + c\sqrt{(\max_k x_k)d\log(dK/\delta)}\right)\beta^r + \sqrt{2KHd\log\left(1 + \alpha^r\eta^{-1}B_{\varphi,\mathbb{A}}n\right)\log(1/\delta)}$$

*where for $i \in [p, r]$, $\beta^i(K,\delta) \triangleq \frac{\eta}{2}B_{\mathbb{A}}^2 + \gamma_K^i + \log(1/\delta)$, and $\gamma_K^i \triangleq d\log(1 + \frac{\beta^i}{\eta}B_{\varphi,\mathbb{A}}HK)$. Also, $C_d \triangleq \frac{3d}{\log(2)}\log\left(1 + \frac{\alpha^r\|\mathbb{A}\|_2^2 B_{\varphi,\mathbb{A}}^2}{\eta\log(2)}\right)$, and c is a universal constant.*

*Proof.* It follows directly by combining Lemma 4 and Lemma 5 using a union bound. $\qquad\square$

### B.1.1  Transition estimation

The goal of this section is to prove the following lemma which bounds the regret due to transition estimation.

**Lemma 4.** *We have, with probability at least $1 - 2\delta$*

$$\sum_{k=1}^{K}V_{\hat{\theta}^p,\theta^r}(s_1^k) - V_{\theta^p,\theta^r}^{\pi}(s_1^k) \le 2H\sqrt{\frac{2\beta^p}{\alpha^p}\beta^p(N,\delta)N\gamma_K^p} + 2H\sqrt{2N\log(1/\delta)}$$

*where $\gamma_K^p := d\log\left(1 + \beta^p\eta^{-1}B_{\varphi,\mathbb{A}}HK\right)$, and $\beta^p(K,\delta) \triangleq \frac{\eta}{2}B_{\mathbb{A}}^2 + \gamma_K^p + \log(1/\delta)$.*

*Proof.* The proof proceeds in two parts. First, we will reveal a bound in terms of the induced local geometry, *i.e.* a bound in terms of KL-divergence. Second, we explicit the bound by transferring the induced local geometry to the euclidean one.

*1) Bound in terms of local geometry.*  We provide a bound on the estimation error of the transition in terms of KL divergences, for that end we show that the estimation error can be decomposed and well controlled. We start by writing the one-step decomposition:

$$V^{\pi}_{\hat{\theta}^{\mathtt{P}},\theta^{\mathtt{r}},1}(s_1^k) - V^{\pi}_{\theta^{\mathtt{P}},\theta^{\mathtt{r}},1}(s_1^k)$$

$$= \mathbb{E}^{\hat{\theta}^{\mathtt{P}}}_{s_1^k,a_1^k}\left[V^{\pi}_{\hat{\theta}^{\mathtt{P}},\theta^{\mathtt{r}},2}\right] - \mathbb{E}^{\theta^{\mathtt{P}}}_{s_1^k,a_1^k}\left[V^{\pi}_{\hat{\theta}^{\mathtt{P}},\theta^{\mathtt{r}},2}\right] + \mathbb{E}^{\theta^{\mathtt{P}}}_{s_1^k,a_1^k}[V^{\pi}_{\hat{\theta}^{\mathtt{P}},\theta^{\mathtt{r}},2} - V^{\pi}_{\theta^{\mathtt{P}},\theta^{\mathtt{r}},2}]$$

$$= \mathbb{E}^{\hat{\theta}^{\mathtt{P}}}_{s_1^k,a_1^k}\left[V^{\pi}_{\hat{\theta}^{\mathtt{P}},\theta^{\mathtt{r}},2}\right] - \mathbb{E}^{\theta^{\mathtt{P}}}_{s_1^k,a_1^k}\left[V^{\pi}_{\hat{\theta}^{\mathtt{P}},\theta^{\mathtt{r}},2}\right] + V^{\pi}_{\hat{\theta}^{\mathtt{P}},\theta^{\mathtt{r}},2}(s_{2k}) - V^{\pi}_{\theta^{\mathtt{P}},\theta^{\mathtt{r}},2}(s_{2k}) + \zeta_1^k$$

$$= \sum_{h=1}^{H} \mathbb{E}^{\hat{\theta}^{\mathtt{P}}}_{s_{hk},a_{hk}}\left[V^{\pi}_{\hat{\theta}^{\mathtt{P}},\theta^{\mathtt{r}},h+1}\right] - \mathbb{E}^{\theta^{\mathtt{P}}}_{s_{hk},a_{hk}}\left[V^{\pi}_{\hat{\theta}^{\mathtt{P}},\theta^{\mathtt{r}},h+1}\right] + \zeta_{hk}$$

where $\zeta_{hk} = \mathbb{E}^{\theta^{\mathtt{P}}}_{s_{hk},a_{hk}}[V^{\pi}_{\hat{\theta}^{\mathtt{P}},\theta^{\mathtt{r}},h+1} - V^{\pi}_{\theta^{\mathtt{P}},\theta^{\mathtt{r}},h+1}] - \left(V^{\pi}_{\hat{\theta}^{\mathtt{P}},\theta^{\mathtt{r}},h+1}(s_{h+1k}) - V^{\pi}_{\theta^{\mathtt{P}},\theta^{\mathtt{r}},h+1}(s_{h+1k})\right)$ is a martingale sequence, and the last equality comes by induction. Here we consider the true reward parameter which verifies $|\mathbb{E}_{\theta^{\mathtt{r}}}[r(s,a)]| \leq 1$ by assumption, therefore $|\zeta_{hk}| \leq 2H$. Using the Azuma-Hoeffding inequality [BLM13], with probability at least $1 - \delta$

$$\sum_{k=1}^{K}\sum_{h=1}^{H} \zeta_{hk} \leq 2H\sqrt{2KH\log(1/\delta)}$$

We finish bounding the first term using Lemma 13, indeed

$$\mathbb{E}^{\hat{\theta}^{\mathtt{P}}}_{s_{hk},a_{hk}}\left[V^{\pi}_{\hat{\theta}^{\mathtt{P}},\theta^{\mathtt{r}},h+1}\right] - \mathbb{E}^{\theta^{\mathtt{P}}}_{s_{hk},a_{hk}}\left[V^{\pi}_{\hat{\theta}^{\mathtt{P}},\theta^{\mathtt{r}},h+1}\right] \leq H\sqrt{2\,\mathrm{KL}_{s_{hk},a_{hk}}(\theta^{\mathtt{P}},\hat{\theta}^{\mathtt{P}})}$$

$$\leq H\min\left\{1, \sqrt{2\,\mathrm{KL}_{s_{hk},a_{hk}}(\theta^{\mathtt{P}},\hat{\theta}^{\mathtt{P}})}\right\},$$

the last inequality follows because $\forall h,\ \mathbb{S}(V_{\hat{\theta}^{\mathtt{P}},\theta^{\mathtt{r}},h+1}) \leq H$.

**Remark 3.** *Traditionally, the expected value difference bound follows from the simulation lemma [RZSD21]. The simulation lemma incurs an extra $\sqrt{H}$ factor compared to our bound.*

We deduce that with probability at least $1 - \delta$:

$$\sum_{k=1}^{K} V_{\hat{\theta}^{\mathtt{P}},\theta^{\mathtt{r}}}(s_1^k) - V^{\pi}_{\theta^{\mathtt{P}},\theta^{\mathtt{r}}}(s_1^k)$$

$$\leq H\sum_{k=1}^{K}\min\left\{1, \sum_{h=1}^{H}\sqrt{2\,\mathrm{KL}_{s_{hk},a_{hk}}(\theta^{\mathtt{P}},\hat{\theta}^{\mathtt{P}})}\right\} + 2H\sqrt{2KH\log(1/\delta)} \quad (11)$$

*2) Bounding the sum of KL divergences.* we explicit the bound of inequality (11) using Assumption 1 along with properties of the exponential family (*cf.* Section D.3). We have for all $(s,a)$,

$$\forall \theta^{\mathtt{P}}, \theta^{\mathtt{P}\prime}, \quad \frac{\alpha^{\mathtt{P}}}{2}\|\theta^{\mathtt{P}\prime} - \theta^{\mathtt{P}}\|^2_{G_{s,a}} \leq \mathrm{KL}_{s,a}(\theta^{\mathtt{P}}, \theta^{\mathtt{P}\prime}) \leq \frac{\beta^{\mathtt{P}}}{2}\|\theta^{\mathtt{P}\prime} - \theta^{\mathtt{P}}\|^2_{G_{s,a}}. \quad (12)$$

This implies that

$$\mathrm{KL}_{s,a}\left(\hat{\theta}^{\mathtt{P}}(k), \theta^{\mathtt{P}}\right) \leq \frac{\beta^{\mathtt{P}}}{2}\left\|\theta^{\mathtt{P}} - \hat{\theta}^{\mathtt{P}}(k)\right\|^2_{G_{s,a}} \leq \beta^{\mathtt{P}}\left\|(\bar{G}_k^{\mathtt{P}})^{-1/2}G_{s,a}(\bar{G}_k^{\mathtt{P}})^{-1/2}\right\|\frac{1}{2}\left\|\theta^{\mathtt{P}} - \hat{\theta}^{\mathtt{P}}(k)\right\|^2_{\bar{G}_k^{\mathtt{P}}},$$

where $\bar{G}_k^{\mathtt{P}} \equiv \bar{G}_{(k-1)H}^{\mathtt{P}} := G_k + (\alpha^{\mathtt{P}})^{-1}\eta\mathbb{A}$ and $G_k \equiv \sum_{\tau=1}^{k-1}\sum_{h=1}^{H} G_{s_s^\tau,a_h^\tau}$.

From Corollary 8, with probability at least $1 - \delta$ and for all $k \in \mathbb{N}$

$$\left\|\theta^{\mathtt{P}} - \hat{\theta}^{\mathtt{P}}(k)\right\|^2_{\bar{G}_k^{\mathtt{P}}} \leq 2\beta^{\mathtt{P}}(k,\delta)/\alpha^{\mathtt{P}}.$$

Also, using Lemma 18, we have

$$\sum_{t=1}^{T}\sum_{h=1}^{H}\min\left\{1, \left\|(\bar{G}_k^{\mathtt{P}})^{-1/2}G_{s,a}(\bar{G}_k^{\mathtt{P}})^{-1/2}\right\|\right\} \leq 2d\log\left(1 + \alpha^{\mathtt{P}}\eta^{-1}B_{\varphi,\mathbb{A}}HK\right).$$

Combining these two results we obtain, with probability at least $1 - \delta$:

$$\sum_{t=1}^{T}\sum_{h=1}^{H}\min\left\{1, \mathrm{KL}_{s_h^t, a_h^t}\left(\hat{\theta}^{\mathrm{p}}(k), \theta^{\mathrm{p}}\right)\right\} \leq \frac{2\beta^{\mathrm{p}}}{\alpha^{\mathrm{p}}}\beta^{\mathrm{p}}(K, \delta)\gamma_K^{\mathrm{p}}. \tag{13}$$

**Remark 4.** *Notice that the minimum with 1 is crucial, indeed, without it the bound deteriorates by a factor $H$ as was the case in [CGM21].*

*3) Combining the bounds.* By applying Cauchy-Schwarz in inequality (11), we obtain, with probability at least $1 - \delta$, and for all $K \in \mathbb{N}$

$$\sum_{k=1}^{K} V_{\hat{\theta}^{\mathrm{p}}, \theta^{\mathrm{r}}}(s_1^k) - V_{\theta^{\mathrm{p}}, \theta^{\mathrm{r}}}^{\pi}(s_1^k) \leq H\sqrt{2\sum_{k=1}^{K}\sum_{h=1}^{H}\mathrm{KL}_{s_{hk}, a_{hk}}(\theta^{\mathrm{p}}, \hat{\theta}^{\mathrm{p}}) + 2H\sqrt{2KH\log(1/\delta)}}.$$

Injecting inequality (13) proves the desired result with probability at least $1 - 2\delta$. $\qquad\square$

### B.1.2 Reward estimation

Now, we provide the bound over the regret due to estimating the reward parameter.

**Lemma 5.** *With probability at least $1 - 3\delta$, the following result holds true.*

$$\sum_{k=1}^{K} V_{\hat{\theta}^{\mathrm{p}}, \tilde{\theta}^{\mathrm{r}}, 1}^{\pi}(s_1^k) - V_{\hat{\theta}^{\mathrm{p}}, \theta^{\mathrm{r}}, 1}^{\pi}(s_1^k) \leq \left(\sqrt{\frac{\beta^r(K, \delta)}{2\alpha^r}} + c\sqrt{(\max_{k \leq K}x_k)d\log(dK/\delta)}\right)\beta^r$$

$$\times \left(\sqrt{C_d\left(1 + \frac{\alpha^r B_{\varphi, A}H}{\eta}\right)} + \sqrt{K\log(e/\delta^2)}\right)\sqrt{Hd\log\left(1 + \alpha^r\eta^{-1}B_{\varphi, \mathbb{A}}HK\right)},$$

*where $\beta^{\mathrm{p}}(K, \delta) \triangleq \frac{\eta}{2}B_{\mathbb{A}}^2 + \gamma_K^{\mathrm{p}} + \log(1/\delta)$, and $\gamma_K^{\mathrm{p}} \triangleq d\log(1 + \frac{\beta^{\mathrm{p}}}{\eta}B_{\varphi, \mathbb{A}}HK)$. Also, $C_d \triangleq \frac{3d}{\log(2)}\log\left(1 + \frac{\alpha^r\|\mathbb{A}\|_2^2 B_{\varphi, \mathbb{A}}^2}{\eta\log(2)}\right)$, and $c$ is a universal constant.*

*Proof.* The reward estimation error in Equation (10) can be written explicitly. Indeed, using Lemma 17

$$V_{\hat{\theta}^{\mathrm{p}}, \tilde{\theta}^{\mathrm{r}}, 1}^{\pi}(s_1^k) - V_{\hat{\theta}^{\mathrm{p}}, \theta^{\mathrm{r}}, 1}^{\pi}(s_1^k) = \mathbb{E}_{(\tilde{s}_h)_{1 \leq h \leq H} \sim \pi|\hat{\theta}^{\mathrm{p}}, s_1^k}\left[\sum_{h=1}^{H}\frac{\mathbb{V}\mathrm{ar}_{\tilde{s}_h, \pi(\tilde{s}_h)}(r)}{2}B^{\top}M_{\tilde{\theta}^{\mathrm{r}} - \theta^{\mathrm{r}}}\varphi(\tilde{s}_h, \pi(\tilde{s}_h))\right]$$

$$\leq \mathbb{E}\left[\sum_{h=1}^{H}\frac{\mathbb{V}\mathrm{ar}_{\tilde{s}_h, \pi(\tilde{s}_h)}(r)}{2}\|\tilde{\theta}^{\mathrm{r}} - \theta^{\mathrm{r}}\|_{\bar{G}_k^{\mathrm{r}}}\|(B^{\top}A_i\varphi(\tilde{s}_h, \pi(\tilde{s}_h)))_{1 \leq i \leq d}\|_{(\bar{G}_k^{\mathrm{r}})^{-1}}\right]$$

$$\leq \|\tilde{\theta}^{\mathrm{r}} - \theta^{\mathrm{r}}\|_{\bar{G}_k^{\mathrm{r}}}\mathbb{E}\left[\sum_{h=1}^{H}\frac{\mathbb{V}\mathrm{ar}_{\tilde{s}_h, \pi(\tilde{s}_h)}(r)}{2}\|(B^{\top}A_i\varphi(\tilde{s}_h, \pi(\tilde{s}_h)))_{1 \leq i \leq d}\|_{(\bar{G}_k^{\mathrm{r}})^{-1}}\right]$$

$$\leq \|\tilde{\theta}^{\mathrm{r}} - \theta^{\mathrm{r}}\|_{\bar{G}_k^{\mathrm{r}}}\frac{\beta^{\mathrm{r}}}{2}\mathbb{E}\left[\underbrace{\sum_{h=1}^{H}\|(A_i\varphi(\tilde{s}_h, \pi(\tilde{s}_h)))_{1 \leq i \leq d}\|_{(\bar{G}_k^{\mathrm{r}})^{-1}}}_{\overset{\mathrm{def}}{=}\widetilde{\mathrm{traj}}_k}\right],$$

where $\mathrm{traj}_k \overset{\mathrm{def}}{=} \sum_{h=1}^{H}\|(A_i\varphi(s_h, \pi(s_h)))_{1 \leq i \leq d}\|_{(G_k^{\mathrm{r}})^{-1}}$.

**Bad rounds.** We separate the analysis of this estimation error into bad and good rounds. Here we analyze the bad rounds, which are define by the following set:

$$\mathcal{T} = \{k \in \mathbb{N}^*, \exists h \in [H], \|(A_i\varphi(\tilde{s}_h, \pi(\tilde{s}_h)))_{1 \leq i \leq d}\|_{(\bar{G}_k^{\mathrm{r}})^{-1}} \geq 1\}$$

540 *1)* We know that $\|(A_i\varphi(\tilde{s}_h,\pi(\tilde{s}_h)))_{1\le i\le d}(A_i\varphi(\tilde{s}_h,\pi(\tilde{s}_h)))^\top_{1\le i\le d}\|_2^2 \le \|\mathbb{A}\|_2^2 B_{\varphi,\mathbb{A}}^2$. Consequently,
541 according to Lemma 19

$$|\mathcal{T}| \le \frac{3d}{\log(2)}\log\left(1 + \frac{\alpha\|\mathbb{A}\|_2^2 B_{\varphi,\mathbb{A}}^2}{\eta\log(2)}\right).$$

542 *2)* Since $G_k$ is positive semi-definite, we have $\bar{G}_k^{\mathtt{r}} \succeq (\alpha^{\mathtt{r}})^{-1}\eta\mathbb{A}$, and in turn, for all state-action
543 couples $(s,a)$, $\|(\bar{G}_k^{\mathtt{r}})^{-1}G_{s,a}\| \le \frac{\alpha^{\mathtt{r}}}{\eta}\|\mathbb{A}^{-1}G_{s,a}\| \le \frac{\alpha^{\mathtt{r}}B_{\varphi,\mathbb{A}}}{\eta}$.

544 This further yields

$$\left\|I + (\bar{G}_k^{\mathtt{r}})^{-1}\sum_{h=1}^{H}G_{s_h^t,a_h^t}\right\| \le 1 + \sum_{h=1}^{H}\left\|(\bar{G}_k^{\mathtt{r}})^{-1}G_{s_h^t,a_h^t}\right\| \le 1 + \frac{\alpha^{\mathtt{r}}B_{\varphi,\mathbb{A}}H}{\eta}.$$

545 Let us define $\bar{G}_{k+H}^{\mathtt{r}} := \bar{G}_k^{\mathtt{r}} + \sum_{h=1}^{H}G_{s_h^k,a_h^k}$. Then,

$$\bar{G}_{k+H}^{-1}G_{s,a} = \left(I + (\bar{G}_k^{\mathtt{r}})^{-1}\sum_{h=1}^{H}G_{s_h^t,a_h^t}\right)^{-1}(\bar{G}_k^{\mathtt{r}})^{-1}G_{s,a}.$$

546 Therefore, for all pairs $(s,a)$,

$$\|(A_i\varphi(\tilde{s}_h,\pi(\tilde{s}_h)))_{1\le i\le d}\|_{(\bar{G}_k^{\mathtt{r}})^{-1}} = \sqrt{\mathrm{tr}((A_i\varphi(\tilde{s}_h,\pi(\tilde{s}_h)))^\top_{1\le i\le d}(\bar{G}_k^{\mathtt{r}})^{-1}(A_i\varphi(\tilde{s}_h,\pi(\tilde{s}_h)))_{1\le i\le d})}$$

$$= \sqrt{\mathrm{tr}(\left(1 + \frac{\alpha^{\mathtt{r}}B_{\varphi,A}H}{\eta}\right)(\bar{G}_{k+H}^{\mathtt{r}})^{-1}G_{s,a})}$$

$$\le \sqrt{\left(1 + \frac{\alpha^{\mathtt{r}}B_{\varphi,A}H}{\eta}\right)}\|(A_i\varphi(\tilde{s}_h,\pi(\tilde{s}_h)))_{1\le i\le d}\|_{(\bar{G}_{k+H}^{\mathtt{r}})^{-1}}$$

547 Since $\|(A_i\varphi(\tilde{s}_h,\pi(\tilde{s}_h)))_{1\le i\le d}\|_{(\bar{G}_{k+H}^{\mathtt{r}})^{-1}} \le 1$, we have $\|(A_i\varphi(\tilde{s}_h,\pi(\tilde{s}_h)))_{1\le i\le d}\|_{(\bar{G}_{k+H}^{\mathtt{r}})^{-1}} \le$
548 $\min\left\{1, \|(A_i\varphi(\tilde{s}_h,\pi(\tilde{s}_h)))_{1\le i\le d}\|_{(\bar{G}_k^{\mathtt{r}})^{-1}}\right\}$. Consequently

$$\sum_{h=1}^{H}\|(A_i\varphi(\tilde{s}_h,\pi(\tilde{s}_h)))_{1\le i\le d}\|_{(\bar{G}_{k+H}^{\mathtt{r}})^{-1}} \le \sqrt{Hd\log(1 + \alpha^{\mathtt{r}}\eta^{-1}B_{\varphi,\mathbb{A}}H)}.$$

549 *3)* From *1)* and *2)*, we deduce that the total regret induced by rounds from $\mathcal{T}$ is bounded.

$$\sum_{k\in\mathcal{T}}\sum_{h\in[H]}V_{\hat{\theta}^{\mathtt{p}},\tilde{\theta}^{\mathtt{r}},1}^{\pi}(s_1^k) - V_{\hat{\theta}^{\mathtt{p}},\theta^{\mathtt{r}},1}^{\pi}(s_1^k) \le \|\tilde{\theta}^{\mathtt{r}} - \theta^{\mathtt{r}}\|_{\bar{G}_k^{\mathtt{r}}}\frac{\beta^{\mathtt{r}}}{2}$$

$$\sqrt{\frac{3d}{\log(2)}\log\left(1 + \frac{\alpha^{\mathtt{r}}\|\mathbb{A}\|_2^2 B_{\varphi,\mathbb{A}}^2}{\eta\log(2)}\right)\left(1 + \frac{\alpha^{\mathtt{r}}B_{\varphi,A}H}{\eta}\right)Hd\log(1 + \alpha^{\mathtt{r}}\eta^{-1}B_{\varphi,\mathbb{A}}H)} \quad (14)$$

550 **Remark 5.** *The bad rounds analysis is one of our most important contributions as it enables us to*
551 *forgo clipping without consequences. Consequently, this is a novel method to control the reward*
552 *estimation error that improves on existing work for whom clipping was essential.*

553 **Good rounds.** Going forward we consider rounds from $\bar{\mathcal{T}}$. Let us define

$$\zeta_k' \stackrel{\text{def}}{=} \mathrm{traj}_k - \mathbb{E}_{(\tilde{s}_h)_{1\le h\le H}\sim\pi|\hat{\theta}^{\mathtt{p}},s_1^k}\left[\widetilde{\mathrm{traj}}_k\right].$$

554 where $\widetilde{\mathrm{traj}}_k$ is the same quantity as $\mathrm{traj}$ but with a random realization of state transitions.
555 Since all feature norms are smaller than one, $(\zeta_k')_k$ is a martingale sequence with $|\zeta_k'| \le$
556 $\sqrt{Hd\log\left(1 + \alpha^{\mathtt{r}}\eta^{-1}B_{\varphi,\mathbb{A}}HK\right)}$. We deduce that with probability at least $1 - \delta$:

$$\sum_{k=1}^{K}\zeta_k' \le \sqrt{2KHd\log\left(1 + \alpha^{\mathtt{r}}\eta^{-1}B_{\varphi,\mathbb{A}}HK\right)\log(1/\delta)}$$

557  Therefore, we have with probability at least $1 - 3\delta$:

$$\sum_{k \in \mathcal{T}^c} V^\pi_{\hat\theta^\mathtt{p}, \tilde\theta^\mathtt{r}, 1}(s_1^k) - V^\pi_{\hat\theta^\mathtt{p}, \theta^\mathtt{r}, 1}(s_1^k) \le \left( \sqrt{\frac{\beta^\mathtt{r}(K, \delta)}{2\alpha^\mathtt{r}}} + c\sqrt{(\max_k x_k) d \log(dK/\delta)} \right)$$
$$\times \beta^\mathtt{r} \sqrt{KHd \log\left(1 + \alpha^\mathtt{r}\eta^{-1} B_{\varphi, \mathbb{A}} KH\right) \log(e/\delta^2)}.$$

558  The last inequality follows from controlling the concentration of the reward parameter. First we ob-
559  serve that (Corollary 10) with probability at least $1 - \delta$, uniformly over $k \in \mathbb{N}$, $\left\| \theta^\mathtt{r} - \hat\theta^\mathtt{r}(k) \right\|^2_{\bar G^\mathtt{r}_k} \le$
560  $\frac{2}{\alpha^\mathtt{r}} \beta^\mathtt{r}(k, \delta)$. Second, we also have that for all $k \ge 1$, with probability at least $1 - \delta$, $\|\xi_k\|_{G^\mathtt{r}_k} \le$
561  $c\sqrt{x_k d \log(d/\delta)}$, we then use a union bound. Combining with Equation (14) we find

$$\sum_{k=1}^K V^\pi_{\hat\theta^\mathtt{p}, \tilde\theta^\mathtt{r}, 1}(s_1^k) - V^\pi_{\hat\theta^\mathtt{p}, \theta^\mathtt{r}, 1}(s_1^k) \le \left( \sqrt{\frac{\beta^\mathtt{r}(K, \delta)}{2\alpha^\mathtt{r}}} + c\sqrt{(\max_k x_k) d \log(dK/\delta)} \right)$$
$$\times \beta^\mathtt{r} \sqrt{KHd \log\left(1 + \alpha^\mathtt{r}\eta^{-1} B_{\varphi, \mathbb{A}} HK\right) \log(e/\delta^2)}.$$

562  This concludes the proof. $\qquad\qquad\square$

563  **Remark 6.** *If we use Lemma 17 without the martingale difference sequence, it will lead to a linear*
564  *regret. Indeed, the span of the sum of norms over an episode is of order $\sqrt{H}$. Using the martingale*
565  *technique instead allows us to retrieve a telescopic sum controlled using the elliptical lemma, this is*
566  *essential to obtaining a sub-linear regret bound.*

567  ## B.2  Learning error

568  We now start the control of an important regret term, due to the distance between the estimated value
569  function and the optimal value function.

570  **Lemma 6.** *If the variance parameter of the injected noise $(\xi_k)_k$ satisfies*

$$x_k \ge \left( H\sqrt{\frac{\beta^\mathtt{p}\beta^\mathtt{p}(k, \delta)}{\alpha^\mathtt{p}\alpha^r}} + \frac{\sqrt{\beta^r\beta^r(k, \delta) \min\{1, \frac{\alpha^\mathtt{p}}{\alpha^r}\}}}{2\alpha^r} \right),$$

571  *then the learning error is controlled with probability at least $1 - 2\delta$ as*

$$\sum_{k=1}^K V^\star_1(s_1^k) - V^\pi_{\hat\theta^\mathtt{p}, \hat\theta^r + \bar\xi_k, 1}(s_1^k) \le \frac{d\beta^r \sqrt{x_k}\left(1 + \sqrt{\log(d/\delta)}\right)}{\Phi(-1)} \sqrt{H \log\left(1 + \alpha^r\eta^{-1} B_{\varphi, \mathbb{A}} HK\right)}$$
$$\times \left( \sqrt{C_d \left(1 + \frac{\alpha^r B_{\varphi, A} H}{\eta}\right)} + \sqrt{K \log(e/\delta^2)} \right),$$

572  *where for $i \in [\mathtt{p}, \mathtt{r}]$, $\beta^i(K, \delta) \triangleq \frac{\eta}{2} B_\mathbb{A}^2 + \gamma_K^i + \log(1/\delta)$, and $\gamma_K^i \triangleq d \log(1 + \frac{\beta^i}{\eta} B_{\varphi, \mathbb{A}} HK)$. Also*
573  $C_d \stackrel{\text{def}}{=} \frac{3d}{\log(2)} \log\left(1 + \frac{\alpha^r \|\mathbb{A}\|_2^2 B_{\varphi, \mathbb{A}}^2}{\eta \log(2)}\right)$, *and $\Phi$ is the normal CDF.*

574  This result basically means that we are no longer obliged to follow optimistic value functions, the
575  perturbed estimation is enough to have a tight bound on the learning error.

576  ### B.2.1  Stochastic optimism

577  The goal here is to show that by injecting our carefully designed noise in the rewards we can ensure
578  optimism with a constant probability. Consider the optimal policy $\pi^\star$, we have:

$$(V_{\hat\theta^\mathtt{p}, \tilde\theta^r, 1} - V^\star_{\hat\theta^\mathtt{p}, \theta^r, 1})(s_1) \ge (Q^\star_{\hat\theta^\mathtt{p}, \tilde\theta^r, 1} - Q^\star_1)(s_1, \pi^\star(s_1))$$
$$\ge \underbrace{V^{\pi^\star}_{\hat\theta^\mathtt{p}, \theta^r}(s_1) - V^{\pi^\star}_{\hat\theta^\mathtt{p}, \theta^r}(s_1)}_{\text{first term}} + \underbrace{V^{\pi^\star}_{\hat\theta^\mathtt{p}, \hat\theta^r}(s_1) - V^{\pi^\star}_{\hat\theta^\mathtt{p}, \theta^r}(s_1)}_{\text{second term}} + \underbrace{V^{\pi^\star}_{\hat\theta^\mathtt{p}, \tilde\theta^r}(s_1) - V^{\pi^\star}_{\hat\theta^\mathtt{p}, \hat\theta^r}(s_1)}_{\text{third term}}$$

**First term.** By assumption, the expected reward under the true parameter satisfies $\mathbb{E}_{\theta^{\mathtt{r}}}[r(s,a)] \in [0,1]$, then $\mathbb{S}\left(\sum_{t=1}^{H}\mathbb{E}_{\theta^{\mathtt{r}}}[r(s_t, \pi(s_t))]\right) \leq H$. Consequently, the first term can be controlled using Lemma 13

$$V_{\theta^{\mathtt{p}},\theta^{\mathtt{r}}}^{\pi^\star}(s_1) - V_{\hat{\theta}^{\mathtt{p}},\theta^{\mathtt{r}}}^{\pi^\star}(s_1) \leq H\sqrt{\mathrm{KL}(P_{\hat{\theta}^{\mathtt{p}}}(s_2,\ldots,s_H), P_{\theta^{\mathtt{p}}}(s_2,\ldots,s_H))}$$

$$\leq H\sqrt{\mathbb{E}_{(\tilde{s}_t)_{t\in[H]}\sim\hat{\theta}^{\mathtt{p}}|s_1^k}\left[\sum_{t=1}^{H}\psi(\tilde{s}_{t+1})^\top M_{\hat{\theta}^{\mathtt{p}}-\theta^{\mathtt{p}}}\varphi(\tilde{s}_t, \pi^\star(\tilde{s}_t)) + Z_{\theta^{\mathtt{p}}}^{\mathtt{p}}(\tilde{s}_t, \pi^\star(\tilde{s}_t)) - Z_{\hat{\theta}^{\mathtt{p}}}^{\mathtt{p}}(\tilde{s}_t, \pi^\star(\tilde{s}_t))\right]}$$

Using Taylor's expansion, for all $h \in [H], \exists \theta_h \in [\theta^{\mathtt{p}}, \hat{\theta}^{\mathtt{p}}]$ such that:

$$\mathbb{E}_{(\tilde{s}_t)_{t\in[H]}\sim\hat{\theta}^{\mathtt{p}}|s_1^k}\left[\psi(\tilde{s}_{t+1})^\top M_{\hat{\theta}^{\mathtt{p}}-\theta^{\mathtt{p}}}\varphi(\tilde{s}_t, \pi^\star(\tilde{s}_t)) + Z_{\theta^{\mathtt{p}}}^{\mathtt{p}}(\tilde{s}_t, \pi^\star(\tilde{s}_t)) - Z_{\hat{\theta}^{\mathtt{p}}}^{\mathtt{p}}(\tilde{s}_t, \pi^\star(\tilde{s}_t))\right]$$

$$= \frac{1}{2}(\hat{\theta}^{\mathtt{p}} - \theta^{\mathtt{p}})^\top \mathbb{E}_{(\tilde{s}_t)_{t\in[H]}\sim\hat{\theta}^{\mathtt{p}}|s_1^k}\left[\nabla_{s_h,\pi^\star(s_h)}^2 Z^{\mathtt{p}}(\theta_h)\right](\hat{\theta}^{\mathtt{p}} - \theta^{\mathtt{p}})$$

$$\leq \frac{\beta^{\mathtt{p}}}{2}\mathbb{E}_{(\tilde{s}_t)_{t\in[H]}\sim\hat{\theta}^{\mathtt{p}}|s_1^k}\left[\|\hat{\theta}^{\mathtt{p}} - \theta^{\mathtt{p}}\|_{G_{\tilde{s}_h,\pi^\star(\tilde{s}_h)}}^2\right].$$

Define $u_k \overset{\text{def}}{=} \sum_{h=1}^{H}\mathbb{E}_{(\tilde{s}_t)_{t\in[H]}\sim\hat{\theta}^{\mathtt{p}}|s_1^k}\left[(A_i\varphi(\tilde{s}_h, \pi^\star(\tilde{s}_h)))_{i\in[d]}\right]$, then

$$V_{\theta^{\mathtt{p}},\theta^{\mathtt{r}}}^{\pi^\star}(s_1) - V_{\hat{\theta}^{\mathtt{p}},\theta^{\mathtt{r}}}^{\pi^\star}(s_1) \leq H\sqrt{\frac{\beta^{\mathtt{p}}}{2}\sum_{h=1}^{H}\mathbb{E}_{(\tilde{s}_t)_{t\in[H]}\sim\hat{\theta}^{\mathtt{p}}|s_1^k}\left[\|\hat{\theta}^{\mathtt{p}} - \theta^{\mathtt{p}}\|_{G_{\tilde{s}_h,\pi^\star(\tilde{s}_h)}}^2\right]}$$

$$\leq H\sqrt{\frac{\beta^{\mathtt{p}}}{2}}\left\|\hat{\theta}^{\mathtt{p}} - \theta^{\mathtt{p}}\right\|_{\sum_{h=1}^{H}\mathbb{E}_{(\tilde{s}_t)_{t\in[H]}\sim\hat{\theta}^{\mathtt{p}}|s_1^k}[G_{\tilde{s}_h,\pi^\star(\tilde{s}_h)}]}$$

$$\leq H\sqrt{\frac{\beta^{\mathtt{p}}}{2}}\left\|\hat{\theta}^{\mathtt{p}} - \theta^{\mathtt{p}}\right\|_{u_k u_k^\top}$$

$$\leq H\sqrt{\frac{\beta^{\mathtt{p}}}{2}}\left\|(\bar{G}_k^{\mathtt{p}})^{-1/2}u_k u_k^\top (\bar{G}_k^{\mathtt{p}})^{-1/2}\right\|\|\hat{\theta}^{\mathtt{p}} - \theta^{\mathtt{p}}\|_{\bar{G}_k^{\mathtt{p}}}$$

$$\leq H\sqrt{\frac{\beta^{\mathtt{p}}}{2}}\|u_k\|_{(\bar{G}_k^{\mathtt{p}})^{-1}}\|\hat{\theta}^{\mathtt{p}} - \theta^{\mathtt{p}}\|_{\bar{G}_k^{\mathtt{p}}}$$

The third line follows because $\forall x \in \mathbb{R}^d, \quad \|x\|_{\sum_{i=1}a_i a_i^\top} \leq \|x\|_{(\sum_{i=1}a_i)(\sum_{i=1}a_i)^\top}$, and the last one follows because $\mathrm{tr}(AB) \leq \mathrm{tr}(A)\mathrm{tr}(B)$ for any two real positive semi-definite matrices $A$ and $B$. We deduce, with probability at least $1 - \delta$:

$$V_{\theta^{\mathtt{p}},\theta^{\mathtt{r}}}^{\pi^\star}(s_1) - V_{\hat{\theta}^{\mathtt{p}},\theta^{\mathtt{r}}}^{\pi^\star}(s_1) \leq H\sqrt{\frac{\beta^{\mathtt{p}}\beta^{\mathtt{p}}(k,\delta)}{\alpha^{\mathtt{p}}}}\left\|\sum_{h=1}^{H}\mathbb{E}_{(\tilde{s}_t)_{t\in[H]}\sim\hat{\theta}^{\mathtt{p}}|s_1^k}\left[(A_i\varphi(\tilde{s}_h, \pi^\star(\tilde{s}_h)))_{i\in[d]}\right]\right\|_{(\bar{G}_k^{\mathtt{p}})^{-1}}$$

**Second term.** We have

$$V_{\hat{\theta}^{\mathtt{p}},\hat{\theta}^{\mathtt{r}}}^{\pi^\star}(s_1) - V_{\hat{\theta}^{\mathtt{p}},\theta^{\mathtt{r}}}^{\pi^\star}(s_1) = \mathbb{E}_{(\tilde{s}_t)_{t\in[H]}\sim\hat{\theta}^{\mathtt{p}}|s_1^k}\left[\sum_{t=1}^{H}\frac{\mathbb{V}\mathrm{ar}^{\theta_t^{\mathtt{r}}}(r)}{2}B^\top M_{\hat{\theta}^{\mathtt{r}}-\theta^{\mathtt{r}}}\varphi(\tilde{s}_t, \pi^\star(\tilde{s}_t))\right]$$

$$= (\hat{\theta}^{\mathtt{r}} - \theta^{\mathtt{r}})^\top \mathbb{E}_{(\tilde{s}_t)_{t\in[H]}\sim\hat{\theta}^{\mathtt{p}}|s_1^k}\left[\sum_{t=1}^{H}\frac{\mathbb{V}\mathrm{ar}^{\theta_t^{\mathtt{r}}}(r)}{2}(A_i\varphi(\tilde{s}_t, \pi^\star(\tilde{s}_t)))_{i\in[d]}\right]B$$

$$\leq \frac{\sqrt{\beta^{\mathtt{r}}}}{2}\|\hat{\theta}^{\mathtt{r}} - \theta^{\mathtt{r}}\|_{\bar{G}_k^{\mathtt{r}}}\left\|\mathbb{E}_{(\tilde{s}_t)_{t\in[H]}\sim\hat{\theta}^{\mathtt{p}}|s_1^k}\left[\sum_{t=1}^{H}(A_i\varphi(\tilde{s}_t, \pi^\star(\tilde{s}_t)))_{i\in[d]}\right]\right\|_{(\bar{G}_k^{\mathtt{r}})^{-1}}$$

The last inequality comes from Cauchy-Schwarz. Applying that the norm (sum) makes appear only symmetric matrices times the variances so that we can bound the latter by $\beta^{\mathtt{r}}$. We conclude that with probability at least $1 - \delta$,

$$V_{\hat{\theta}^{\mathtt{p}},\hat{\theta}^{\mathtt{r}}}^{\pi^\star}(s_1) - V_{\hat{\theta}^{\mathtt{p}},\tilde{\theta}^{\mathtt{r}}}^{\pi^\star}(s_1) \leq \frac{\beta^{\mathtt{r}}\sqrt{\beta^{\mathtt{r}}(k,\delta)}}{\sqrt{2\alpha^{\mathtt{r}}}}\left\|\mathbb{E}_{(\tilde{s}_t)_{t\in[H]}\sim\hat{\theta}^{\mathtt{p}}|s_1^k}\left[\sum_{t=1}^{H}(A_i\varphi(\tilde{s}_t, \pi^\star(\tilde{s}_t)))_{i\in[d]}\right]\right\|_{(\bar{G}_k^{\mathtt{r}})^{-1}}$$

We want to write all the norms in the same matrix. Therefore, with probability at least $1 - \delta$,

$$
V_{\hat{\theta}^{\mathrm{p}}, \hat{\theta}^{\mathrm{r}}}^{\pi^\star}(s_1) - V_{\hat{\theta}^{\mathrm{p}}, \tilde{\theta}^{\mathrm{r}}}^{\pi^\star}(s_1) \leq \sqrt{\frac{\beta^{\mathrm{r}} \beta^{\mathrm{r}}(k, \delta) \min\{1, \frac{\alpha^{\mathrm{p}}}{\alpha^{\mathrm{r}}}\}}{2\alpha^{\mathrm{r}}}}
$$

$$
\times \left\| \mathbb{E}_{(\tilde{s}_t)_{t \in [H]} \sim \hat{\theta}^{\mathrm{p}} | s_1^k} \left[ \sum_{t=1}^{H} (A_i \varphi(\tilde{s}_t, \pi^\star(\tilde{s}_t)))_{i \in [d]} \right] \right\|_{(\bar{G}_k^{\mathrm{p}})^{-1}}
$$

**Third term.** We have

$$
V_{\hat{\theta}^{\mathrm{p}}, \hat{\theta}^{\mathrm{r}}, 1}^{\pi^\star}(s_1) - V_{\hat{\theta}^{\mathrm{p}}, \tilde{\theta}^{\mathrm{r}}, 1}^{\pi^\star}(s_1) = \mathbb{E}_{(\tilde{s}_t)_{t \in [H]} \sim \hat{\theta}^{\mathrm{p}} | s_1^k} \left[ \sum_{t=1}^{H} \frac{\mathbb{Var}^{\theta_j^{\mathrm{r}}}(r)}{2} B^\top M_{\hat{\theta}^{\mathrm{r}} - \tilde{\theta}^{\mathrm{r}}} \varphi(\tilde{s}_t, \pi^\star(\tilde{s}_t)) \right]
$$

$$
= \xi_k^\top \, \mathbb{E}_{(\tilde{s}_t)_{t \in [H]} \sim \hat{\theta}^{\mathrm{p}} | s_1^k} \left[ \sum_{t=1}^{H} \frac{\mathbb{Var}^{\theta_j^{\mathrm{r}}}(r)}{2} (A_i \varphi(\tilde{s}_t, \pi^\star(\tilde{s}_t)))_{i \in [d]} \right] B
$$

Given the normal CDF $\Phi$, we obtain that with probability at least $\Phi(-1)$

$$
V_{\hat{\theta}^{\mathrm{p}}, \hat{\theta}^{\mathrm{r}}}^{\pi^\star}(s_1) - V_{\hat{\theta}^{\mathrm{p}}, \tilde{\theta}^{\mathrm{r}}}^{\pi^\star}(s_1) \geq \sqrt{x_k \alpha^{\mathrm{r}}} \left\| \left[ \sum_{t=1}^{H} \frac{\mathbb{Var}^{\theta_j^{\mathrm{r}}}(r)}{2} (A_i \varphi(\tilde{s}_t, \pi^\star(\tilde{s}_t)))_{i \in [d]} \right] \right\|_{(\bar{G}_k^{\mathrm{p}})^{-1}}
$$

Choosing $x_k \geq \left( H \sqrt{\frac{\beta^{\mathrm{p}} \beta^{\mathrm{p}}(k, \delta)}{\alpha^{\mathrm{p}} \alpha^{\mathrm{r}}}} + \frac{\sqrt{\beta^{\mathrm{r}} \beta^{\mathrm{r}}(k, \delta) \min\{1, \frac{\alpha^{\mathrm{p}}}{\alpha^{\mathrm{r}}}\}}}{2\alpha^{\mathrm{r}}} \right)$ and using Lemma 12, we find that the perturbed value function is optimistic with probability at least $\Phi(-1)$.

### B.2.2 Controlling the learning error

In this section we see the core difference with optimistic algorithms. On the one hand, optimistic approaches require the value function generating the agent's policy to be larger than the optimal one with large probability, and can therefore ensure that the learning error is negative. On the other hand, `BEF-RLSVI` only ensures that the value function is optimistic with a constant probability: intuitively when this event holds the learning happens, and if it does not then the policy is still close to a good one thanks to the decreasing estimation error.

**Upper bound on $V_1^\star$.** Let us draw $(\bar{\xi}_k)_{k \in [K]}$ i.i.d copies of $(\xi_k)_{k \in [K]}$. Define the optimism event at episode $k$:
$$
\bar{O}_k = \{ V_{\hat{\theta}^{\mathrm{p}}, \hat{\theta}^{\mathrm{r}} + \bar{\xi}_k, 1}(s_1^k) - V_1^\star(s_1^k) \geq 0 \} \tag{15}
$$
we know that $\mathbb{P}(\bar{O}_k) \geq \Phi(-1)$. This event provides the upper bound:
$$
V_1^\star(s_1^k) \leq \mathbb{E}_{\bar{\xi}_k | \bar{O}_k}[V_{\hat{\theta}^{\mathrm{p}}, \hat{\theta}^{\mathrm{r}} + \bar{\xi}_k, 1}(s_1^k)] \tag{16}
$$

**Lower bound on $V_{\hat{\theta}^{\mathrm{p}}, \tilde{\theta}^{\mathrm{r}}}$.** We define this bound with an optimization problem under concentration of the noise. Consider $\underline{V}_1(s_1^k)$ is the solution of
$$
\min_{\xi_k} V_{\hat{\theta}^{\mathrm{p}}, \hat{\theta}^{\mathrm{r}} + \xi_k, 1}(s_{1^k}) \tag{17}
$$
$$
\|\xi_k\|_{\bar{G}_k^{\mathrm{p}}} \leq \sqrt{x_k d \log(d/\delta)}, \quad \forall t \in [H]
$$

Under the concentration of our injected noise, we obtain
$$
\underline{V}_1(s_1^k) \leq V_{\hat{\theta}^{\mathrm{p}}, \tilde{\theta}^{\mathrm{r}}}(s_1^k) \tag{18}
$$

**Combining the error bounds.** Combining the upper bound of Equation (16) with the lower bound of Equation (18), we get, with probability at least $1 - \delta$:
$$
V_1^\star(s_1^k) - V_{\hat{\theta}^{\mathrm{p}}, \hat{\theta}^{\mathrm{r}} + \bar{\xi}_k, 1}(s_1^k) \leq \mathbb{E}_{\bar{\xi}_k | \bar{O}_k}[V_{\hat{\theta}^{\mathrm{p}}, \hat{\theta}^{\mathrm{r}} + \bar{\xi}_k, 1}(s_1^k) - \underline{V}_1(s_1^k)]
$$

611 Also, using the tower rule,

$$\mathbb{E}_{\bar{\xi}_k}[V_{\hat{\theta}^{\mathrm{p}},\hat{\theta}^{\mathrm{r}}+\bar{\xi}_k,1}(s_1^k) - \underline{V}_1(s_1^k)]$$
$$= \mathbb{E}_{\bar{\xi}_k|\bar{O}_k}[V_{\hat{\theta}^{\mathrm{p}},\hat{\theta}^{\mathrm{r}}+\bar{\xi}_k,1}(s_1^k) - \underline{V}_1(s_1^k)]\mathbb{P}(\bar{O}_k) + \mathbb{E}_{\bar{\xi}_k|\bar{O}_k^{\mathrm{c}}}[V_{\hat{\theta}^{\mathrm{p}},\hat{\theta}^{\mathrm{r}}+\bar{\xi}_k,1}(s_1^k) - \underline{V}_1(s_1^k)]\mathbb{P}(\bar{O}_k^{\mathrm{c}})$$

612 Therefore,

$$V_1^{\star}(s_1^k) - V_{\hat{\theta}^{\mathrm{p}},\hat{\theta}^{\mathrm{r}}+\bar{\xi}_k,1}(s_1^k)$$
$$\leq \left(\mathbb{E}_{\bar{\xi}_k}[V_{\hat{\theta}^{\mathrm{p}},\hat{\theta}^{\mathrm{r}}+\bar{\xi}_k,1}(s_1^k) - \underline{V}_1(s_1^k)] - \mathbb{E}_{\bar{\xi}_k|\bar{O}_k^{\mathrm{c}}}[V_{\hat{\theta}^{\mathrm{p}},\hat{\theta}^{\mathrm{r}}+\bar{\xi}_k,1}(s_1^k) - \underline{V}_1(s_1^k)]\mathbb{P}(\bar{O}_k^{\mathrm{c}})\right)/\mathbb{P}(\bar{O}_k)$$
$$= \left(\mathbb{E}_{\xi_k}[V_{\hat{\theta}^{\mathrm{p}},\hat{\theta}^{\mathrm{r}}+\xi_k,1}^{\pi}(s_1^k) - \underline{V}_1^{\pi}(s_1^k)] - \mathbb{E}_{\xi_k|\bar{O}_k^{\mathrm{c}}}[V_{\hat{\theta}^{\mathrm{p}},\hat{\theta}^{\mathrm{r}}+\xi_k,1}(s_1^k) - \underline{V}_1(s_1^k)]\mathbb{P}(\bar{O}_k^{\mathrm{c}})\right)/\mathbb{P}(\bar{O}_k).$$

613 The last line follows since $\xi_k$ and $\bar{\xi}_k$ are i.i.d.

614 The rest of the analysis proceeds similarly to the proof of the reward estimation.

615 Let us call the argument of the minimum in Equation (17) as $\underline{\xi}_k$. Using Lemma 17, we find

$$V_{\hat{\theta}^{\mathrm{p}},\tilde{\theta}^{\mathrm{r}},1}^{\pi}(s_1^k) - V_{\hat{\theta}^{\mathrm{p}},\hat{\theta}^{\mathrm{r}}+\underline{\xi}_k,1}^{\pi}(s_1^k)$$

$$= \mathbb{E}_{(\tilde{s}_h)_{1\leq h\leq H}\sim\pi|\hat{\theta}^{\mathrm{p}},s_1^k}\left[\sum_{h=1}^{H}\frac{\mathbb{V}\mathrm{ar}_{\tilde{s}_h,\pi(\tilde{s}_h)}(r)}{2}B^{\top}M_{\tilde{\theta}^{\mathrm{r}}-\hat{\theta}^{\mathrm{r}}-\underline{\xi}_k}\varphi(\tilde{s}_h,\pi(\tilde{s}_h))\right]$$

$$\leq \mathbb{E}\left[\sum_{h=1}^{H}\frac{\mathbb{V}\mathrm{ar}_{\tilde{s}_h,\pi(\tilde{s}_h)}(r)}{2}\|\tilde{\theta}^{\mathrm{r}}-\hat{\theta}^{\mathrm{r}}-\underline{\xi}_k\|_{\bar{G}_k^{\mathrm{p}}}\|(B^{\top}A_i\varphi(\tilde{s}_h,\pi(\tilde{s}_h)))_{1\leq i\leq d}\|_{(\bar{G}_k^{\mathrm{p}})^{-1}}\right]$$

$$\leq \|\tilde{\theta}^{\mathrm{r}}-\hat{\theta}^{\mathrm{r}}-\underline{\xi}_k\|_{\bar{G}_k^{\mathrm{p}}}\mathbb{E}\left[\sum_{h=1}^{H}\frac{\mathbb{V}\mathrm{ar}_{\tilde{s}_h,\pi(\tilde{s}_h)}(r)}{2}\|(B^{\top}A_i\varphi(\tilde{s}_h,\pi(\tilde{s}_h)))_{1\leq i\leq d}\|_{(\bar{G}_k^{\mathrm{p}})^{-1}}\right]$$

$$\leq \|\tilde{\xi}_k-\underline{\xi}_k\|_{\bar{G}_k^{\mathrm{p}}}\frac{\beta^{\mathrm{r}}}{2}\mathbb{E}\left[\sum_{h=1}^{H}\|(A_i\varphi(\tilde{s}_h,\pi(\tilde{s}_h)))_{1\leq i\leq d}\|_{(\bar{G}_k^{\mathrm{p}})^{-1}}\right]$$

616 Then,

$$\mathbb{E}_{\tilde{\xi}_k}\left[V_{\hat{\theta}^{\mathrm{p}},\tilde{\theta}^{\mathrm{r}},1}^{\pi}(s_1^k) - V_{\hat{\theta}^{\mathrm{p}},\hat{\theta}^{\mathrm{r}}+\underline{\xi}_k,1}^{\pi}(s_1^k)\right]$$
$$\leq \frac{\beta^{\mathrm{r}}}{2}\mathbb{E}_{\tilde{\xi}_k}[\|\tilde{\xi}_k-\underline{\xi}_k\|_{\bar{G}_k^{\mathrm{p}}}]\mathbb{E}_{(\tilde{s}_h)\sim\pi|\hat{\theta}^{\mathrm{p}}}\left[\sum_{h=1}^{H}\|(A_i\varphi(\tilde{s}_h,\pi(\tilde{s}_h)))_{1\leq i\leq d}\|_{(\bar{G}_k^{\mathrm{p}})^{-1}}\right].$$

617 Also,

$$\left|\mathbb{E}_{\xi_k|\bar{O}_k^{\mathrm{c}}}[V_{\hat{\theta}^{\mathrm{p}},\hat{\theta}^{\mathrm{r}}+\xi_k,1}(s_1^k) - \underline{V}_1(s_1^k)]\right|$$
$$\leq \frac{\beta^{\mathrm{r}}}{2}\mathbb{E}_{\tilde{\xi}_k|\bar{O}_k^{\mathrm{c}}}[\|\tilde{\xi}_k-\underline{\xi}_k\|_{\bar{G}_k^{\mathrm{p}}}]\mathbb{E}_{(\tilde{s}_h)\sim\pi|\hat{\theta}^{\mathrm{p}}}\left[\sum_{h=1}^{H}\|(A_i\varphi(\tilde{s}_h,\pi(\tilde{s}_h)))_{1\leq i\leq d}\|_{(\bar{G}_k^{\mathrm{p}})^{-1}}\right]$$
$$\leq \frac{\beta^{\mathrm{r}}}{2}\mathbb{E}_{\tilde{\xi}_k}[\|\tilde{\xi}_k-\underline{\xi}_k\|_{\bar{G}_k^{\mathrm{p}}}]\mathbb{E}_{(\tilde{s}_h)\sim\pi|\hat{\theta}^{\mathrm{p}}}\left[\sum_{h=1}^{H}\|(A_i\varphi(\tilde{s}_h,\pi(\tilde{s}_h)))_{1\leq i\leq d}\|_{(\bar{G}_k^{\mathrm{p}})^{-1}}\right].$$

618 We have a bound on the expected value of the sum of feature norms in the proof of Lemma 5. Also,

$$\mathbb{E}_{\tilde{\xi}_k}[\|\tilde{\xi}_k-\underline{\xi}_k\|_{\bar{G}_k^{\mathrm{p}}}] \leq \mathbb{E}_{\tilde{\xi}_k}[\|\tilde{\xi}_k\|_{\bar{G}_k^{\mathrm{p}}}] + \mathbb{E}_{\tilde{\xi}_k}[\|\underline{\xi}_k\|_{\bar{G}_k^{\mathrm{p}}}]$$
$$\leq \sqrt{\mathbb{E}_{\tilde{\xi}_k}[\|\tilde{\xi}_k\|_{\bar{G}_k^{\mathrm{p}}}^2]} + \sqrt{x_k d\log(d/\delta)}$$
$$\leq \sqrt{x_k d} + \sqrt{x_k d\log(d/\delta)}$$

619 The second line follows from Cauchy-Schwarz and by definition of $\underline{\xi}_k$. The last line is due to the
620 fact that $x_k(\bar{G}_k^{\mathrm{p}})^{-1} \sim \mathcal{N}(0, x_k I_d)$, which implies $\|\tilde{\xi}_k\|_{\bar{G}_k^{\mathrm{p}}}^2 \sim \mathcal{N}(0, dx_k)$. We conclude the proof by
621 taking the sum of feature norms from the proof of Lemma 5.

We conclude that with probability at least $1 - 2\delta$:

$$\sum_{k=1}^{K} V_1^\star(s_1^k) - V_{\hat{\theta}^{\mathtt{p}}, \hat{\theta}^{\mathtt{r}} + \bar{\xi}_k, 1}(s_1^k) \leq \frac{\beta^{\mathtt{r}}}{\Phi(-1)}(\sqrt{x_k d} + \sqrt{x_k d \log(d/\delta)})$$

$$\left[ \sqrt{\frac{3d}{\log(2)} \log\left(1 + \frac{\alpha^{\mathtt{r}} \|\mathbb{A}\|_2^2 B_{\varphi,\mathbb{A}}^2}{\eta \log(2)}\right) \left(1 + \frac{\alpha^{\mathtt{r}} B_{\varphi,A} H}{\eta}\right) H d \log(1 + \alpha^{\mathtt{r}} \eta^{-1} B_{\varphi,\mathbb{A}} H)} \right.$$

$$\left. + \sqrt{KHd \log\left(1 + \alpha^{\mathtt{r}} \eta^{-1} B_{\varphi,\mathbb{A}} HK\right) \log(e/\delta^2)} \right]$$

# C   Concentrations

## C.1   Concentration of the transition parameter

We recall the important concentration of the maximum likelihood estimator for general bilinear exponential families (*cf.* Theorem 1 of [CGM21]).

**Theorem 7.** *Suppose* $\{\mathcal{F}_t\}_{t=0}^{\infty}$ *is a filtration such that for each* $t$, *(i)* $s_{t+1}$ *is* $\mathcal{F}_t$*-measurable, (ii)* $(s_t, a_t)$ *is* $\mathcal{F}_{t-1}$ *measurable, and (iii) given* $(s_t, a_t)$, $s_{t+1} \sim P_{\theta^{\mathtt{p}}}^{\mathtt{p}}(\cdot \mid s_t, a_t)$ *according to the exponential family defined by Equation* (1). *Let* $\hat{\theta}^{\mathtt{p}}(k)$ *be the penalized MLE defined by Equation* (6), *and let* $Z_{s,a}^{\mathtt{p}}(\theta)$ *be strictly convex in* $\theta$ *for all* $(s, a)$. *Then, for any* $\delta \in (0, 1]$, *with probability at least* $1 - \delta$, *the following holds uniformly over all* $n \in \mathbb{N}$ :

$$\sum_{t=1}^{k} \mathrm{KL}_{s_t, a_t}\left(\hat{\theta}^{\mathtt{p}}(k), \theta^{\mathtt{p}}\right) + \frac{\eta}{2}\left\|\theta^{\mathtt{p}} - \hat{\theta}^{\mathtt{p}}(k)\right\|_{\mathbb{A}}^2 - \frac{\eta}{2}\|\theta^{\mathtt{p}}\|_{\mathbb{A}}^2 \leq \log\left(\frac{C_{\mathrm{A},k}^{\mathtt{p}}}{\delta}\right),$$

*where* $C_{\mathrm{A},k}^{\mathtt{p}} = \left(\int_{\mathbb{R}^d} \exp\left(-\frac{\eta}{2}\|\theta'\|_{\mathbb{A}}^2\right)d\theta'\right) / \left(\int_{\mathbb{R}^d} \exp\left(-\sum_{t=1}^{k} \mathrm{KL}_{s_t, a_t}(\theta_k, \theta') - \frac{\eta}{2}\|\theta' - \theta_k\|_{\mathbb{A}}^2\right)d\theta'\right).$

*Define* $G_{s,a} \stackrel{\mathrm{def}}{=} \left(\varphi(s,a)^\top A_i^\top A_j \varphi(s,a)\right)_{i,j \in [d]}$, *we have*

$$C_{\mathbb{A},k}^{\mathtt{p}} \leq \det\left(I + \beta^{\mathtt{p}} \eta^{-1} \mathbb{A}^{-1} \sum_{t=1}^{k} G_{s_t, a_t}\right),$$

*where* $\beta^{\mathtt{p}} = \sup_{\theta, s, a} \lambda_{\max}\left(\mathbb{C}_{s,a}^\theta[\psi(s')]\right)$.

A proof of this result can be found in the work [CGM21]. We provide an almost similar proof for the concentration of rewards in the next section.

**Corollary 8.** *The previous theorem implies a simple euclidean confidence region. Indeed, with probability at least* $1 - \delta$, *for all* $k \in \mathbb{N}$

$$\left\|\theta^{\mathtt{p}} - \hat{\theta}^{\mathtt{p}}(k)\right\|_{\bar{G}_n^{\mathtt{p}}}^2 \leq \frac{2}{\alpha^{\mathtt{p}}} \beta^{\mathtt{p}}(k, \delta),$$

*where* $\beta^{\mathtt{p}}(k, \delta) \stackrel{\mathrm{def}}{=} \beta_{(k-1)H}^{\mathtt{p}}(\delta) = \frac{2}{2}B_A^2 + \log\left(2C_{A,k}^{\mathtt{p}}/\delta\right)$.

*Proof.* The result follows from the following simple calculations:

$$\frac{1}{2}\left\|\theta^{\mathtt{p}} - \hat{\theta}^{\mathtt{p}}(k)\right\|_{\bar{G}_k}^2 = \frac{(\alpha^{\mathtt{p}})^{-1}\eta}{2}\left\|\theta^{\mathtt{p}} - \hat{\theta}^{\mathtt{p}}(k)\right\|_{\mathbb{A}}^2 + \sum_{\tau=1}^{k-1}\sum_{h=1}^{H} \frac{1}{2}\left\|\theta^{\mathtt{p}} - \hat{\theta}^{\mathtt{p}}(k)\right\|_{G_{s_h^\tau, a_h^\tau}}^2$$

$$\leq (\alpha^{\mathtt{p}})^{-1}\left(\frac{\eta}{2}\left\|\theta^{\mathtt{p}} - \hat{\theta}^{\mathtt{p}}(k)\right\|_{\mathbb{A}}^2 + \sum_{\tau=1}^{k-1}\sum_{h=1}^{H} \mathrm{KL}_{s_h^\tau, a_h^\tau}(\theta_k, \theta)\right).$$

$\square$

## C.2 Concentration of the reward parameter (contribution)

**Theorem 9.** *Suppose $\{\mathcal{F}_t\}_{t=0}^{\infty}$ is a filtration such that for each $t$, (i) $r(s_t, a_t)$ is $\mathcal{F}_t$-measurable, (ii) $(s_t, a_t)$ is $\mathcal{F}_{t-1}$ measurable, and (iii) given $(s_t, a_t)$, $r(s_t, a_t) \sim P_{\theta^r}^r(\cdot \mid s_t, a_t)$ according to the exponential family defined by (2). Let $\hat{\theta}^r(k)$ be the penalized MLE defined by Equation (8), and let $Z_{s,a}^r(\theta)$ be strictly convex in $\theta$ for all $(s, a)$. Then, for any $\delta \in (0, 1]$, with probability at least $1 - \delta$, the following holds uniformly over all $k \in \mathbb{N}$ :*

$$\sum_{t=1}^{k} \mathrm{KL}_{s_t, a_t}\left(\hat{\theta}^r(k), \theta^r\right) + \frac{\eta}{2}\left\|\theta^r - \hat{\theta}^r(k)\right\|_{\mathbb{A}}^2 - \frac{\eta}{2}\|\theta^r\|_{\mathbb{A}}^2 \leq \log\left(\frac{C_{\mathrm{A},k}^r}{\delta}\right),$$

*where $C_{\mathrm{A},k}^r = \left(\int_{\mathbb{R}^d} \exp\left(-\frac{\eta}{2}\|\theta'\|_{\mathbb{A}}^2\right)d\theta'\right) / \left(\int_{\mathbb{R}^d} \exp\left(-\sum_{t=1}^{k} \mathrm{KL}_{s_t, a_t}(\theta_k, \theta') - \frac{\eta}{2}\|\theta' - \theta_k\|_{\mathbb{A}}^2\right)d\theta'\right)$.
Define $G_{s,a} \overset{\mathrm{def}}{=} \left(\varphi(s,a)^\top A_i^\top A_j \varphi(s,a)\right)_{i,j \in [d]}$, we have*

$$C_{\mathbb{A},k} \leq \det\left(I + \beta^r \eta^{-1} \mathbb{A}^{-1} \sum_{t=1}^{k} G_{s_t, a_t}\right),$$

*where $\beta^r := \|B\|_2^2 \sup_{\theta, s, a} \mathbb{V}\mathrm{ar}_{s,a}^\theta(r)$.*

*Proof.* We proceed similar to the proof of Theorem 1 in [CG19].

**Step 1: Martingale construction.** First, observe that by assuming strict convexity, the log-partition function $Z_{s,a}^r$ becomes a Legendre function. Now for the conditional exponential family model, the KL divergence between $\mathbb{P}_{\theta^r}^r(\cdot \mid s, a)$ and $\mathbb{P}_{\theta'^r}^r(\cdot \mid s, a)$ can be expressed as a Bregman divergence associated to $Z_{s,a}^r$ with the parameters reversed, i.e.

$$\mathrm{KL}_{s,a}\left(\theta^r, \theta^{r\prime}\right) := \mathrm{KL}\left(P_{\theta^r}(\cdot \mid s, a), P_{\theta^{r\prime}}(\cdot \mid s, a)\right) = B_{Z_{s,a}}\left(\theta^{r\prime}, \theta^r\right).$$

Now, for any $\lambda \in \mathbb{R}^d$, we introduce the function $B_{Z_{n,\alpha}, \theta^r}(\lambda) = B_{Z_{n,\alpha}}\left(\theta^r + \lambda, \lambda\right)$ and define

$$M_n^\lambda = \exp\left(\lambda^\top S_n - \sum_{t=1}^{n} B_{Z_{n_t, a_t}, \theta^r}(\lambda)\right)$$

where $\forall i \leq d$, we denote $(S_n)_i = \sum_{t=1}^{n}\left(r(s_t, a_t) - \mathbb{E}_{s_t, a_t}^{\theta^r}[r]\right)B^\top A_i \varphi(s_t, a_t)$. Note that $M_n^\lambda > 0$ and it is $\mathcal{F}_{n^-}$ measurable. Furthermore, we have for all $(s, a)$,

$$\mathbb{E}_{s,a}^{\theta^r}\left[\exp\left(\sum_{i=1}^{d} \lambda_i \left(r(s_t, a_t) - \mathbb{E}_{s_t, a_t}^{\theta^r}[r]\right)B^\top A_i \varphi(s_t, a_t)\right)\right]$$

$$= \exp\left(-\lambda^\top \nabla Z_{s,a}^r(\theta^r)\right)\int_{\mathcal{S}} \exp\left(\sum_{i=1}^{d}(\theta_i^r + \lambda_i)B^\top A_i \varphi(s, a) - Z_{s,a}^r(\theta^r)\right)dr$$

$$= \exp\left(Z_{s,a}^r(\theta^r + \lambda) - Z_{s,a}^r(\theta^r) - \lambda^\top \nabla Z_{s,a}^r(\theta^r)\right) = \exp\left(B_{Z_{s,a}^r}(\theta^r)\right)$$

This implies $\mathbb{E}\left[\exp\left(\lambda^\top S_n\right) \mid \mathcal{F}_{n-1}\right] = \exp\left(\lambda^\top S_{n-1} + B_{Z_{n_n, a_n}, \theta^r}(\lambda)\right)$ thus $\mathbb{E}\left[M_n^\lambda \mid \mathcal{F}_{n-1}\right] = M_{n-1}^\lambda$. Therefore $\{M_n^\lambda\}_{n=0}^{\infty}$ is a non-negative martingale adapted to the filtration $\{\mathcal{F}_n\}_{n=0}^{\infty}$ and actually satisfies $\mathbb{E}\left[M_n^\lambda\right] = 1$. For any prior density $q(\theta)$ for $\theta$, we now define a mixture of martingales

$$M_n = \int_{\mathbb{R}^d} M_n^\lambda q\left(\theta^r + \lambda\right)d\lambda \tag{19}$$

Then $\{M_n\}_{n=0}^{\infty}$ is also a non-negative martingale adapted to $\{\mathcal{F}_n\}_{n=0}^{\infty}$ and in fact, $\mathbb{E}[M_n] = 1$.

**Step 2: Method of mixtures.** Considering the prior density $\mathcal{N}(0, (\eta\mathbb{A})^{-1})$, we obtain from (19) that

$$M_n = c_0 \int_{\mathbb{R}^d} \exp\left( \lambda^\top S_n - \sum_{t=1}^n B_{Z^{\mathtt{r}}_{x_t,a_t},\theta^{\mathtt{r}}}(\lambda) - \frac{\eta}{2}\|\theta^{\mathtt{r}} + \lambda\|_{\mathbb{A}}^2 \right) d\lambda, \tag{20}$$

where $c_0 = \frac{1}{\int_{\mathbb{R}^d} \exp\left(-\frac{\eta}{2}\|\theta'\|_{\mathbb{A}}^2\right)d\theta'}$. We now introduce the function $Z^{\mathtt{r}}_n(\theta) = \sum_{t=1}^n Z^{\mathtt{r}}_{s_t,a_t}(\theta)$. Note that $Z^{\mathtt{r}}_n$ is a also Legendre function and its associated Bregman divergence satisfies

$$B_{Z^{\mathtt{r}}_n}(\theta',\theta) = \sum_{t=1}^n \left( Z^{\mathtt{r}}_{s_t,a_t}(\theta') - Z^{\mathtt{r}}_{s_t,a_t}(\theta) - (\theta'-\theta)^\top \nabla Z^{\mathtt{r}}_{S_t,a_t}(\theta) \right) = \sum_{t=1}^n B_{Z^{\mathtt{r}}_{s_t,\alpha_t}}(\theta',\theta)$$

Furthermore, we have $\sum_{t=1}^n B_{Z^{\mathtt{r}}_{s_t,\alpha_t},\theta^{\mathtt{r}}}(\lambda) = B_{Z^{\mathtt{r}}_n,\theta^{\mathtt{r}}}(\lambda)$. From the penalized likelihood formula (8), recall that

$$\forall i \leq d, \quad \sum_{t=1}^n \nabla_i Z^{\mathtt{r}}_{s_t,a_t}\left(\hat{\theta}^{\mathtt{r}}(k)\right) + \frac{\eta}{2}\nabla_i\|\hat{\theta}^{\mathtt{r}}(k)\|_{\mathbb{A}}^2 = \sum_{t=1}^k r_t B^\top A_i \varphi(s_t,a_t).$$

This yields

$$S_k = \sum_{t=1}^k \left( \nabla Z^{\mathtt{r}}_{s_t,a_t}\left(\hat{\theta}^{\mathtt{r}}(k)\right) - \nabla Z^{\mathtt{r}}_{s_t,a_t}(\theta^{\mathtt{r}}) \right) + \eta\mathbb{A}\hat{\theta}^{\mathtt{r}}(k) = \nabla Z^{\mathtt{r}}_k\left(\hat{\theta}^{\mathtt{r}}(k)\right) - \nabla Z^{\mathtt{r}}_k(\theta^{\mathtt{r}}) + \eta\mathbb{A}\hat{\theta}^{\mathtt{r}}(k) \tag{21}$$

We now obtain from (20) and (21) that

$$M_k = c_0 \cdot \exp\left(-\frac{\eta}{2}\|\theta^{\mathtt{r}}\|_A^2\right) \int_{\mathbb{R}^d} \exp\left( \lambda^\top x_k - B_{Z_k,\theta^*}(\lambda) + g_k(\lambda) \right) d\lambda, \tag{22}$$

where we introduced $g_k(\lambda) = \frac{\eta}{2}\left( 2\lambda^\top\mathbb{A}\hat{\theta}^{\mathtt{r}}(k) + \|\theta^{\mathtt{r}}\|_{\mathbb{A}}^2 - \|\theta^{\mathtt{r}} + \lambda\|_{\mathbb{A}}^2 \right)$ and $x_k = \nabla Z^{\mathtt{r}}_k\left(\hat{\theta}^{\mathtt{r}}(k)\right) - \nabla Z^{\mathtt{r}}_k(\theta^{\mathtt{r}})$.

Now, note that $\sup_{\lambda\in\mathbb{R}^d} g_k(\lambda) = \frac{\eta}{2}\left\|\theta^{\mathtt{r}} - \hat{\theta}^{\mathtt{r}}(k)\right\|_{\mathbb{A}}^2$, where the supremum is attained at $\lambda^\star = \hat{\theta}^{\mathtt{r}}(k) - \theta^{\mathtt{r}}$. We then have

$$
\begin{aligned}
g_k(\lambda) &= g_n(\lambda) + \sup_{\lambda\in\mathbb{R}^\star} g_k(\lambda) - g_k(\lambda^\star) \\
&= \frac{\eta}{2}\left\|\hat{\theta}^{\mathtt{r}}(k) - \theta^{\mathtt{r}}\right\|_{\mathbb{A}}^2 + \eta(\lambda - \lambda^\star)^\top \mathbb{A}(\theta^{\mathtt{r}} + \lambda^\star) + \frac{\eta}{2}\|\theta^{\mathtt{r}} + \lambda^\star\|_A^2 - \frac{\eta}{2}\|\theta^{\mathtt{r}} + \lambda\|_{\mathbb{A}}^2 \\
&= B_{Z^{\mathtt{r}}_0}\left(\theta^{\mathtt{r}}, \hat{\theta}^{\mathtt{r}}(k)\right) + (\lambda - \lambda^\star)^\top \nabla Z^{\mathtt{r}}_0(\theta^{\mathtt{r}} + \lambda^\star) + Z^{\mathtt{r}}_0(\theta^{\mathtt{r}} + \lambda^\star) - Z^{\mathtt{r}}_0(\theta^{\mathtt{r}} + \lambda) \tag{23}
\end{aligned}
$$

where we have introduced the Legendre function $Z^{\mathtt{r}}_0(\theta) = \frac{\eta}{2}\|\theta\|_{\mathbb{A}}^2$. We now have from (27) that

$$
\begin{aligned}
\sup_{\lambda\in\mathbb{R}^d} &\left( \lambda^\top x_n - B_{Z^{\mathtt{r}}_n,\theta^{\mathtt{r}}}(\lambda) \right) \\
&= B^\star_{Z^{\mathtt{r}}_n,\theta^{\mathtt{r}}}(x_n) = B^\star_{Z^{\mathtt{r}}_n,\theta^{\mathtt{r}}}\left( \nabla Z^{\mathtt{r}}_n\left(\hat{\theta}^{\mathtt{r}}(n)\right) - \nabla Z^{\mathtt{r}}_n(\theta^{\mathtt{r}}) \right) = B_{Z^{\mathtt{r}n}}\left(\theta^{\mathtt{r}}, \hat{\theta}^{\mathtt{r}}(n)\right).
\end{aligned}
$$

Further, any optimal $\lambda$ must satisfy

$$\nabla Z^{\mathtt{r}}_n(\theta^{\mathtt{r}} + \lambda) - \nabla Z^{\mathtt{r}}_n(\theta^{\mathtt{r}}) = x_n \implies \nabla Z^{\mathtt{r}}_n(\theta^{\mathtt{r}} + \lambda) = \nabla Z^{\mathtt{r}}_n\left(\hat{\theta}^{\mathtt{r}}(n)\right).$$

One possible solution is $\lambda = \lambda^\star$. Now, since $Z^{\mathtt{r}}_n$ is strictly convex, the supremum is indeed attained at $\lambda = \lambda^\star$. We then have

$$
\begin{aligned}
\lambda^\top x_n &- B_{Z^{\mathtt{r}}_n,\theta^{\mathtt{r}}}(\lambda) \\
&= \lambda^\top x_n - B_{Z^{\mathtt{r}}_n,\theta^{\mathtt{r}}}(\lambda) + B_{Z^{\mathtt{r}}_n}\left(\theta^{\mathtt{r}}, \hat{\theta}^{\mathtt{r}}(n)\right) - \left( \lambda^\star x_n - B_{Z^{\mathtt{r}}_n,\theta^{\mathtt{r}}}(\lambda^\star) \right) \\
&= B_{Z^{\mathtt{r}}_n}\left(\theta^{\mathtt{r}}, \hat{\theta}^{\mathtt{r}}(n)\right) + (\lambda - \lambda^\star)^\top \nabla Z^{\mathtt{r}}_n(\theta^{\mathtt{r}} + \lambda^\star) + B_{Z^{\mathtt{r}}_n,\theta^*}(\lambda^\star) - B_{Z^{\mathtt{r}}_n,\theta^*}(\lambda) \\
&\quad - (\lambda - \lambda^\star)^\top \nabla Z^{\mathtt{r}}_n(\theta^{\mathtt{r}}) \\
&= B_{Z^{\mathtt{r}}_n}\left(\theta^{\mathtt{r}}, \hat{\theta}^{\mathtt{r}}(n)\right) + (\lambda - \lambda^\star)^\top \nabla Z^{\mathtt{r}}_n(\theta^{\mathtt{r}} + \lambda^\star) + Z^{\mathtt{r}}_n(\theta^{\mathtt{r}} + \lambda^\star) - Z^{\mathtt{r}}_n(\theta^{\mathtt{r}} + \lambda) \tag{24}
\end{aligned}
$$

Plugging Equation (23) and Equation (24) in Equation (22), we obtain

$$
M_n = c_0 \cdot \exp\left( \sum_{j \in \{0,n\}} B_{Z_j^{\mathtt{r}}}\left(\theta^{\mathtt{r}}, \theta_j\right) - \frac{\eta}{2} \left\|\theta^{\mathtt{r}}\right\|_A^2 \right)
$$

$$
\times \int_{\mathbb{R}^d} \exp\left( \sum_{j \in \{0,n\}} \left( (\lambda - \lambda^\star)^\top \nabla Z_j^{\mathtt{r}}(\theta^{\mathtt{r}} + \lambda^\star) + Z_j^{\mathtt{r}}(\theta^{\mathtt{r}} + \lambda^\star) - Z_j^{\mathtt{r}}(\theta^{\mathtt{r}} + \lambda) \right) \right) d\lambda
$$

$$
= c_0 \cdot \exp\left( \sum_{j \in \{0,n\}} B_{Z_j^{\mathtt{r}}}\left(\theta^{\mathtt{r}}, \hat{\theta}^{\mathtt{r}}(n)\right) - \frac{\eta}{2} \left\|\theta^{\mathtt{r}}\right\|^2 \right)
$$

$$
\times \exp\left( - \sum_{j \in \{0,n\}} \left( (\theta^{\mathtt{r}} + \lambda^\star)^\top \nabla Z_j^{\mathtt{r}}(\theta^{\mathtt{r}} + \lambda^\star) - Z_j^{\mathtt{r}}(\theta^{\mathtt{r}} + \lambda^\star) \right) \right)
$$

$$
\times \int_{\mathbb{R}^d} \exp\left( \sum_{j \in \{0,n\}} \left( (\theta^{\mathtt{r}} + \lambda)^\top \nabla Z_j^{\mathtt{r}}(\theta^{\mathtt{r}} + \lambda^\star) - Z_j^{\mathtt{r}}(\theta^{\mathtt{r}} + \lambda) \right) \right) d\lambda
$$

$$
= \frac{c_0}{c_{\mathtt{n}}} \exp\left( \sum_{j \in \{0,n\}} B_{Z_j^{\mathtt{r}}}\left(\theta^{\mathtt{r}}, \hat{\theta}^{\mathtt{r}}(n)\right) - \frac{\eta}{2} \left\|\theta^{\mathtt{r}}\right\|_{\mathbb{A}}^2 \right)
$$

$$
\times \frac{\int_{\mathbb{R}^d} \exp\left( \sum_{j \in \{0,n\}} \left( (\theta^{\mathtt{r}} + \lambda)^\top \nabla Z_j^{\mathtt{r}}(\theta^{\mathtt{r}} + \lambda^\star) - Z_j^{\mathtt{r}}(\theta^{\mathtt{r}} + \lambda) \right) \right) d\lambda}{\int_{\mathbb{R}^d} \exp\left( \sum_{j \in \{0,n\}} \left( (\theta')^\top \nabla Z_j^{\mathtt{r}}(\theta^{\mathtt{r}} + \lambda^\star) - Z_j^{\mathtt{r}}(\theta') \right) \right) d\theta'}
$$

$$
= \frac{c_0}{c_n} \cdot \exp\left( B_{Z_n}\left(\theta^{\mathtt{r}}, \hat{\theta}^{\mathtt{r}}(n)\right) + B_{Z_0}\left(\theta^{\mathtt{r}}, \hat{\theta}^{\mathtt{r}}(n)\right) - \frac{\eta}{2} \left\|\theta^{\mathtt{r}}\right\|_{\mathbb{A}}^2 \right),
$$

where we introduced $c_n = \frac{\exp\left(\sum_{j \in \{0,n\}}\left((\theta^{\mathtt{r}} + \lambda^\star)^\top \nabla Z_j^{\mathtt{r}}(\theta^{\mathtt{r}} + \lambda^\star) - Z_j^{\mathtt{r}}(\theta^{\mathtt{r}} + \lambda^\star)\right)\right)}{\int_{\mathbb{R}^d} \exp\left(\sum_{j \in \{0,n\}}\left((\theta')^\top \nabla Z_j^{\mathtt{r}}(\theta^{\mathtt{r}} + \lambda^\star) - Z_j^{\mathtt{r}}(\theta')\right)\right) d\theta'}$. Since $\lambda^\star = \hat{\theta}^{\mathtt{r}}(n) - \theta^{\mathtt{r}}$,
we have

$$
c_n = \frac{1}{\int_{\mathbb{R}^d} \exp\left(-\sum_{j \in \{0,n\}} B_{Z_j^{\mathtt{r}}}(\theta', \theta^{\mathtt{r}} + \lambda^\star)\right) d\theta'} = \frac{1}{\int_{\mathbb{R}^d} \exp\left(-\sum_{t=1}^n B_{Z_{s_t,a_t}}\left(\theta', \hat{\theta}^{\mathtt{r}}(n)\right) - \frac{\eta}{2}\left\|\theta' - \hat{\theta}^{\mathtt{r}}(n)\right\|_{\mathbb{A}'}^2\right) d\theta'}
$$

Therefore, we have from (5) that

$$
C_{A,n} := \frac{c_n}{c_0} = \frac{\int_{\mathbb{R}^d} \exp\left(-\frac{\eta}{2}\left\|\theta'\right\|_{\mathbb{A}}^2\right) d\theta'}{\int_{\mathbb{R}^d} \exp\left(-\sum_{t=1}^n \mathrm{KL}_{s_t,a_t}\left(\hat{\theta}^{\mathtt{r}}(n), \theta'\right) - \frac{\eta}{2}\left\|\theta' - \hat{\theta}^{\mathtt{r}}(n)\right\|_{\mathbb{A}}^2\right) d\theta'}
$$

An application of Markov's inequality now yields

$$
\mathbb{P}\left[\sum_{t=1}^n \mathrm{KL}_{s_t,a_t}\left(\hat{\theta}^{\mathtt{r}}(n), \theta^{\mathtt{r}}\right) + \frac{\eta}{2}\left\|\theta^{\mathtt{r}} - \hat{\theta}^{\mathtt{r}}(n)\right\|_{\mathbb{A}}^2 - \frac{\eta}{2}\left\|\theta^{\mathtt{r}}\right\|_{\mathbb{A}}^2 \geq \log\left(\frac{C_{A,n}}{\delta}\right)\right] = \mathbb{P}\left[M_n \geq \frac{1}{\delta}\right] \leq \delta\mathbb{E}\left[M_n\right] = \delta
$$

**Step 3: A stopped martingale and its control.** Let $N$ be a stopping time with respect to the filtration $\{\mathcal{F}_n\}_{n=0}^\infty$. Now, by the martingale convergence theorem, $M_\infty = \lim_{n \to \infty} M_n$ is almost surely well-defined, and thus $M_N$ is well-defined as well irrespective of whether $N < \infty$ or not. Let $Q_n = M_{\min\{N,n\}}$ be a stopped version of $\{M_n\}_n$. Then an application of Fatou's lemma yields

$$
\mathbb{E}\left[M_N\right] = \mathbb{E}\left[\liminf_{n \to \infty} Q_n\right] \leq \liminf_{n \to \infty} \mathbb{E}\left[Q_n\right] = \liminf_{n \to \infty} \mathbb{E}\left[M_{\min\{N,n\}}\right] \leq 1,
$$

since the stopped martingale $\left\{M_{\min\{N,n\}}\right\}_{n \geq 1}$ is also a martingale. Therefore, by the properties of $M_n$, (12) also holds for any random stopping time $N < \infty$. To complete the proof, we now employ a random stopping time construction as in Abbasi-Yadkori et al. (2011)

We define a random stopping time $N$ by

$$N = \min\left\{ n \geq 1 : \sum_{t=1}^{n} \mathrm{KL}_{s_t,a_t}\left(\hat{\theta}^{\mathtt{r}}(n), \theta^{\mathtt{r}}\right) + \frac{\eta}{2}\left\|\theta^{\mathtt{r}} - \hat{\theta}^{\mathtt{r}}(n)\right\|_{A}^{2} - \frac{\eta}{2}\|\theta^{\mathtt{r}}\|_{A}^{2} \geq \log\left(\frac{C_{A,n}}{\delta}\right) \right\}$$

with $\min\{\emptyset\} := \infty$ by convention. We then have

$$\mathbb{P}\left[\exists n \geq 1, \sum_{t=1}^{n} \mathrm{KL}_{s_t,a_t}\left(\hat{\theta}^{\mathtt{r}}(n), \theta^{\mathtt{r}}\right) + \frac{\eta}{2}\left\|\theta^{\mathtt{r}} - \hat{\theta}^{\mathtt{r}}(n)\right\|_{\mathrm{A}}^{2} - \frac{\eta}{2}\|\theta^{\mathtt{r}}\|_{\mathrm{A}}^{2} \geq \log\left(\frac{C_{A,n}}{\delta}\right)\right] = \mathbb{P}[N < \infty] \leq \delta,$$

which concludes the proof of the first part.

**Proof of second part: upper bound on $C_{A,n}$.** First, we have for some $\tilde{\theta} \in \left[\hat{\theta}^{\mathtt{r}}(n), \theta'\right]_{\infty}$ that

$$\mathrm{KL}_{s,a}\left(\hat{\theta}^{\mathtt{r}}(n), \theta'\right) = \frac{1}{2}\sum_{i,j=1}^{d}\left(\theta' - \hat{\theta}^{\mathtt{r}}(n)\right)_{i} \mathbb{V}\mathrm{ar}_{s,a}^{\theta}(r) \times \varphi(s,a)^{\top} A_i^{\top} BB^{\top} A_j \varphi(s,a)\left(\theta' - \hat{\theta}^{\mathtt{r}}(n)\right)_{j} \tag{25}$$

Now (25) implies that

$$\sum_{t=1}^{n} \mathrm{KL}_{s_t,a_t}\left(\hat{\theta}^{\mathtt{r}}(n), \theta'\right) \leq \frac{\beta}{2}\sum_{t=1}^{n}\sum_{i,j=1}^{d}\left(\theta' - \hat{\theta}^{\mathtt{r}}(n)\right)_{i}\varphi\left(s_t, a_t\right)^{\top} A_i^{\top} A_j \varphi\left(s_t, a_t\right)\left(\theta' - \hat{\theta}^{\mathtt{r}}(n)\right)_{j}$$

$$= \frac{\beta^{\mathtt{r}}}{2}\left\|\theta' - \hat{\theta}^{\mathtt{r}}(n)\right\|_{\sum_{t=1}^{n} G_{s_t,a_t}}^{2},$$

where $\beta^{\mathtt{r}} := \lambda_{\max}\left(BB^{\top}\right) \times \sup_{\theta,s,a} \mathbb{V}\mathrm{ar}_{s,a}^{\theta}(r)$ and $\forall i,j \leq d$, $(G_{s,a})_{i,j} := \varphi(s,a)^{\top} A_i^{\top} A_j \varphi(s,a)$. Therefore, we obtain

$$C_{\mathrm{A},n} \leq \frac{\int_{\mathbb{R}^d} \exp\left(-\frac{\eta}{2}\|\theta'\|_{\mathrm{A}}^2\right) d\theta'}{\int_{\mathbb{R}^d} \exp\left(-\frac{1}{2}\left\|\theta' - \hat{\theta}^{\mathtt{r}}(n)\right\|_{\left(\beta^{\mathtt{r}}\sum_{t=1}^{n} G_{s_t,a_t} + \eta\mathrm{A}\right)}^2\right) d\theta'}$$

$$= \frac{(2\pi)^{d/2}}{\det(\eta\mathbb{A})^{1/2}} \times \frac{\det\left(\beta^{\mathtt{r}}\sum_{t=1}^{n} G_{s_t,a_t} + \eta\mathbb{A}\right)^{1/2}}{(2\pi)^{d/2}} = \det\left(I + \beta^{\mathtt{r}}\eta^{-1}\mathbb{A}^{-1}\sum_{t=1}^{n} G_{s_t,a_t}\right),$$

which completes the proof of the second part.

$\square$

**Corollary 10.** *Here also, the theorem implies a euclidean control. With probability at least $1 - \delta$ uniformly over $k \in \mathbb{N}$*

$$\left\|\theta^{r} - \hat{\theta}^{r}(k)\right\|_{\bar{G}_k^r}^2 \leq \frac{2}{\alpha^r}\beta^r(k,\delta),$$

*where $\beta^r(k,\delta) \stackrel{\text{def}}{=} \beta_{(k-1)H}^r(\delta) = \frac{2}{2}B_A^2 + \log\left(2C_{A,k}^r/\delta\right)$.*

## C.3 Gaussian concentration and anti-concentration

**Lemma 11** (Gaussian concentration, ref. Appendix A in [AL17]). *Let $\bar{\xi}_{tk} \sim \mathcal{N}(0, H\nu_k(\delta)\Sigma_{tk}^{-1})$. For any $\delta > 0$, with probability $1 - \delta$*

$$\|\bar{\xi}_{tk}\|_{\Sigma_{tk}} \leq c\sqrt{Hd\nu_k(\delta)\log(d/\delta)} \tag{26}$$

*for some absolute constant $c$.*

**Lemma 12** (Gaussian anti-concentration, ref. Appendix A in [AL17]). *Let $\xi \sim \mathcal{N}(0, I_d)$, for any $u \in \mathbb{R}^d$ with $\|u\| = 1$, we have:*

$$\mathbb{P}(u^{\top}\xi \geq 1) \geq \Phi(-1),$$

*where $\Phi$ is the normal CDF.*

Thanks to lower bounds on the error function, we have the following bound on the probability of anti-concentration $\Phi(-1) \geq 1/(4\sqrt{e\pi})$.

 # D   Technical results

 ## D.1   A transportation lemma

 For any function $f : \mathcal{X} \to \mathbb{R}$, we define its span as $\mathbb{S}(f) := \max_{x \in \mathcal{X}} f(x) - \min_{x \in \mathcal{X}} f(x)$.
 For a probability distribution $P$ supported on the set $\mathcal{X}$, let $\mathbb{E}_P[f] := \mathbb{E}_P[f(X)]$ and $\mathbb{V}_P[f] :=$
 $\mathbb{V}_P[f(X)] = \mathbb{E}_P\left[f(X)^2\right] - \mathbb{E}_P[f(X)]^2$ denote the mean and variance of the random variable
 $f(X)$, respectively. We now state the following transportation inequalities, which can be adapted
 from [BLM13] (Lemma 4.18).

 **Lemma 13.** *(Transportation inequalities) Assume $f$ is such that $S(f)$ and $\mathbb{V}_P[f]$ are finite. Then it*
 *holds*

$$\forall Q \ll P, \quad \mathbb{E}_Q[f] - \mathbb{E}_P[f] \leq \sqrt{2\mathbb{V}_P[f]\mathrm{KL}(Q,P)} + \frac{2S(f)}{3}\mathrm{KL}(Q,P)$$

$$\forall Q \ll P, \quad \mathbb{E}_P[f] - \mathbb{E}_Q[f] \leq \sqrt{2\mathbb{V}_P[f]\mathrm{KL}(Q,P)}$$

 ## D.2   Bregman divergence

For a Legendre function $F : \mathbb{R}^d \to \mathbb{R}$, the Bregman divergence between $\theta', \theta \in \mathbb{R}^d$ associated with
$F$ is defined as $B_F\left(\theta', \theta\right) := F\left(\theta'\right) - F(\theta) - \left(\theta' - \theta\right)^\top \nabla F(\theta)$. Now, for any fixed $\theta \in \mathbb{R}^d$, we
introduce the function

$$B_{F,\theta}(\lambda) := B_F(\theta + \lambda, \lambda) = F(\theta + \lambda) - F(\theta) - \lambda^\top \nabla F(\theta).$$

It then follows that $B_{F,\theta}$ is a convex function, and we define its dual as

$$B_{F,\theta}^\star(x) = \sup_{\lambda \in \mathbb{R}^d} \left(\lambda^\top x - B_{F,\theta}(\lambda)\right)$$

 We have for any $\theta, \theta' \in \mathbb{R}^d$:

$$B_F\left(\theta', \theta\right) = B_{F,\theta'}^\star\left(\nabla F(\theta) - \nabla F\left(\theta'\right)\right) \tag{27}$$

 To see this, we observe that

$$B_{F,\theta'}^\star\left(\nabla F(\theta) - \nabla F\left(\theta'\right)\right)$$
$$= \sup_{\lambda \in \mathbb{R}^d} \lambda^\top\left(\nabla F(\theta) - \nabla F\left(\theta'\right)\right) - \left[F\left(\theta' + \lambda\right) - F\left(\theta'\right) - \lambda^\top \nabla F\left(\theta'\right)\right]$$
$$= \sup_{\lambda \in \mathbb{R}^d} \lambda^\top \nabla F(\theta) - F\left(\theta' + \lambda\right) + F\left(\theta'\right).$$

Now an optimal $\lambda$ must satisfy $\nabla F(\theta) = \nabla F\left(\theta' + \lambda\right)$. One possible choice is $\lambda = \theta - \theta'$. Since, by
definition, $F$ is strictly convex, the supremum will indeed be attained at $\lambda = \theta - \theta'$. Plugin-in this
value, we obtain

$$B_{F,\theta'}^\star\left(\nabla F(\theta) - \nabla F\left(\theta'\right)\right) = \left(\theta - \theta'\right)^\top \nabla F(\theta) - F(\theta) + F\left(\theta'\right) = B_F\left(\theta', \theta\right).$$

 Note that (27) holds for any convex function $F$. Only difference is that, in this case, $B_F(\cdot, \cdot)$ will not
 correspond to the Bregman divergence.

 ## D.3   Properties of the bilinear exponential family

 In this section, we detail some useful results related to exponential families in our model.

 ### D.3.1   Derivatives

 **Lemma 14.** *(Gradients) We provide the derivatives of the log-partitions in closed form. As usual*
 *with exponential families, these are intimately linked to moments of the random variable. We have:*

$$\left(\nabla_i Z_{s,a}^{\mathbb{p}}\right)(\theta) = \mathbb{E}_{s,a}^\theta\left[\psi\left(s'\right)\right]^\top A_i \varphi(s,a).$$

 *And*

$$\left(\nabla_i Z_{s,a}^r\right)(\theta) = \mathbb{E}_{s,a}^\theta\left[r\right] B^\top A_i \varphi(s,a).$$

 *Proof.* We prove the lemma as follows

$$\left(\nabla_i Z^{\mathtt{p}}_{s,a}\right)(\theta) = \int_{\mathcal{S}} \psi\left(s'\right)^{\top} A_i \varphi(s,a) \frac{\exp\left(\sum_{i=1}^{d} \theta_i \psi\left(s'\right)^{\top} A_i \varphi(s,a)\right)}{\int_{\mathcal{S}} \exp\left(\sum_{i=1}^{d} \theta_t \psi\left(s'\right)^{\top} A_i \varphi(s,a)\right) ds'} ds'$$

$$= \mathbb{E}^{\theta}_{s,a}\left[\psi\left(s'\right)\right]^{\top} A_i \varphi(s,a)$$

$$\left(\nabla_i Z^{\mathtt{r}}_{s,a}\right)(\theta) = \int_{\mathcal{S}} rB^{\top} A_i \varphi(s,a) \frac{\exp\left(r \sum_{i=1}^{d} \theta_i B^{\top} A_i \varphi(s,a)\right)}{\int_{\mathcal{S}} \exp\left(r \sum_{i=1}^{d} \theta_i B^{\top} A_i \varphi(s,a)\right) dr} dr$$

$$= \mathbb{E}^{\theta}_{s,a}\left[r\right] B^{\top} A_i \varphi(s,a)$$

 $\qquad\qquad\square$

 **Lemma 15.** *(Hessians) The entries of the Hessians of the log partition functions are given by*

$$\left(\nabla^2_{i,j} Z^{\mathtt{p}}_{s,a}\right)(\theta) = \varphi(s,a)^{\top} A_i^{\top} \mathbb{C}^{\theta}_{s,a}\left[\psi\left(s'\right)\right] A_j \varphi(s,a),$$

 *where* $\mathbb{C}^{\theta}_{s,a}\left[\psi\left(s'\right)\right] \stackrel{\text{def}}{=} \mathbb{E}^{\theta}_{s,a}\left[\psi\left(s'\right)\psi\left(s'\right)^{\top}\right] - \mathbb{E}^{\theta}_{s,a}\left[\psi\left(s'\right)\right]\mathbb{E}^{\theta}_{s,a}\left[\psi\left(s'\right)^{\top}\right].$

 *Similarly,*

$$\left(\nabla^2_{i,j} Z^{\mathtt{r}}_{s,a}\right)(\theta) = \mathbb{V}\mathrm{ar}^{\theta}_{s,a}(r) \times \varphi(s,a)^{\top} A_i^{\top} BB^{\top} A_j \varphi(s,a),$$

 *where* $\mathbb{V}\mathrm{ar}^{\theta}_{s,a}(r) \stackrel{\text{def}}{=} \left(\mathbb{E}^{\theta}_{s,a}\left[r^2\right] - \mathbb{E}^{\theta}_{s,a}\left[r\right]^2\right)$ *is the variance of the reward under* $\theta$.

 *Proof.* We prove these formulas by differentiating under the integral sign.

$$\left(\nabla^2_{i,j} Z^{\mathtt{p}}_{s,a}\right)(\theta) = \int_{\mathcal{S}} \psi\left(s'\right)^{\top} A_i \varphi(s,a) \psi\left(s'\right)^{\top} A_j \varphi(s,a) \frac{\exp\left(\sum_{i=1}^{d} \theta_i \psi\left(s'\right)^{\top} A_i \varphi(s,a)\right)}{\int_{\mathcal{S}} \exp\left(\sum_{i=1}^{d} \theta_i \psi\left(s'\right)^{\top} A_i \varphi(s,a)\right) ds'} ds'$$

$$- \int_{\mathcal{S}} \psi\left(s'\right)^{\top} A_i \varphi(s,a) \frac{\exp\left(\sum_{i=1}^{d} \theta_i \psi\left(s'\right)^{\top} A_i \varphi(s,a)\right)}{\int_{\mathcal{S}} \exp\left(\sum_{i=1}^{d} \theta_i \psi\left(s'\right)^{\top} A_i \varphi(s,a)\right) ds'} ds' \left(\nabla_j Z_{s,a}\right)(\theta)$$

$$= \mathbb{E}^{\theta}_{s,a}\left[\psi\left(s'\right)^{\top} A_i \varphi(s,a) \psi\left(s'\right)^{\top} A_j \varphi(s,a)\right]$$

$$- \mathbb{E}^{\theta}_{s,a}\left[\psi\left(s'\right)^{\top} A_i \varphi(s,a)\right] \mathbb{E}^{\theta}_{s,a}\left[\psi\left(s'\right)^{\top} A_j \varphi(s,a)\right]$$

$$= \varphi(s,a)^{\top} A_i^{\top} \left(\mathbb{E}^{\theta}_{s,a}\left[\psi\left(s'\right)\psi\left(s'\right)^{\top}\right] - \mathbb{E}^{\theta}_{s,a}\left[\psi\left(s'\right)\right]\mathbb{E}^{\theta}_{s,a}\left[\psi\left(s'\right)^{\top}\right]\right) A_j \varphi(s,a)$$

$$= \varphi(s,a)^{\top} A_i^{\top} \mathbb{C}^{\theta}_{s,a}\left[\psi\left(s'\right)\right] A_j \varphi(s,a),$$

where we introduce in the last line the $p \times p$ covariance matrix given by

$$\mathbb{C}^{\theta}_{s,a}\left[\psi\left(s'\right)\right] = \mathbb{E}^{\theta}_{s,a}\left[\psi\left(s'\right)\psi\left(s'\right)^{\top}\right] - \mathbb{E}^{\theta}_{s,a}\left[\psi\left(s'\right)\right]\mathbb{E}^{\theta}_{s,a}\left[\psi\left(s'\right)^{\top}\right]$$

 The proof of the form of the Hessian for the reward partition function follows the same steps as
 above. $\qquad\square$

 **Lemma 16.** *(KL Divergences) For any two* $\theta, \theta'$ *and for some pair* $(s,a)$,

$$\exists \tilde{\theta} \in [\theta, \theta']_{\infty}, \quad \mathrm{KL}\left(P^{\mathtt{p}}_{\theta}(\cdot \mid s,a), P^{\mathtt{p}}_{\theta'}(\cdot \mid s,a)\right) = \frac{1}{2}\left(\theta - \theta'\right)^{\top}\left(\nabla^2 Z^{\mathtt{p}}_{s,a}\right)(\tilde{\theta})\left(\theta - \theta'\right),$$

 *where* $[\theta, \theta']_{\infty}$ *denotes the* $d$-*dimensional hypercube joining* $\theta$ *to* $\theta'$.

 *Similarly*

$$\exists \tilde{\theta} \in [\theta, \theta']_{\infty}, \quad \mathrm{KL}\left(P^{\mathtt{r}}_{\theta}(\cdot \mid s,a), P^{\mathtt{r}}_{\theta'}(\cdot \mid s,a)\right) = \frac{1}{2}\left(\theta - \theta'\right)^{\top}\left(\nabla^2 Z^{\mathtt{r}}_{s,a}\right)(\tilde{\theta})\left(\theta - \theta'\right).$$

*Proof.* We start by writing:

$$\log\left(\frac{P^{\mathsf{p}}_{\theta}(s' \mid s,a)}{P^{\mathsf{p}}_{\theta'}(s' \mid s,a)}\right) = \sum_{i=1}^{d} (\theta_i - \theta'_i)\,\psi(s')^{\top} A_i \varphi(s,a) - Z^{\mathsf{p}}_{s,a}(\theta) + Z^{\mathsf{p}}_{s,a}(\theta'),$$

then

$$\mathrm{KL}\left(P^{\mathsf{p}}_{\theta}(\cdot \mid s,a), P^{\mathsf{p}}_{\theta'}(\cdot \mid s,a)\right) = \sum_{i=1}^{d} (\theta_i - \theta'_i)\,\mathbb{E}^{\theta}_{s,a}[\psi(s')]^{\top} A_i \varphi(s,a) - Z^{\mathsf{p}}_{s,a}(\theta) + Z^{\mathsf{p}}_{s,a}(\theta')$$

$$= \frac{1}{2} (\theta - \theta')^{\top} (\nabla^2 Z^{\mathsf{p}}_{s,a})(\tilde{\theta})(\theta - \theta'),$$

where in the last line, we used, by a Taylor expansion, that $Z_{s,a}(\theta') = Z_{s,a}(\theta) + (\nabla Z_{s,a}(\theta))^{\top}(\theta' - \theta) + \frac{1}{2}(\theta - \theta')^{\top}\left(\nabla^2 Z_{s,a}(\tilde{\theta})\right)(\theta - \theta')$ for some $\tilde{\theta} \in [\theta, \theta']_{\infty}$.

The proof of the form of the KL divergence for the reward follows the same steps as above. □

### D.3.2 A transportation lemma for rewards

**Lemma 17.** *We provide a closed-form formula for the difference of expected rewards under two distinct parameters:*

$$\exists \theta_3 \in [\theta_1, \theta_2], \qquad \mathbb{E}^{\theta_1}_{s,a}[r] = \mathbb{E}^{\theta_2}_{s,a}[r] + \frac{\mathbb{V}\mathrm{ar}^{\theta_3}_{s,a}(r)}{2} B^{\top} M_{\theta_1 - \theta_2}\varphi(s,a)$$

*Proof.* Let's recall the gradient of the reward log partition function:

$$(\nabla_i Z^{\mathsf{r}}_{s,a})(\theta^{\mathsf{r}}) = \mathbb{E}^{\theta^{\mathsf{r}}}_{s,a}[r]\,B^{\top} A_i \varphi(s,a)$$

then for all $\theta^{\mathsf{r}\prime}$ we have:

$$\mathbb{E}^{\theta^{\mathsf{r}}}_{s,a}[r] = \frac{1}{B^{\top} M_{\theta^{\mathsf{r}\prime}}\varphi(s,a)}\nabla_i Z^{\mathsf{r}}_{s,a}(\theta^{\mathsf{r}})^{\top}\theta^{\mathsf{r}\prime}$$

Let $\theta_1, \theta_2 \in \mathbb{R}^d$, using Taylor-Cauchy's formula there exists $\theta_3 \in [\theta_1, \theta_2]$ such that:

$$\mathbb{E}^{\theta_1}_{s,a}[r] = \mathbb{E}^{\theta_2}_{s,a}[r] + \frac{1}{2B^{\top} M_{\theta^{\mathsf{r}\prime}}\varphi(s,a)}(\theta_1 - \theta_2)^{\top}\nabla^2 Z^{\mathsf{r}}_{s,a}(\theta_3)^{\top}\theta^{\mathsf{r}\prime}$$

We know that $(\nabla^2_{i,j} Z^{\mathsf{r}}_{s,a})(\theta) = \mathbb{V}\mathrm{ar}^{\theta}_{s,a}(r) \times \varphi(s,a)^{\top} A_i^{\top} BB^{\top} A_j \varphi(s,a)$, choosing $\theta^{\mathsf{r}\prime} = \theta_1 - \theta_2$ we find:

$$\mathbb{E}^{\theta_1}_{s,a}[r] = \mathbb{E}^{\theta_2}_{s,a}[r] + \frac{\mathbb{V}\mathrm{ar}^{\theta_3}_{s,a}(r)}{2} B^{\top} M_{\theta_1 - \theta_2}\varphi(s,a).$$

□

## D.4 Elliptical potentials and elliptical lemma

### D.4.1 Elliptical lemma

Here we show a lemma that is popular for regret control in linear MDPs and linear Bandits.

First, consider the notations: $G_{s,a} := (\varphi(s,a)^{\top} A_i^{\top} A_j \varphi(s,a))_{1 \leq i,j \leq d}$, $\bar{G}^{\mathsf{e}}_n \equiv \bar{G}^{\mathsf{e}}_{(k-1)H} := G_n + (\alpha^{\mathsf{e}})^{-1}\eta A$, and $G_n \equiv G_{(k-1)H} := \sum_{\tau=1}^{k-1}\sum_{h=1}^{H} G_{s^{\tau}_s, a^{\tau}_h}$. Where $\mathsf{e}$ represents either $\mathsf{r}$ or $\mathsf{p}$, we omit the superscript $\mathsf{e}$ w.l.o.g in the rest of this section.

**Lemma 18.** *(Elliptical lemma and variant for bounded potentials) Let $c \in \mathbb{R}^+$, we can bound the sum of feature norms as follows*

$$\sum_{t=1}^{T}\min\{c, \sum_{h=1}^{H}\left\|\bar{G}^{-1/2}_n G_{s,a}\bar{G}^{-1/2}_n\right\|\} \leq \frac{c}{\log(1+c)}d\log\left(1 + \alpha\eta^{-1}B_{\varphi,\mathbb{A}}n\right).$$

*where $B_{\varphi,\mathbb{A}} := \sup_{s,a}\left\|\mathbb{A}^{-1}G_{s,a}\right\|$.*

*Further, we have*

$$\sum_{t=1}^{T}\sum_{h=1}^{H}\left\|\bar{G}^{-1/2}_n G_{s,a}\bar{G}^{-1/2}_n\right\| \leq 2d\log\left(1 + \alpha\eta^{-1}B_{\varphi,\mathbb{A}}n\right) + \frac{3dH}{\log(2)}\log\left(1 + \frac{\alpha\|A\|^2_2 B^2_{\varphi,\mathbb{A}}}{\eta\log(2)}\right)$$

*Proof.* First we have

$$\|\bar{G}_n^{-1/2}G_{s,a}\bar{G}_n^{-1/2}\| = \sqrt{\operatorname{tr}(\bar{G}_n^{-1/2}G_{s,a}\bar{G}_n^{-1/2}\bar{G}_n^{-1/2}G_{s,a}\bar{G}_n^{-1/2})}$$
$$\leq \operatorname{tr}(\bar{G}_n^{-1/2}G_{s,a}\bar{G}_n^{-1/2}) = \operatorname{tr}(\bar{G}_n^{-1}G_{s,a}) = \operatorname{tr}(\boldsymbol{a}_h^\top \bar{G}_n^{-1}\boldsymbol{a}_h)$$

the last line is because $G_{s,a} = \boldsymbol{a}_h\boldsymbol{a}_h^\top$, where $\boldsymbol{a}_h = (A_i\varphi(s_h,a_h))_{i\in[d]}$.

**First result.** Consider $h \in [H]$, denote $(\lambda_{h,i})i \in [d]$ the eigenvalues of $\boldsymbol{a}_h^\top \bar{G}_n^{-1}\boldsymbol{a}_h$. $\bar{G}_n$ is positive definite hence $\lambda_{h,i} > 0, \forall h, i$, then

$$\min\{c, \sum_{h=1}^H \operatorname{tr}(\boldsymbol{a}_h^\top \bar{G}_n^{-1}\boldsymbol{a}_h)\} = \min\{c, \sum_{h=1}^H \sum_{i=1}^d \lambda_{h,i}\}$$

$$\leq \frac{c}{\log(1+c)} \sum_{h=1}^H \sum_{i=1}^d \log(1+\lambda_{h,i}) \qquad \text{(log is concave)}$$

$$\leq \frac{c}{\log(1+c)} \sum_{h=1}^H \log(\prod_{i=1}^d 1+\lambda_{h,i}) = \frac{c}{\log(1+c)} \sum_{h=1}^H \log\det(I + \boldsymbol{a}_h^\top \bar{G}_n^{-1}\boldsymbol{a}_h)$$

$$\leq \frac{c}{\log(1+c)} \log\left(\frac{\det(\bar{G}_n + \sum_{h=1}^H G_{s_h,a_h})}{\det(\bar{G}_n)}\right)$$

where the last line follows from the matrix determinant lemma:
$$\det\left(\bar{G}_n + \boldsymbol{a}_h\boldsymbol{a}_h^\top\right) = \det(I + \boldsymbol{a}_h^\top \bar{G}_n^{-1}\boldsymbol{a}_h)\det(\bar{G}_n)$$

Therefore:
$$\sum_{t=1}^T \min\{c, \sum_{h=1}^H \left\|\bar{G}_n^{-1}G_{s_h^t,a_h^t}\right\|\} \leq \frac{c}{\log(1+c)} \sum_{t=1}^T \log\frac{\det\left(\bar{G}_{n+H}\right)}{\det\left(\bar{G}_n\right)},$$

We can now control the R.H.S. of the above equation, as

$$\sum_{t=1}^T \log\frac{\det\left(\bar{G}_{n+H}\right)}{\det\left(\bar{G}_n\right)} = \sum_{t=1}^T \log\frac{\det\left(\bar{G}_{tH}\right)}{\det\left(\bar{G}_{(t-1)H}\right)} = \log\frac{\det\left(\bar{G}_{TH}\right)}{\det\left(\bar{G}_0\right)}$$

$$= \log\frac{\det\left(\bar{G}_N\right)}{\det\left((\alpha^{\mathrm{p}})^{-1}\eta\mathbb{A}\right)} = \log\det\left(I + \alpha\eta^{-1}\,\mathrm{A}^{-1}G_N\right)$$

$$\leq d\log\left(1 + \frac{\alpha^{\mathrm{p}}\eta^{-1}}{d}\operatorname{tr}\left(\mathbb{A}^{-1}G_n\right)\right) \qquad \text{(Trace-determinant (or AM-GM) inequality)}$$

$$\leq d\log\left(1 + \alpha^{\mathrm{p}}\eta^{-1}B_{\varphi,\mathbb{A}}n\right)$$

This concludes the proof of the first result.

**Second result.** First, we have $\sup_{s,a}\|G_{s,a}\|_2 \leq \|A\|_2 B_{\varphi,\mathbb{A}}$.

Fix an episode $k \in [K], n = (k-1)H$, using Lemma 19, we know that the number of times $h \in [h]$ such that $\left\|\bar{G}_n^{-1}G_{s_h,a_h}\right\| \geq 1$ is smaller than $\frac{3d}{\log(2)}\log\left(1 + \frac{\alpha(\|A\|_2 B_{\varphi,\mathbb{A}})^2}{\eta\log(2)}\right)$. Let us call $\mathcal{T}_k := \{h \in [H]\left\|\bar{G}_{(k-1)h}^{-1}G_{s_h,a_h}\right\| \leq 1\}$, then

$$\sum_{t=1}^T \sum_{h=1}^H \left\|\bar{G}_n^{-1}G_{s_h^t,a_h^t}\right\| \leq \frac{3d}{\log(2)}\log\left(1 + \frac{\alpha\|A\|_2^2 B_{\varphi,\mathbb{A}}^2}{\eta\log(2)}\right) + \sum_{h\in\mathcal{T}_k}\min\{1, \left\|\bar{G}_n^{-1}G_{s_h^t,a_h^t}\right\|\}$$

the sum of the right hand side is similar to the first result. Although the sum is not contiguous, the previous bound holds since if $h_1 < h_2, \det(\bar{G}_{n+h_1}) \leq \det(\bar{G}_{n+h_2})$, this concludes the proof. $\square$

**Remark 7.** *We can also write from the lemma in terms of* $\|(A_i\varphi(\tilde{s}_h,\pi(\tilde{s}_h)))_{1\leq i\leq d}\|_{(\bar{G}_k^r)^{-1}}$ *by skipping the norm upper bound at the beginning of the proof:*

$$\sum_{t=1}^T \min\{c, \sum_{h=1}^H \|(A_i\varphi(\tilde{s}_h,\pi(\tilde{s}_h)))_{1\leq i\leq d}\|_{(\bar{G}_k^r)^{-1}}\} \leq \frac{c}{\log(1+c)}d\log\left(1 + \alpha\eta^{-1}B_{\varphi,\mathbb{A}}n\right).$$

*and*

$$\sum_{t=1}^{T}\sum_{h=1}^{H}\|(A_i\varphi(\tilde{s}_h,\pi(\tilde{s}_h)))_{1\leq i\leq d}\|_{(\bar{G}_k^r)^{-1}} \leq 2d\log\left(1+\alpha\eta^{-1}B_{\varphi,\mathbb{A}}n\right)$$
$$+\frac{3dH}{\log(2)}\log\left(1+\frac{\alpha\|A\|_2^2 B_{\varphi,\mathbb{A}}^2}{\eta\log(2)}\right)$$

### D.4.2  Elliptical potentials: finite number of large feature norms (contribution)

**Lemma 19.** *(Worst case elliptical potentials, adaptation of Exercise 19.3 [LS20] for matrices) Let* $V_0 = \lambda I$ *and* $a_1,\ldots,a_n \in \mathbb{R}^{d\times p}$ *be a sequence of matrices with* $\|a_t\|_2 \leq L$ *for all* $t \in [n]$. *Let* $V_t = V_0 + \sum_{s=1}^{t} a_s a_s^\top$, *then*

$$\left|\{t\in\mathbb{N}^*,\|a_t\|_{V_{t-1}^{-1}}\geq 1\}\right| \leq \frac{3d}{\log(2)}\log\left(1+\frac{L^2}{\lambda\log(2)}\right)$$

*Proof.* Let $\mathcal{T}$ be the set of rounds $t$ when $\|a_t\|_{V_{t-1}^{-1}} \geq 1$ and $G_t = V_0 + \sum_{s=1}^{t}\mathbb{I}_{\mathcal{T}}(s)a_s a_s^\top$. Then

$$\left(\frac{d\lambda+|\mathcal{T}|L^2}{d}\right)^d \geq \left(\frac{\text{trace}\,(G_n)}{d}\right)^d$$
$$\geq \det(G_n) \qquad\qquad \text{(Trace-determinant inequality)}$$
$$= \det(V_0)\prod_{t\in T}\left(1+\|a_t\|_{G_{t-1}^{-1}}^2\right)$$
$$\geq \det(V_0)\prod_{t\in T}\left(1+\|a_t\|_{V_{t-1}^{-1}}^2\right)$$
$$\geq \lambda^d 2^{|\mathcal{T}|}$$

where the third line follows from the matrix determinant lemma:

$$\det\left(\bar{G}_n + \boldsymbol{a}_h\boldsymbol{a}_h^\top\right) = \det(I + \boldsymbol{a}_h^\top\bar{G}_n^{-1}\boldsymbol{a}_h)\det(\bar{G}_n).$$

Rearranging and taking the logarithm shows that

$$|\mathcal{T}| \leq \frac{d}{\log(2)}\log\left(1+\frac{|\mathcal{T}|L^2}{d\lambda}\right)$$

Abbreviate $x = d/\log(2)$ and $y = L^2/d\lambda$, which are both positive. Then

$$x\log(1+y(3x\log(1+xy))) \leq x\log\left(1+3x^2y^2\right) \leq x\log(1+xy)^3 = 3x\log(1+xy).$$

Since $z - x\log(1+yz)$ is decreasing for $z \geq 3x\log(1+xy)$ it follows that

$$|\mathcal{T}| \leq 3x\log(1+xy) = \frac{3d}{\log(2)}\log\left(1+\frac{L^2}{\lambda\log(2)}\right).$$

$\square$

## E  Tractable planning with random Fourier transform

**A Primer on random Fourier transforms.** We start by defining the Random Fourier Transform and its most relevant property. Let us consider the transition model of Equation (1), we have

$$\mathbb{P}(s' \mid s,a,\theta) = \exp\left(\psi(s')M_\theta\varphi(s,a) - Z_\theta(s,a)\right) = \mathbb{E}_{p(w,b)}\left[f\left(\psi(s'),w,b\right)f\left(M_\theta\varphi(s,a),w,b\right)\right],$$

where $f(x,w,b) = \sqrt{2}\cos(w^\top x + b)$ are the random Fourier bases. $p(w,b) = \mathcal{N}(0,\sigma^{-2}I) \times \mathcal{U}([0,2\pi])$, such that $\mathcal{N}$ is the Gaussian distribution, $\mathcal{U}$ is the Uniform distribution, and $p(w,b)$ is a coupling among them.

Notice that this provides an alternative approach to decompose the transition kernel and obtain linearity of the value function. Moreover, since $\forall x, w \in \mathbb{R}^d, b \in \mathbb{R}, |f(x, w, b)| \leq \sqrt{2}$, we can use Hoeffding's inequality to prove that a Monte-Carlo approximation of $\mathbb{P}(s' \mid s, a, \theta)$ using $N$ sample pairs of $(w, b)$ guarantees an error smaller than $\epsilon$ with probability at least $1 - 2\exp(-N\epsilon^2/4)$. [RR07] proves a stronger result: it provides an algorithm approximating the Gaussian kernel for which the following uniform convergence bound holds.

**Lemma 20.** *Let $\mathcal{M}$ be a compact subset of $\mathcal{R}^p$ with diameter $\mathrm{diam}(\mathcal{M})$. Then, using the explicit mapping $\mathbf{z}$ defined in Algorithm 1 in [RR07] with $N$ samples, we have*

$$\Pr\left[\sup_{x,y\in\mathcal{M}} |\mathbf{z}(\mathbf{x})'\mathbf{z}(\mathbf{y}) - k(\mathbf{y},\mathbf{x})| \geq \epsilon\right] \leq 2^8 \left(\frac{\sigma_p\,\mathrm{diam}(\mathcal{M})}{\epsilon}\right)^2 \exp\left(-\frac{N\epsilon^2}{4(p+2)}\right)$$

*where $\sigma_p^2 \equiv E_p[\omega'\omega]$ is the second moment of the Fourier transform of $k$.*

Further, it implies that if $N = \Omega\left(\frac{p}{\epsilon^2} \log \frac{\sigma_p\,\mathrm{diam}(\mathcal{M})}{\epsilon}\right)$, then $\sup_{x,y\in\mathcal{M}} |\mathbf{z}(\mathbf{x})'\mathbf{z}(\mathbf{y}) - k(\mathbf{y},\mathbf{x})| \leq \epsilon$ with constant probability.

**Application to planning in** `BEF-RLSVI`**.** Since our regret analysis is done under the high probability event of bounded estimation parameters, we know that the spaces of $\psi(s')$ and $M_\theta\varphi(s,a)$ are bounded and the diameter depends on the dimensions. We abstain from explicating the exact diameter as it only influences the number of samples logarithmically. Using $N \approx p/\epsilon^2$ samples, we can construct a uniform $\epsilon$-approximation of $\mathbb{P}(s' \mid s, a, \theta)$.

Let's call $\hat{V}_h$ the estimated value function using Algorithm 3 with the above approximation of transition. Here, we elucidate the span of this estimation of value function. First we have:

$$\hat{V}_H^\pi - V_H^\pi = \int_{s'} (\hat{P} - P)(s' \mid s, a) r(s', \pi(s'))\,\mathrm{d}s' \leq \epsilon dH^{3/2}$$

Here, we use the facts that $\mathbb{S}\left(V_{\hat{\theta}, \tilde{\theta}^\times, h}\right) \leq dH^{3/2}$ (*cf.* Section B.2) and the error in approximating $P$ is bounded by $\epsilon$, i.e. $\sup_{s',s,a} |(\hat{P} - P)(s'|s,a)| \leq \epsilon$.

Assume that at step $h+1$, we have $\hat{V}_{h+1}^\pi - V_{h+1}^\pi \leq \sum_{j=1}^{h+1} \epsilon^j \alpha_{h+1,j}$. Then, we obtain

$$\hat{V}_h^\pi - V_h^\pi \leq \int_{s'} (\hat{P} - P)(s' \mid s, a)\hat{V}_{h+1}^\pi(s')\,\mathrm{d}s' + \int_{s'} P(s' \mid s, a)(\hat{V}_{h+1}^\pi - V_{h+1}^\pi)(s')\,\mathrm{d}s'$$

$$= \int_{s'} (\hat{P} - P)(s' \mid s, a)(V_{h+1}^\pi + \hat{V}_{h+1}^\pi - V_{h+1}^\pi)\,\mathrm{d}s' + \int_{s'} P(s' \mid s, a)(\hat{V}_{h+1}^\pi - V_{h+1}^\pi)(s')\,\mathrm{d}s'$$

$$\leq \epsilon(dH^{3/2} + \sum_{j=1}^{h+1} \epsilon^j \alpha_{h+1,j}) + \sum_{j=1}^{h+1} \epsilon^j \alpha_{h+1,j}$$

$$\leq \epsilon(dH^{3/2} + \alpha_{h+1,1}) + \sum_{j=2}^{h+1} \epsilon^j (\alpha_{h+1,j-1} + \alpha_{h+1,j}) + \epsilon^{h+2} \alpha_{h+1,h+1}$$

Using the fact that $\alpha_{1,1} = dH^{3/2}$ and with a proper induction, we find that:

$$\hat{V}_1^\pi - V_1^\pi \leq \epsilon dH^{5/2} \frac{1 - \epsilon^{H-h}}{1 - \epsilon} \underset{H\to\infty}{\leq} \epsilon dH^{5/2}$$

This concludes the proof of the arguments provided in § Planning of Section 4. This means that the extra regret due to planning with the approximation by RFT features is of order $\mathcal{O}(\epsilon dH^{5/2}K)$. By choosing an $\epsilon$ of order $1/(H\sqrt{K})$, we deduce that approximating the probability kernel with $\mathcal{O}(pH^2K)$ samples induces a tractable planning procedure without harming the regret.

**Remark 8.** *The reader might be tempted to combine the finite approximation using RFT with algorithms from the linear reinforcement learning literature [JYWJ20]. However, note that the dimensionality of the linear space induced by RFT is polynomial in $H$ and $K$. Consequently, applying algorithms designed with the assumption of linear value function would incur a linear regret.*