# OpenReview forum: "Bilinear Exponential Family of MDPs: Frequentist Regret Bound with Tractable Exploration $\&$ Planning"
_NeurIPS.cc/2022/Conference — NeurIPS 2022 Submitted_

### Official Review · Reviewer_jyJc · 2022-07-02

**Rating:** 6
**Confidence:** 3
**Soundness:** 3 good
**Presentation:** 3 good
**Contribution:** 3 good

**Summary:**

This paper studies model-based reinforcement learning for episodic Markov decision processes whose rewards and transitions are parametrized by bilinear exponential families with features of state and action. To balance the exploration-exploitation trade-off, the author suggested a randomized algorithm that injects a calibrated Gaussian noise in the parameter of rewards. The proposed algorithm achieves $\tilde{\mathcal{O}}(\sqrt{d^3 H^3 K})$ regret bound with high probability.


**Questions:**

I would appreciate it if the author could tell me their opinion on the aformentioned 'Weakness' part.

Some more questions are as follows:

Line 93: I think $\theta$ denotes $(\theta^p, \theta^r)$. It would be better to mention that before referring to the optimal policy for clarity.

Line 191: For the $Q$ function that the agent calculates using the estimated parameter, if $Q^{\pi}_{\hat{\theta}^p, \tilde{\theta}^r, h}$ in line 191 is the same $Q\_{\tilde{\theta}, h}$ as the calculated in Algorithm 2, shouldn't it be correct that there should be no $\pi$ symbol in $Q$? Also, I think it should be fixed in the regret decomposition (eq. 10).

Line 230 (Stochastic optimism): I am a little confused because of the $\pi$ notation in the value function calculated with the estimated parameter (e.g., $V^{\pi}\_{\hat{\theta}^p, \tilde{\theta}^r, 1})$, but I am curious how the $Q^*_{\hat{\theta}^p, \tilde{\theta}^r, 1}$ appears in the first inequality. Also, I am not sure how the second inequality decomposes?

Most of explanations were clear and well written, but with appearance of a lot of notations, it would be very helpful to understand this paper if the author mentions why the inequality holds in the appendix.

**Limitations:**

I think there are no issues related to social impact. However, although this paper is highly related to theoretical part, considering that many recently published theoretical papers about model-based RL also present numerical experiments, I think it would be better if there was an experimental result in this paper.

**Strengths And Weaknesses:**

<Strengths>

1. In this paper, the author not only presented a provable algorithm for a model-based algorithm whose transitions and rewards are parametrized by a bilinear exponential family that extends the linear transition assumption but also improved the regret by $\sqrt{H}$ compared to the upper-confidence based algorithm in a similar setting.

2. Unlike the other model-based algorithms that assume the reward information is known, this paper solves the problem under the assumption that neither transition model nor reward information is known. In addition, it is interesting that the analysis is carried out by dividing the “Good rounds” and “Bad rounds” according to the weighted norm of the state-action feature without using the handcrafted clipping value function.

3. Based on the concentration results of the upper-confidence based algorithm in a similar problem setting, the author presented a randomized algorithm which is known to be practical but difficult to analyze with a frequentist regret bound guarantee for the first time.

<Weakness>

1.  Although the author dealt with the setting where the model is parametrized by the bilinear exponential family, but it seems to solve the problem by using linearity in RKHS. This seems to rely on the information on the RKHS (e.g., RBF kernel $k(x,y)$) corresponding to the transition probability that the agent estimates. If not, what is the difference between the problem in this paper and the linear MDP method in ([JYWJ20], [ZBB+20], Yang & Wang;2020)  ?

c.f) Yang & Wang, 2020: Reinforcement Learning in Feature Space: Matrix Bandit, Kernels, and Regret Bound

2. As the author said, the most existing works on model-based RL usually assume that information about reward is given. The reason is that it is more difficult to estimate the transition model than the reward, and in the end, the part for reward estimation in regret is not the leading term. This can be seen in the main theorem presented in the paper. Therefore, it is questionable whether learning without information on reward can be said to be a special contribution.

---

> ### Author Response · Authors · 2022-08-02
> **Discussion with reviewer jyJc**
>
> Thank you for your time, careful review, and for the kind words about the soundness of our results and the novelty of our analysis.
>
> **Comparison with linear MDPs:** We emphasize that our observation of linearity does not simply reduce the problem to a linear MDP. This is because our linearity is in an infinite-dimensional RKHS and also we don't have linearity in the parameter. Therefore, classical linear RL algorithms (JYWJ20, ZBB+20, Yang and Wang;2020) cannot be used for this setting. Moreover, while we also solve planning using linearity, the latter is merely a direct consequence of the -realistic- considered model. In contrast, in the linear RL literature, linearity is a strong assumption especially in a finite dimensional space. Please refer to the general comment for a detailed discussion.
>
> **Reward estimation:** We agree that estimating the transition model is harder than rewards and we also agree that compared to [CGM21], estimating the reward is only a minor improvement. However, the base algorithm, RLSVI, was proposed in [ZBB+20] assuming a known transition, so in regard to the linear literature our work is novel and our contribution is significant.
>
> **Notations:** We agree that our notation is confusing at times, to clarify, $\pi$ should be indexed by $k$ and it is the BEF-RLSVI's policy at step $k$, everything denoted by $\star$ will be changed to $\pi^\star$ and just means that it's the value function under that set of parameters while acting with the optimal policy.
>
> **Stochastic optimism sketch:** We would like to clarify the reasoning in line 230, the first inequality comes from two facts: **1)** If the MDP's parameters are $\hat{\theta}^p,\tilde{\theta}^r$ then $\pi_k$ is the optimal policy and we obtain: $V_{\hat{\theta}^p,\tilde{\theta}^r,1}^{\pi_k} (s_1) = Q_{\hat{\theta}^p,\tilde{\theta}^r,1}^{\pi_k} (s_1, \pi_k(s_1)) \ge Q_{\hat{\theta}^p,\tilde{\theta}^r,1}^{\pi^\star} (s_1, \pi^\star (s_1))$. And **2)** By definition we have: $V_{\theta^p,\theta^r,1}^{\pi^\star} (s_1) = Q_{\theta^p,\theta^r,1}^{\pi^\star}(s_1,\pi^\star(s_1))$. The decomposition in below line 230 can be understood using: $Q_{\hat{\theta}^p,\tilde{\theta}^r,1}^{\pi^\star} (s_1, \pi^\star (s_1)) - Q_{\theta^p,\theta^r,1}^{\pi^\star}(s_1,\pi^\star(s_1)) = V_{\hat{\theta}^p,\tilde{\theta}^r,1}^{\pi^\star} (s_1) - V_{\theta^p,\theta^r,1}^{\pi^\star} (s_1)$, the other terms just telescope. We will add more comments about this in the revised version so that the decomposition is clear for the reader.
>
> **Minor comments:** We confirm that $\theta$ denotes $(\theta^p,\theta^r)$, the $\pi$ policy is indeed the one derived in algorithm 2, we insist however about our choice of indexing the value function with the policy because of its utility in the proof of the stochastic optimism for example. \textit{i.e.} we sometimes need to consider the estimated state-action value function but with the optimal policy instead.
>
> **Experiments:** We agree about the advantages of empirical evaluation and would like to confirm that it is indeed a direction we intend to explore in the future. However, we focus here on the theoretical challenges of the considered setting and we decided to put emphasis on such tools, results and explanations rather than extensive experiments. Consequently, this work is fairly extensive and slightly notation heavy for the NeurIPS page limit so we think it is best to devise a longer version in the future to include experiments.
>
> We wish to thank you again for your time, careful review, and for acknowledging the strength of our contribution. We hope that our response clarified some confusions and would appreciate if you can adjust the score accordingly.

---

> > ### Comment · Reviewer_jyJc · 2022-08-08
> > **Acknowledgement of Rebuttal**
> >
> > Thanks for the author's detailed response.
> > I have no further questions.

---

### Official Review · Reviewer_rPG8 · 2022-07-11

**Rating:** 6
**Confidence:** 3
**Soundness:** 3 good
**Presentation:** 3 good
**Contribution:** 3 good

**Summary:**

This paper studies episodic reinforcement learning where the reward and transition probability functions belong to parametric bilinear exponential families. A randomized least square value iteration algorithm is proposed to perform tractable exploration and planning for bilinear exponential family of MDPs. Novel analysis techniques are introduced to show that the regret of the proposed algorithm is bounded by $\widetilde{\mathcal{O}}(\sqrt{d^3H^3K})$.

**Questions:**

Some comments and questions are as follows:

1. The notation for matrix $\mathbb{A}$ in Algorithm 1 is used before its definition. It's better to move earlier the definition of $\mathbb{A}$.
2. Compared with the original definition of the bilinear exponential family model in [CGM21], the factor $h(s',s,a)$ is omitted. Is this critical for the observation of linearity of transitions in Eq.(4)?
3. It is said in line 41-43 that linear MDP is a special case of the bilinear exponential family model. Could the authors elaborate on this?
4. In line 11 of Algorithm 1, solving $\hat\theta^p(k)$ and $\hat\theta^r(k)$ involves integral approximations as remarked in line 176-177. Can this be implemented efficiently? More details on this should be provided.
5. The $({\hat\theta}^p, \tilde\theta^r)$ in line 6 of Algorithm 1 should be indexed with $k$? Similarly, in the regret decomposition in Eq.(9) and the corresponding analysis in Section 5 and Appendix B, what is $\hat \theta^p$? Is it an arbitrary estimator or some estimate constructed during running Algorithm 1?
6. Following the previous question, it is not clear why $\zeta_{hk}$ in line 508 in the appendix is a martingale sequence. Does $V_{\hat\theta^p,\theta^r,h+1}$ depend on previous data? Details about this $\hat\theta^p$ need to be specified.

---
Minor issues:

1. In line 22, '$t = 1, \ldots, H$', missing a comma.
2. It seems that the definition of $\mathbb{E}_{s,a}^{\tilde\theta^r}[r]$ is missing in the main context.
3. In line 40, a few extra '-'.
4. The inner product in Eq.(5) and the equation under line 191 should be $\langle\cdot, \cdot \rangle_{\mathcal{H}}$?
5. In line 140, '$[p, r]$' should be {$p, r$}?

**Limitations:**

The authors have adequately addressed the limitations and potential negative societal impact of their work.

**Strengths And Weaknesses:**

This paper is well-written and easy to follow. The authors have clearly summarized their contributions and discussed the improvement in their method over existing works. The theoretical analysis seems to be novel and non-trivial. However, the bilinear exponential family model defined in this paper seems to be more restrictive than the original definition in [CGM21]. Also, some notations are confusing. See more details below.

---

> ### Author Response · Authors · 2022-08-02
> **Discussion with reviewer rPG8**
>
> We would like to thank you for your time, thorough feedback, and for the kind words about the clarity of the contribution and novelty of the analysis.
>
> **Model expressivity:** Regarding the difference with the original model of [CGM21], thank you for catching this honest mistake, it is indeed less generic, we followed the definition of [LLS+21] that we thought was the same as [CGM21]. Note however that the model we consider still recovers the original model if the base measure $h$ verifies: $\exists h_1, h_2: h(s',s,a) \propto h_1(s') h_2(s,a)$, this is true because we can keep linearity in Eq 4 simply by multiplying $\phi^{\mathrm{p}}(s, a)$ by $h_2(s,a)$ and $\mu^{\mathrm{P}}(s^{\prime})$ by $h_1(s^{\prime})$. We do not believe that this is restrictive as it is verified by all the examples provided in [CGM21] (see Section 4 therein). It also seems intuitive for the base measure to decouple $(s,a)$ from $s'$ like is the case in the exponent.
>
> **Linear MDPs:** Thank you for catching the error in line 43, the comparison with linear RL is indeed not straightforward, please refer to the general comment for more details about this comparison.
>
> **Notations:** The notation for matrix A in Algorithm 1 is indeed used before its definition, we will move its definition as suggested. Thank you for catching the typo in $(\hat{\theta}^p,\tilde{\theta}^r )$, it should indeed be indexed with k, they are the parameters estimated in the same algorithm, we will modify this in the final draft. Also, $\zeta_{hk}$ being a martingale sequence follows since $V_{\hat{\theta}^p (k),\theta^r,h+1}$ depends on $\hat{\theta}^p (k)$ that comes from previous data. We apologize and we will fix these typos.
>
> **Estimation tractability:** Regarding the complexity of the maximum likelihood estimation, we know that this is tractable for simple distributions like the Gaussian and for Linearly controlled dynamical systems. For generic transitions, it may indeed requires integral approximations, however, we believe that this estimation problem is far simpler than the planning problem since the latter traditionally involves approximating an integral for all $s^{\prime}, a$. We will add a proper discussion in the paper as we feel that this is an important information to the reader.
>
> **Minor comments:** We thank you for your minor comments, we will fix mentioned typos and add the definition of the expected reward as suggested.
>
> We would like to thank you again for your careful reading and for helping us improve the quality of our writing, and we hope that you adjust your score if you believe that we clarified the issues that were raised.

---

> > ### Comment · Reviewer_rPG8 · 2022-08-05
> > **Thanks for the clarifications**
> >
> > First I thank the authors for their explanations. I still have a few questions and comments as follows:
> >
> > 1. Regarding the model of the bilinear exponential family, I agree that when $h(s', s, a) \propto h_1(s') h_2(s,a)$, everything is fine. But my concern is that when $h(s', s, a)$ cannot be decomposed in this way, then I believe we don't have Eq. (4). Is this correct?
> > 2. Regarding solving $\hat \theta^p(k)$ and $\hat\theta^r(k)$, my point is that this is the main computational overhead for the proposed algorithm. It should be discussed the computational complexity of solving these.
> > 3. For the linear MDP studied in [JYWJ20], I'm wondering why is this work model-based? Although it is assumed that the transition kernel is a linear function, their LSVI-UCB algorithm does not estimate the transition model at all but instead directly estimates the value functions. In this sense, their algorithm is model-free, the same as the BEF-RLSVI in this paper.
> > 4. It is claimed in the general comment that the analysis technique here can be applied to linear RL. One thing I'm wondering is that in the regret analysis of linear MDP in [JYWJ20], some covering argument is used in order to apply the self-normalized concentration inequality, and how is this bypassed in this paper?

---

> > > ### Author Response · Authors · 2022-08-07
> > > **Discussion with reviewer rPG8**
> > >
> > > We thank the reviewer for interacting with our rebuttal and provide some clarifications to the raised concerns.
> > >
> > > 1. **Definition of the Bilinear Exponential Family Model:** Yes, you are correct. Nonetheless, as we said before, our intuition is that the separability is not restrictive as it is verified by all the examples provided in [CGM21].
> > > 2. **Maximum likelihood estimation:** We are not sure whether this is a major computational overhead in common cases. But we agree that a discussion on this estimation complexity is important and we will add it as suggested.
> > > 3. **Model free / Model based distinction:**
> > >     - We agree that in the work of [JYWJ20] the estimation comes back to an estimation of the value function. In this sense, it can be considered as model free. However, there is a subtlety here: the setting is model-based while the algorithm is model-free.
> > >     - The setting and algorithm in our paper are both model based since we need to estimate the model and rewards to estimate the value function. We could obtain a model-free algorithm for this setting in two ways: 1) Using UCRL like [CCM21], but the planning would be intractable. 2) Applying a linear RL algorithm on the linear approximation mentioned in lines 200 to 202. However, the latter entails a regret scaling with a dimension of the approximation, which is of order $\mathcal{O}(p H^2 K)$.
> > >     - Notice that the complexity of learning the value function parameter or learning the transition and reward parameters in [JYWJ20] is similar. Also, the setting and assumptions are the same. Therefore, we don't see the advantage of a model-free algorithm over a model-based one in this case.
> > > 4. **About the covering argument** This is bypassed in our paper by using transportation inequalities, concentrations of the parameters, and a "good vs bad rounds" analysis that enables us to avoid clipping. At a higher level, we can paraphrase by saying that we first show that the "bad rounds" are finite and we analyze the good rounds by the transportation and concentration inequalities.

---

> > > > ### Comment · Reviewer_rPG8 · 2022-08-08
> > > > **Discussion with the authors**
> > > >
> > > > Thanks again for the clarifications!
> > > >
> > > > Yes, you're right that BEF-RLSVI is model-based. I apologize for my mistake.
> > > >
> > > > Regarding the computation of the maximum likelihood estimation, I took a quick look of [CCM21], and it seems that they also only mentioned that it can be solved via integral approximation in the general case, similar to the comments in line 176~177 in the current paper. I would appreciate it if the authors can provide more details on this.
> > > >
> > > > I think the discrepancy between the model assumptions here and that in [CCM21] should be noted in the paper, especially as the algorithms are compared in Table 1.

---

> > > > > ### Author Response · Authors · 2022-08-09
> > > > > **Discussion with reviewer rPG8**
> > > > >
> > > > > We thank the reviewer again for engaging with us to improve our manuscript.
> > > > >
> > > > > 1. **Computation of the maximum likelihood:** We include here some useful references to shed light about the approximation techniques that exist for the exponential family. We will add an appendix to discuss this estimation further as it seems to be more interesting than we initially believed.
> > > > >     - Integral approximation techniques:
> > > > >         1. The paper "Annealed Importance Sampling", Neal 1998', approximates ML computations using simulated annealing, a method consisting in starting from a tractable distribution and updating it sequentially to resemble the distribution at hand.
> > > > >         2. "Probabilistic Structured Predictors", Vembu et al 2012', proposes MCMC techniques for approximating the partition function.
> > > > >         3. "On Contrastive Divergence Learning", Carreira-Perpinan and Hinton 2005', shows that optimizing a different objective, called the contrastive divergence leads to a good approximation of the ML.
> > > > >     - If the support of the distribution and its natural parameter are bound, the paper "A Computationally Efficient Method for Learning Exponential Family Distributions", Shah et al 2021', shows that an $\alpha$-approximation can be derived in $\mathcal{O}(\operatorname{poly}(k/\alpha))$ time. The latter assumes a specific definition of compactness of the representation as well as knowledge of the support and shows how to re-parameterize the density to a specific class of exponential families that are easier to study.
> > > > >     - Score matching: this is a technique that avoids approximation the partition function and is well studied in literature, see "Maximum likelihood estimation and large-sample inference for generalized linear and nonlinear regression models", JØRGENSEN et al 1983'. More recently, "Exponential Family Model-Based Reinforcement Learning via Score Matching" Li et al 2021', proposed an adaptation of this technique to the exact setting we consider. The latter shows that under certain conditions, the estimation can be solved in $\mathcal{O}(d^3)$ time.
> > > > >     - "Kernel Exponential Family Estimation via Doubly Dual Embedding" Dai et al, 2019', studies exponential families such that the natural parameter belongs to some RKHS. The latter proposes a method that improves over score matching in time and memory complexity.
> > > > >
> > > > > 2. We agree that the assumptions made should be very clear to the reader. As such, we will add the separability assumption to Table 1 as suggested, and we will add a relevant discussion as well.

---

> > > > > > ### Comment · Reviewer_rPG8 · 2022-08-09
> > > > > > **Discussion with the authors**
> > > > > >
> > > > > > Thanks for the clarifications! I encourage the authors to include these details in the paper.
> > > > > >
> > > > > > I don't have any further questions, and I will adjust my rating to 6 accordingly. I hope our conversation can help you revise the paper.

---

### Official Review · Reviewer_gSYF · 2022-07-11

**Rating:** 7
**Confidence:** 4
**Soundness:** 4 excellent
**Presentation:** 4 excellent
**Contribution:** 3 good

**Summary:**

The work consider RLSVI-type algorithms on a bilinear exponential family MDP.
They make several contribution, including achieving tractable exploration.

I believe this work represents a significant improvement over existing art, both in terms of result as well as techniques involved. The work is well explained and it flows easily.

**Questions:**

See above in details

**Limitations:**

In the specific linear setting considered by the authors, only a restricted linear MDP model seems to be covered

**Strengths And Weaknesses:**

I appreciate the work: I think it is well explained, there is a solid contribution to a well defined problem.
I therefore recommend acceptance.

However, I have one major complaint / observation.
This work is model based, but the authors claim that it applies to the linear MDP model. Specifically, there is a choice that results in linear MDP. However, as I understand, not all linear MDP as defined in (Jin et al, 20’) can be solved with this algorithm. This is because fixing \phi on a linear MDP still yields many choice for \mu (proportional to the state space). In this case, one would need order S (the state space) parameters to fully describe the transition kernel of the linear MDP. Jin et al, 20’ bypasses this problem as it only needs to compute inner products, but the submission here puts a specific model on P; I would expect this model to be of order S in terms of dimensionality. Rather, the model assumed here seems to be that of Yang et al, “Reinforcement learning in feature space: Matrix bandit, kernels, and regret bound” which is far more restrictive. If that’s the case, the comparison / improvement with Zanette et al “Frequentist Regret Bounds for Randomized Least-Squares Value Iteration” is also not as clear, as the two operates in quite different models. More precisely, this work seems to be model based (as explained above); as such, it requires estimating different quantities that the work by Zanette et al. It follows that the improvement that removes the artificial clipping of the value function (and improves the bound) used in Zanette et al. may be enabled by the setting where the current algorithm operates. I think this aspect should be clarified, both in the rebuttal as well as in the paper (apologies if I missed something).

---

> ### Author Response · Authors · 2022-08-02
> **Discussion with reviewer gSYF**
>
> We would like to thank you for acknowledging our contributions, and for the kind words regarding significance of the improvements over existing art both in results and theory, and the quality of explanations.
>
> **Regarding our position with respect to literature**, please refer to the general comment where we recall some key aspects of the linear setting. For instance, we emphasize that linear RL settings (Jin et al, '20; Zanette et al, '19; Yang and Wang, '19) are all model-based, and that the literature is yet to provide a single example of a continuous state-action space MDP that is well represented by said models (with finite parameters). This being said, we would also like to acknowledge an error in line 43 of our submission, indeed we -mistakenly- stated that the bilinear exponential family model encompasses the linear one. We thank the reviewer for catching this and we ensure that this will be removed in the revised version. We also want to clarify that similar to Jin et al, 2020, the considered transition kernel is represented using the scalar products, and can therefore be fully described using an order of $d$ parameters.
>
> **Concerning the clipping improvement:** Our result to remove clipping is actually applicable for the linear setting as well. The main result that allows our improvement is Lemma 19 and, interestingly, it would be even easier to apply it to the linear setting.
> Let us explain how our enhancement follows in two steps. 1) The need to clip emerges when in order to bound the regret, we make appear a quantity of the form: $R_T \le \sum_t \langle x^* - x_t, \theta \rangle $ (see Eq 14 and Eq 15 of Zanette et al, '20, and Eq 15 of Jin et al, '19), and then use Cauchy-Schwarz to obtain $R_T \le \sum_t ||x^* - x_t ||_{ V_{t-1}^{-1}}  ||\theta||_{V_{t-1}}$. The problem is that $||x^* - x_t||_{V_{t-1}^{-1}}$ can be large. Clipping techniques handle this if we know that the rewards are bounded.
> 2) Our solution is to show that the large norms are not a real problem because they only occur a finite amount of times (see Lemma 19). Therefore, our improvement is applicable to RLSVI of Zanette et al, '20.
>
> We would like to thank you a second time for their careful review, we hope to have answered your concerns and we pledge to include relevant discussions in the revised version. If you feel that your concerns have been answered, we would appreciate it if you can adjust your score accordingly.

---

### Author Response · Authors · 2022-08-02
**General comment**

We would like to thank the reviewers for acknowledging the strengths and soundness of the contribution as well as for their thoughtful comments and efforts towards improving our manuscript. In the following, we highlight general concerns of reviewers that were common and our effort to address these concerns. We then address comments specific to each reviewer by responding to them directly.

We would like to highlight the contrast with the linear RL literature. By definition, a linear MDP (cf [JYWJ20], [ZBB+20]) is such that: for each $t \in[H]$ there exist a feature map $\psi_{t}: \mathcal{S} \rightarrow \mathbb{R}^{d}, s \mapsto \psi_{t}(s)$ and a parameter $\theta_{t}^{r} \in \mathbb{R}^{d}$ such that:
$r_{t}(s, a)=\phi_{t}(s, a)^{\top} \theta_{t}^{r}$ and $P_{t}\left(s^{\prime} \mid s, a\right)=\phi_{t}(s, a)^{\top} \psi_{t}\left(s^{\prime}\right)$. The only difference between [JYWJ20] and [ZBB+20] is that the latter assumes known feature $\psi_t$ while the former does not. Note that the setting of (Yang and Wang, 2019a) is very similar since they assume a bilinear transition: $P(s' \mid s, a) = \phi(s, a)^{\top} M^* \psi(s')$ where $M^*$ is the parameter.

We can now clarify certain confusions:

**First**, we highlight that this entire line of work is model-based since it explicitly assumes a model on the transition.

**Second**, although it can be appealing to assume a linear transition model (due to its interplay with the Bellman operator), assuming a *finite* dimensional linear transition model has been acknowledged to be a stringent, and not practical assumption. Indeed, this model was only shown to capture tabular MDPs and we have yet to see any concrete example of continuous state-action spaces that it is able to capture efficiently (that is, with few parameters).

**Third**, while the bilinear exponential family model is very different, we insist that our proof techniques also hold for linear MDPs: Indeed, **A)** our analysis uses transportation inequalities (Lemma 13) that elegantly bound our regret by the complexity of learning a bilinear form (the exponent of the transition model), **B)** Controlling the latter is like controlling the regret in linear MDPs, and in both cases it proceeds similarly to the analysis of linear bandits, **C)** Our analytical improvements, *e.g.* Lemma 19 (rendering clipping unnecessary) and Lemma 18, intervene in step *B* of the analysis, which is identical in linear RL. Consequently, the contributions of Lemma 18 and Lemma 19 also hold for the analysis of Linear RL algorithms.

---

> ### Comment · Area_Chair_ZgFv · 2022-08-05
> **Related works missing**
>
> It seems that, after revision, a number of important related works on model-based regret are are still missing:
>
> 1) Yang and Wang, 2019. Reinforcement learning in feature space: Matrix bandit, kernels, and regret bound
> It studies a bilinear family, including both finite-dim models and infinite-dim kernel models.
>
> 2) Ayoub, et al. Model-Based Reinforcement Learning with Value-Targeted Regression, 2020.
> It provides a regret for a general nonlinear model family. Shows that the regret scales linearly with model dimension. This result seems to subsume results of this paper.
>
> 3) Foster et al. The Statistical Complexity of Interactive Decision Making, 2022.
> This paper provides a general framework for regret analysis, and one section of it specifically shows how to apply to bilinear class MDP.
>
> 1,2) appear to be the closest related work to the submission. It is a bit disappointing that authors didn't mention the connections at all.

---

> > ### Author Response · Authors · 2022-08-07
> > **Discussion with Area Chair ZgFv**
> >
> > We thanks the Area Chair ZgFv for his comments and mentioning the relevant references. Here, we add brief discussions on these references in comparison with our work. We will add them further in the final version.
> >
> > 1. **Yang and Wang, 2019:** We agree that this is an important related work, and we will discuss its relation with our work in the revised version. In particular we will discuss:
> >    - *Similarity in setting:* their RKHS setting is close to ours and can recover our value function.
> >    - *Optimality of our regret bound:* applying their result yields a regret bound $H \log(T)^d$ higher than ours. The $\log(T)^d$ is the order of the information gain, see "Gaussian process optimization in the bandit setting: No regret and experimental design" Srinivas et al. 09'. Whereas our result is very close to an existing lower bound, see lines 164 to 169.
> >     - *Difference in setting:* Their RKHS result dose not recover our result, the explanation is that they only assume knowledge of the RKHS and not the precise features like in their finite dimensional result. More precisely, they have to deal with estimation in an RKHS, which is not similar to this paper nor to their linear setting. Estimation in an RKHS can be computationally inefficient and incurs a larger regret.
> > 2. **Ayoub et al, 2020:**
> >     - *Similarity in setting:* their kernel version can recover our setting like that of Yang and Wang, 2019. Their setting seems realistic as well. Indeed, they claim that a special kind for queuing networks admits a discrete-time Bernoulli approximation, which is well recovered by their assumptions. However, the quality of this approximation is not discussed therefore we are unsure of its theoretical validity.
> >     - *Differences in the analysis:* They use an Eluder-dimension based analysis, therefore, in our case it reduces to RKHS analysis similar to Yang and Wang, 2019. Indeed, "A Short Note on the Relationship of Information Gain and Eluder Dimension" by Huang et al, 2021 showed that for RKHS settings, Eluder dimension and the information gain are strictly equivalent. And while the discussion of bounds using Eluder dimension is out of the scope of our paper, we recall from Huang et al, 2021, verbatim, that "Eluder dimension was originally proposed as a general complexity measure of function classes, but the common examples of where it is known to be small are function spaces (vector spaces)".
> >     - *Intractable planning:* Planning is not tractable with UCB-approaches, and Eluder dimension analysis cannot work in our setting since it assumes finite covering numbers, it is not clear whether the latter can be modified to work for us since our value is not bounded as it is not clipped. This is rather a significant contribution of our paper as we avoided the non-linear behavior of value functions emerging from UCB-style algorithms thanks to optimism and clipping.
> >     - *Conclusion:* We respectfully disagree that the said paper subsumes our results. In fact, the regret bound achieved by their analysis in our setting is $\log(T)^d \sqrt{H}$ higher to ours. Also, their algorithm's planning is intractable as opposed to ours.
> >     - *Closely related to the area chair's request:* We have discussed (lines 256 to 260) the relationship of our paper with existing work that generalize linear RL by assuming that the considered value functions belong to some RKHS or to some space with bounded Eluder dimension.
> > 3. **Foster et al, 2022:** We refer to lines 273 to 276. We have already included a discussion about this paper.
> >
> > We would like to thank the Area Chair a second time for interacting with us and we hope that our response can clarify the missing connections.

---

> > > ### Comment · Area_Chair_ZgFv · 2022-08-08
> > > **re**
> > >
> > > Thanks for the clarifications. It helps understanding. Note that these detailed discussions should be in the paper.
> > >
> > > - It is easy to show that approximate log covering number for RKHS reduces to effective dimension of RKHS. It is not correct to say that Eluder dimension analysis doesn't work in the submission's setting.
> > >
> > > We strongly encourage authors to expand the related work section, to not overlook closely related existing results.

---

> > > > ### Author Response · Authors · 2022-08-09
> > > > **Discussion with Area Chair ZgFv**
> > > >
> > > > Thank you for your helpful input and for engaging with our rebuttal.
> > > >
> > > > Regarding our claim that the Eluder dimension analysis cannot work in our setting: We apologize for our choice of words, indeed, we meant that the Eluder dimension analysis isn't suitable for our algorithm and not for the setting as a whole. We explain our claim hereafter:
> > > > 1. Ayoub et al, 20': it is assumed that the parameter space has a finite diameter (see Corollary 2), and the latter is essential to obtaining a finite covering number. Indeed, since the UCB approach transfers the boundedness to the estimated parameter, then the estimated value function space has a finite covering as well.
> > > > 2. Yang and Wang, 2019': It uses this boundedness as well in Eq. 5 therein.
> > > > 3. For BEF-RLSVI: since we inject a noise in the estimation and we don't use clipping, the parameter $\tilde{\theta}$ is unbounded. Therefore we cannot readily apply the proof techniques from the literature, and we are unsure how to adapt them to obtain a finite effective dimension or Eluder dimension.
> > > >
> > > > If you believe that the Eluder dimension proof argument can still be used, please don't hesitate to share any intuition about adapting the analysis for our algorithm.
> > > >
> > > > We would like to thank you again for your help in improving the quality of our submission. We will add all the discussions of this rebuttal in the revised version.

---

### Meta-Review · Area_Chair_ZgFv · 2022-09-02

**Recommendation:** Reject
**Confidence:** Less certain

**Metareview:**

The paper presents a tractable algorithm for bilinear exponential MDP with regret bound that improves from the best known result and achieves \sqft{d^3 HK} regret. The result appears to be correct with strong technical analysis. Reviewers and ACs appreciate merits of the analysis for this specific problem class.

However, both the reviewer team and the AC found that the authors miss to discuss several important and closely related works, such as Zanette et al, '19; Yang and Wang, '19 and a line of works on kernel RL and model-based RL with Eluder dimension analysis. In particular, Table 1 only compares the new result with several recent results on specific MDP models published after 2021, which is far from comprehensive. During the rebuttal, the authors acknowledged that they were not aware of these related works. However,  they didn’t revise the submission to include the missing discussions pointed by the reviewer.

It remains unclear how the submission’s analysis relates to the aforementioned results that were not discussed in the paper. The authors provided some high-level discussion after rebuttal, but they would need a lot more technical details to be convincing. For example, regret analysis using Eluder dimension for general function class is often a go-to benchmark for non-linear models. The proposed model appears to be a generalized linear model, which is a standard special case of the Eluder dimension analysis. Then one would expect such analysis to lead to a O(d poly(H)\sqrt{T}) regret, (with \sqrt{d} coming from Eluder dimension and \sqrt{d} coming from metric dimension), better than result of this paper.  Note that this is just a conjecture, and rigorously working out this analysis would likely need extra work (nontrivial, as the authors pointed out). However, it is still not appropriate to overlook the possibility of using a more general analysis and just focus on a specific parametric model. A careful and honest discussion is necessary.

Beyond using Eluder dimension, there are actually a handful of RL theory papers on general function approximation and general model classes. We strongly recommend the authors to redo their paper survey and properly place their contribution in the context of state-of-art RL theory. We have reviewed a very competitive batch of RL papers this year. This submission has strengths but falls on the borderline. After consulting with the senior AC member who is also expert in RL theory, we regretful commend the authors further revise the paper and submit to the next venue.

**Award:**

No

---

### Decision · Program_Chairs · 2022-09-14

Reject